# E3 ubiquitin ligase CHIP facilitates cAMP and cGMP signalling cross-talk by polyubiquitinating PDE9A

Xiaoyan Hao[1,11], Zhengwei Hu[1,11], Mengjie Li[1], Shuo Zhang[1], Mibo Tang[1], Chenwei Hao[1], Shasha Qi[1], Yuanyuan Liang[1], Michael F Almeida [ID][2], Kaitlan Smith [ID][2], Chunyan Zuo[1], Yanmei Feng[1], Mengnan Guo[1], Dongrui Ma[1], Shuangjie Li [ID][1], Zhiyun Wang[1], Yuemeng Sun[1], Zhifen Deng[1], Chengyuan Mao[1,3,4,5], Zongping Xia[1], Yong Jiang [ID][6,7,8], Yanxia Gao[6,7,8,9], Yuming Xu [ID][1,3,4,5 ✉], Jonathan C Schisler [ID][2 ✉] & Changhe Shi [ID][1,3,4,5,10 ✉]

## Abstract

The carboxyl terminus of Hsc70-interacting protein (CHIP) is pivotal for managing misfolded and aggregated proteins via chaperone networks and degradation pathways. In a preclinical rodent model of CHIP-related ataxia, we observed that CHIP mutations lead to increased levels of phosphodiesterase 9A (PDE9A), whose role in this context remains poorly understood. Here, we investigated the molecular mechanisms underlying the role of PDE9A in CHIP-related ataxia and demonstrated that CHIP binds to PDE9A, facilitating its polyubiquitination and autophagic degradation. Conversely, dysfunctional CHIP disrupts this process, resulting in PDE9A accumulation, increased cGMP hydrolysis, and impaired PKG phosphorylation of CHIP at serine 19. This cascade further amplifies PDE9A accumulation, ultimately disrupting mitophagy and triggering neuronal apoptosis. Elevated PKA levels inhibit PDE9A degradation, further exacerbating this neuronal dysfunction. Notably, pharmacological inhibition of PDE9A via Bay 73-6691 or virus-mediated CHIP expression restored the balance of cGMP/cAMP signalling. These interventions protect against cerebellar neuropathologies, particularly Purkinje neuron mitophagy dysfunction. Thus, PDE9A upregulation considerably exacerbates ataxia associated with CHIP mutations, and targeting the interaction between PDE9A and CHIP is an innovative therapeutic strategy for CHIP-related ataxia.

**Keywords** CHIP; cGMP-cAMP Signalling Crosstalk; PDE9A; Ubiquitin-lysosome Pathway
**Subject Categories** Molecular Biology of Disease; Post-translational Modifications & Proteolysis; Signal Transduction

## Introduction

*STUB1* encodes the carboxyl terminus of the Hsc70-interacting protein (CHIP), which has dual-molecular chaperone function and E3 ubiquitin ligase activity (Narayan et al, 2015; Stankiewicz et al, 2010; Zhang et al, 2020b). In the brain, CHIP ensures the correct trafficking of aberrant proteins to the ubiquitin‑proteasome or autophagy‑lysosome systems, thus maintaining cellular function and preventing substrate protein dysfunction or toxicity (Chakraborty and Edkins, 2022; Sun-Wang et al, 2021). Investigations employing animal models with genetic functional loss of *CHIP* have revealed exacerbated cardiac responses to haemodynamic or ischaemic stress, accelerated ageing and neurodegenerative pathologies (Ranek et al, 2020).

Our previous research revealed the first *CHIP* monogenetic mutations in a family with cerebellar ataxia, cognitive impairment, and hypogonadism; this led to the classification of a new autosomal recessive form of spinocerebellar ataxia (SCA), known as SCAR16. Further studies have identified dominant patterns of *CHIP* mutations in ataxia patients, leading to the designation of SCA48 (Ronnebaum et al, 2014; Shi et al, 2014). Analogous to the human phenotype, *CHIP* mutation rodent models also exhibit progressive cerebellar ataxia, cerebellar neuronal degeneration and death, concomitant with a marked increase in the cerebral phosphodiesterase 9A (PDE9A) protein level, which progressively increases as the disease progresses (Shi et al, 2018b). This finding indicates that the upregulation of the PDE9A protein is a potential cause of *CHIP*

[1]Department of Neurology, The First Affiliated Hospital of Zhengzhou University, Zhengzhou University, Zhengzhou 450000 Henan, China. [2]McAllister Heart Institute and the Department of Pharmacology, The University of North Carolina at Chapel Hill, Chapel Hill, NC 27599, USA. [3]Institute of Neuroscience, Zhengzhou University, Zhengzhou 450000 Henan, China. [4]Henan Key Laboratory of Cerebrovascular Diseases, The First Affiliated Hospital of Zhengzhou University, Zhengzhou University, Zhengzhou 450000 Henan, China. [5]NHC Key Laboratory of Prevention and Treatment of Cerebrovascular Diseases, The First Affiliated Hospital of Zhengzhou University, Zhengzhou University, Zhengzhou 450000 Henan, China. [6]State Key Laboratory of Antiviral Drugs, The First Affiliated Hospital, Zhengzhou University, Zhengzhou 450000, China. [7]Henan Key Laboratory of Critical Care Medicine, Department of Emergency Medicine, The First Affiliated Hospital, Zhengzhou University, Zhengzhou 450000, China. [8]Institute of Infection and Immunity, Henan Academy of Innovations in Medical Science, Zhengzhou 450000, China. [9]Department of Emergency Medicine, The First Affiliated Hospital of Zhengzhou University, Medical Key Laboratory of Poisoning Diseases of Henan Province, Zhengzhou, China. [10]Tianjian Laboratory of Advanced Biomedical Sciences, Zhengzhou University, Zhengzhou 450000 Henan, China. [11]These authors contributed equally: Xiaoyan Hao, Zhengwei Hu. ✉E-mail: xuyuming@zzu.edu.cn; schisler@unc.edu; shichanghe@zzu.edu.cn

mutation-induced pathogenesis, but the underlying mechanisms remain elusive.

Notably, PDE9A, which is predominantly expressed in brain tissues, exhibits high selectivity and affinity for cGMP (Wang et al, 2017). Cyclic guanosine monophosphate (cGMP) is a key signalling molecule in various cellular contexts that is capable of activating protein kinase G (PKG) and phosphorylating downstream target proteins, including ion channels, kinases, and substrates (Yang et al, 2013). Through its interaction with cGMP, the binding domain of PKG binds to cGMP, activates the catalytic domain of PKG, and affects the phosphorylation state of various effector molecules, thereby playing a crucial role in neurodegenerative diseases (NDDs) (VerPlank et al, 2020). The catalytic site of PDE9A binds cofactors and hydrolyses cGMP, leading to the inactivation of PKG and the inhibition of target protein phosphorylation, thereby influencing multiple cellular processes. In addition, cGMP signalling often interacts with cyclic adenosine monophosphate (cAMP) signalling to maintain intracellular signal transduction (Zaccolo and Movsesian, 2007).

Moreover, recent discoveries have shown that PKG directly phosphorylates CHIP at serine 19 to increase the CHIP level and extend its half-life; this phosphorylation (CHIP-pS19) increases CHIP stability and enhances its affinity for HSP70 (Ranek et al, 2020). CHIP ubiquitinates the catalytic subunit of protein kinase A (PKAc) in a chaperone-dependent manner, leading to the proteasomal degradation of PKAc. This process attenuates the cAMP–PKA signalling pathway (Rinaldi et al, 2019). Therefore, *CHIP* mutation may affect the intracellular alterations of cGMP/cAMP signalling pathways.

To investigate the possible role of PDE9A in CHIP-associated ataxia, we tested whether PDE9A serves as a ubiquitination substrate for CHIP. Elevated PDE9A in CHIP-mutated brains results in the downregulation of cGMP and inactivation of PKG, further directly diminishing the CHIP level and its phosphorylation. This finding suggests that CHIP plays a central role in cGMP/cAMP signalling crosstalk. CHIP mutation prompts the upregulation of cAMP, increasing PKAc and further competitively inhibiting the ubiquitin-mediated degradation of PDE9A. Pharmacological inhibition of PDE9A- or AAV-mediated CHIP expression alleviates neurotoxic damage in *CHIP* mutation models, offering new therapeutic avenues for CHIP-related neurodegenerative diseases.

## Results

### Reciprocal regulation and interaction of CHIP and PDE9A

PDE9A is a 593 amino acid protein with an N-terminal regulatory site and a C-terminal catalytic domain. CHIP operates in the protein quality control system through its bifunctional TPR and U-box domains (Omori and Kotera, 2007). CHIP is fundamentally structured around three major domains, TPR (26–127 aa), CC (128–226 aa), and the U-box (227–300 aa), and a mutation at position p.246M in *CHIP* results in inactivation of its E3 ligase activity while preserving its cochaperone functionality (T246M). Conversely, the K30A mutation disrupts the binding functionality of the TPR domain to proteins such as HSP70 and affects the E3-ligase activity of CHIP (K30A) (Fig. 1A). Previous studies have

demonstrated that inactivating mutations in *CHIP* lead to increased PDE9A levels. To elucidate the specific regulatory mechanisms by which a reduction in CHIP levels results in increased PDE9A levels, we aimed to determine the dose–response relationship and directionality of this regulation. Moreover, we sought to verify whether the increase in PDE9A protein levels induced by CHIP is due to enhanced synthesis or reduced degradation. Using exogenous expression of CHIP and PDE9A in a cell culture system, we found that CHIP decreased the level of PDE9A in a dose-dependent manner (Fig. 1B), whereas PDE9A upregulation decreased CHIP levels also in a dose-dependent manner (Fig. 1C). The reduction in steady-state protein levels was not reflected at the mRNA level, suggesting that mutual negative regulation between PDE9A and CHIP occurs at the post-translational level (Fig. 1D–G; Appendix Fig. S1A,B). Furthermore, we observed colocalisation of CHIP and PDE9A in the cytoplasm (Fig. 1H).

To confirm that the PDE9A and CHIP proteins may regulate intracellular protein homeostasis posttranslationally through mutual inhibition of their degradation pathways, we utilized HEK293T cells to exogenously express the PDE9A and CHIP proteins. We observed that the half-life of CHIP is approximately six hours, which increased substantially in cells treated with the proteasome inhibitor bortezomib rather than the autophagy inhibitor bafilomycin A1, indicating proteasome-dependent degradation of CHIP. This aligns with previous findings that delineate proteasomal degradation of CHIP, thereby reaffirming the estimated six-hour half-life of CHIP (Naito et al, 2010). In contrast, the four-hour half-life of PDE9A was attenuated in cells treated with the autophagy inhibitor bafilomycin A1 but remained unaffected by bortezomib treatment, indicative of autophagy-mediated degradation of PDE9A (Fig. 1I). Meanwhile, CHIP increased the autophagy-mediated degradation of PDE9A, reducing its half-life. Likewise, PDE9A enhanced the proteasomal degradation of CHIP, shortening its half-life (Appendix Fig. S1C). To further confirm the negative regulatory effect of PDE9A activity on the half-life of CHIP, we treated CHIP cells with the PDE9A inhibitor Bay 73-6691, which resulted in decreased PDE9A and extended the half-life of CHIP (Fig. 1J). This illustrates the reciprocal inhibitory effects between CHIP and PDE9A, with CHIP promoting the autophagic degradation of PDE9A and PDE9A inhibitors suppressing the proteasomal degradation of CHIP.

Removing the TPR domain rendered both the GC and U-box domains incapable of interacting with PDE9A. However, upon excision of the U-box domain, PDE9A had a pronounced interaction with the TPR domain, confirming that PDE9A chiefly binds with the TPR domain of CHIP (Fig. 2A). CHIP facilitates the degradation of PDE9A through the autophagy–lysosomal pathway. In addition, the interaction between CHIP and HSP70 enhances the lysosomal degradation of substrate proteins (Chakraborty and Edkins, 2022; Kumar et al, 2022). We predicted that PDE9A interacts with HSP70 via hydrogen bonding (Appendix Fig. S2A,B, Appendix Table S1). Moreover, HSP70 colocalises with PDE9A and the TPR domain of CHIP in the cytoplasm, indicating the potential formation of a complex between PDE9A, HSP70, and the TPR domain of CHIP; this complex likely promotes the recognition and binding of PDE9A by CHIP (Fig. 2A; Appendix Fig. S2C–S2D). Collectively, our findings demonstrate that CHIP interacts with PDE9A through its TPR domain.

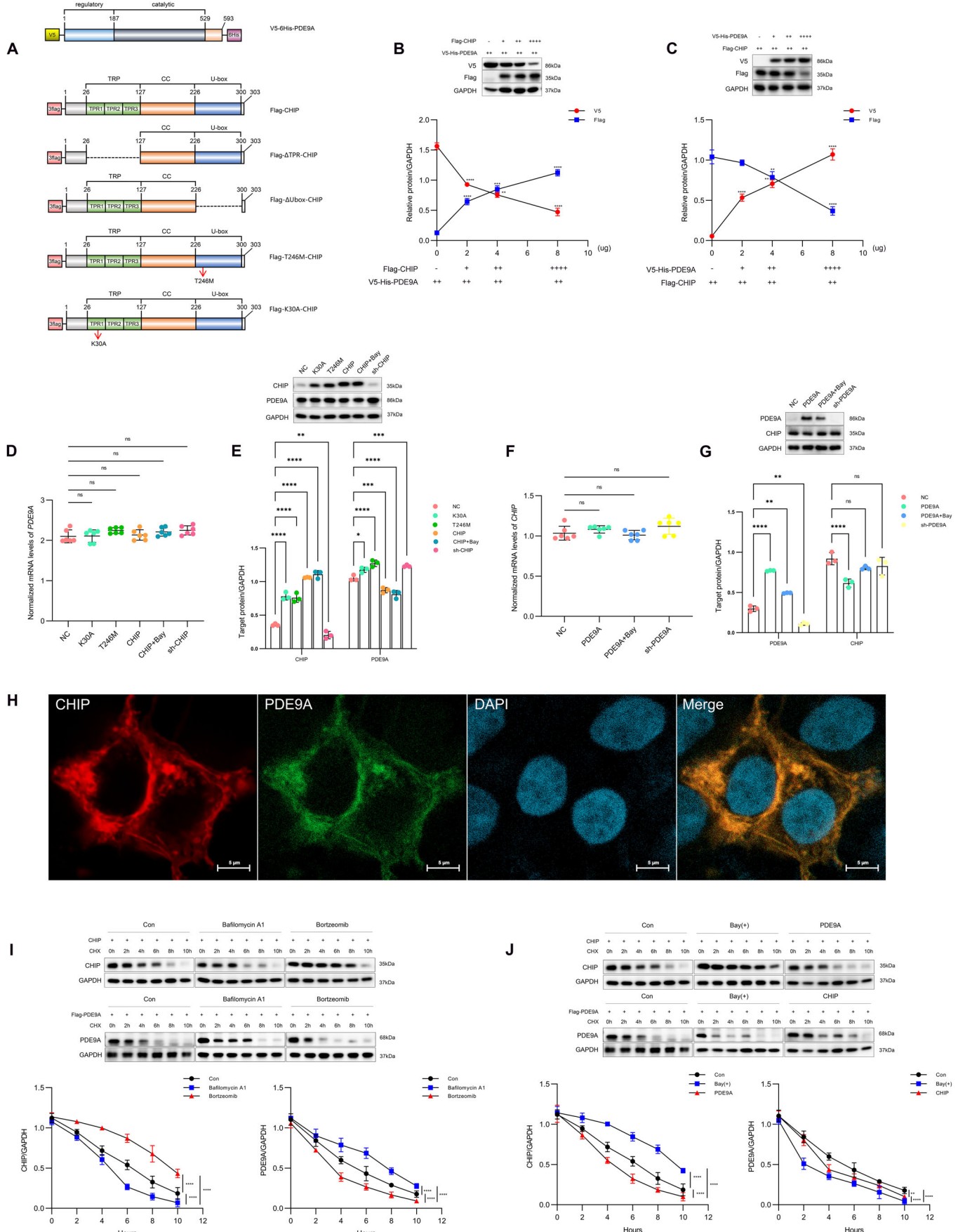

Figure 1.  Bidirectional degradation regulation between the PDE9A and CHIP proteins.

(A) Schematic representation of PDE9A and CHIP structures; ΔTPR signifies the TPR domain of CHIP deletion, and ΔU-box indicates the U-box domain of CHIP deletion. The T246M mutation at the p.T246M locus results in inactivation of the U-box domain of CHIP, whereas the K30A mutation at the p.K30A locus leads to inactivation of the TPR domain. (B) Upper: In a set of experiments involving four 10 cm diameter Petri dishes for plasmid transfection, each dish was transfected with a total of 12 μg of plasmid DNA, followed by cotransfection of 4 μg of *Flag-CHIP* plasmid with increasing contents of *V5-His-PDE9A* (0, 2, 4, and 8 μg) and decreasing contents of blank-vector plasmid (8, 6, 4, and 0 μg) into HEK293T cells. This was followed by western blot analysis to examine the CHIP-mediated regulation of PDE9A levels. Lower: summary data, $n = 3$ biological replicates/group, two-way ANOVA (2WANOVA) for each condition, 0 vs. 2, 2 vs. 4, 4 vs. 8, Tukey's multiple comparisons test (mct), V5: ****$P < 0.0001$ for all comparisons, **$P = 0.0025$; Flag: ****$P < 0.0001$ for all comparisons, ***$P = 0.0005$, $P > 0.05$ no marks. (C) Similar methodology. Upper: HEK293T cells were cotransfected with 4 μg of *V5-His-PDE9A* plasmid with increasing concentrations (0, 2, 4, and 8 μg) of *Flag-CHIP* and decreasing concentrations of blank-vector plasmid (8, 6, 4, and 0 μg), followed by western blot analysis to elucidate PDE9A-mediated alterations in CHIP. Lower: summary data, $n = 3$ biological replicates/group. 2WANOVA, Tukey mct, V5: ****$P < 0.0001$ for all comparisons, **$P = 0.0095$; Flag: **$P = 0.0069$, ****$P < 0.0001$, $P > 0.05$ no marks. (D) Changes in *PDE9A* mRNA levels following transfection with various *CHIP* mutant plasmids involved transfecting each well of a six-well plate with 2 μg of blank-vector plasmid, along with 2 μg of each mutant CHIP variant, into HEK293T cells; NC consisted of 2 μg of blank-vector plasmid, T246M represents homozygous mutations at the p.T246M site, K30A represents homozygous mutations at the p.K30A site, and CHIP+Bay indicates cells overexpressing CHIP treated with 200 μg/mL Bay 73-6691 (PDE9A inhibitor, dissolved in cell culture medium using a sonicator); sh-CHIP represents shRNAs targeting *CHIP* mRNA expression, summary data, $n = 6$ biological replicates/group, 1WANOVA, Tukey mct, $^{ns}P > 0.05$ for all comparisons (E) Changes in PDE9A protein levels following transfection with various *CHIP* mutant plasmids involved transfecting each well of a six-well plate with 2 μg of *PDE9A* plasmid, along with 2 μg of each *CHIP* variant, into HEK293T cells. Upper: Blots showing the *CHIP* mRNA variation affecting subsequent PDE9A protein levels. Lower: summary data, $n = 3$ biological replicates/group, 2WANOVA, Tukey mct, CHIP: ****$P < 0.001$ for all comparisons, **$P = 0.0051$; PDE9A: *$P = 0.0274$, ****$P < 0.0001$ for all comparisons, ***$P = 0.0010$, ***$P = 0.0007$. (F) Changes in *CHIP* mRNA levels following transfection with *PDE9A* mutant plasmids, cotransfection of 2 μg of various *PDE9A* mutant plasmids and 2 μg of blank-vector plasmid, PDE9A+Bay represents HEK293T cells overexpressing *PDE9A* treated with 200 μg/mL Bay 73-6691; *sh-PDE9A* indicates RNAi-mediated *PDE9A* knockdown. Summary data: $n = 6$ biological replicates/group, 1WANOVA, Tukey mct, $^{ns}P > 0.05$ for all comparisons. (G) Changes in CHIP protein levels following transfection with *PDE9A* mutant plasmids, cotransfection of 2 μg of various *PDE9A* mutant plasmids and 2 μg of *CHIP* plasmid into HEK293T cells. Upper: Blots showing the *CHIP* mRNA variation affecting subsequent PDE9A protein levels. Lower: summary data, $n = 3$ biological replicates/group. 2WANOVA, Tukey mct, PDE9A: ****$P < 0.0001$, **$P = 0.0017$, **$P = 0.0021$; CHIP: ****$P < 0.0001$, $^{ns}P = 0.0681$, $^{ns}P = 0.1775$. (H) After HEK293T cells were cotransfected with 2 μg of *CHIP-HA* and 2 μg of *PDE9A–Flag* plasmids, immunofluorescence staining was used to observe the colocalization of PDE9A and CHIP within the cytoplasm. Imaging was performed through multiphoto laser scanning microscopy. Experiment replicated × 3. CHIP (red), PDE9A (green), and DPAI (blue). (I) Cotransfections were performed in each well of a six-well plate by introducing 2 μg of *CHIP-HA* and 2 μg of empty vector plasmid or 2 μg of *PDE9A–Flag* and 2 μg of empty vector plasmid into HEK293T cells, followed by treatment with 50 nM bafilomycin A1 (autophagy inhibitor) or 100 nM bortezomib (proteasome inhibitor, PS-341) with cycloheximide (CHX, which inhibits protein synthesis) added at six time points (0, 2, 4, 6, 8, and 10 h). Upper: Blot assessment of the half-lives and degradation pathways of CHIP and PDE9A. Lower: summary data, $n = 3$ biological replicates/group, comparing slope differences by analysis of covariance (ANCOVA), ****$P < 0.0001$ for all comparisons. (J) A similar methodology was used, followed by treatment with 200 μg/mL Bay 73-6691 and cotransfection of 2 μg *CHIP* plasmid and 2 μg *PDE9A* plasmid into HEK293T cells. Upper: Blotting was used to assess the impact of PDE9A and Bay 73-6691 on the degradation of CHIP and the impact of CHIP and Bay 73-6691 on the degradation of PDE9A. The blot and statistical values for the Control (Con) group are identical to those for the Control (Con) group in (I). Lower: summary data, ANCOVA for each condition, $n = 3$ biological replicates/group, ANCOVA, ****$P < 0.0001$ for all comparisons, **$P = 0.0038$. Each summary panel shows the means ± SDs, summary plot (D–G) and regressions (B, C, I, J). The $P$-value are shown from left to right. Source data are available online for this figure.

## CHIP mediates K63- and K27-linked polyubiquitination of PDE9A

The U-box domain governs substrate recognition, catalysing ubiquitin transfer and dictating the diversity and selectivity of ubiquitinated substrates. With E2 ubiquitin-conjugating enzymes, CHIP catalyses the polyubiquitination of substrate proteins via multiple lysine residues on ubiquitin (K6, K11, K27, K29, K33, K48 and K63). The distinct polyubiquitin chains result in various downstream consequences, including recognition by the 26S proteasome and subsequent degradation of the protein substrate and modulation of other cellular processes, including autophagy, mitochondrial function and lysosomal biosynthesis, through ubiquitin-regulated substrate activity (Chen et al, 2020; Ferreira et al, 2013; Liu et al, 2021; Vasco Ferreira et al, 2015). In particular, CHIP-mediated K48, K29 and K11 ubiquitin chains result in proteasomal degradation, K63 ubiquitination is implicated in autophagy degradation and signal transduction, and K27 ubiquitination is involved in altering mitochondrial function (Sun-Wang et al, 2021; Zhou and Zhang, 2022).

Given the established E3 ubiquitin ligase function of CHIP and its role in directing substrates to either the proteasome pathway or the autophagy pathway, we sought to elucidate the ubiquitination pattern of PDE9A within cells. We initially confirmed that CHIP mediates the polyubiquitination of PDE9A (Appendix Fig. S3A,B). To identify the ubiquitin chain linkages on PDE9A, we

subsequently utilized a panel of ubiquitin mutants that allowed single lysine residues to form chains or lack specific lysines (Fig. 2B). The results revealed polyubiquitin chains at the K48, K63, K11, K27 and K29 lysine residues, with the K11-linked ubiquitin chain being synthesised to a lesser extent (Fig. 2C; Appendix Fig. S3C). The K63R and K27R mutations abrogated ubiquitination, indicating that CHIP directly catalyses K63- and K27-linked PDE9A ubiquitination (Fig. 2D; Appendix Fig. S3D). In addition, PDE9A, comprising 593 amino acids and 30 lysine residues, was analysed via LS/MS to identify the CHIP-mediated ubiquitination site at the 186th lysine residue of PDE9A (Fig. 2E). In addition, a comparative analysis of ubiquitination patterns between wild-type PDE9A (PDE9A) and a mutant variant (PDE9A-K186R), in the context of CHIP and various ubiquitin forms (Ub-WT, Ub-K27 and Ub-K63), demonstrated that robust polyubiquitination of PDE9A is dependent on K186 (Fig. 2F; Appendix Fig. S3E). Mutation of PDE9A at K186R disrupted CHIP-mediated ubiquitination, resulting in an increased half-life and overaccumulation of PDE9A in cells (Appendix Fig. S3F).

These results confirmed that PDE9A interacts with the TPR domain. In-cell ubiquitination assays revealed that wild-type CHIP (CHIP) enhances PDE9A ubiquitination, as indicated by the presence of a high-molecular-weight ubiquitin smear on immuno-precipitated PDE9A. This process requires the intact TPR and U-box domains of CHIP, as confirmed by domain deletion experiments (Appendix Fig. S3G,H). Concomitantly, disruptive

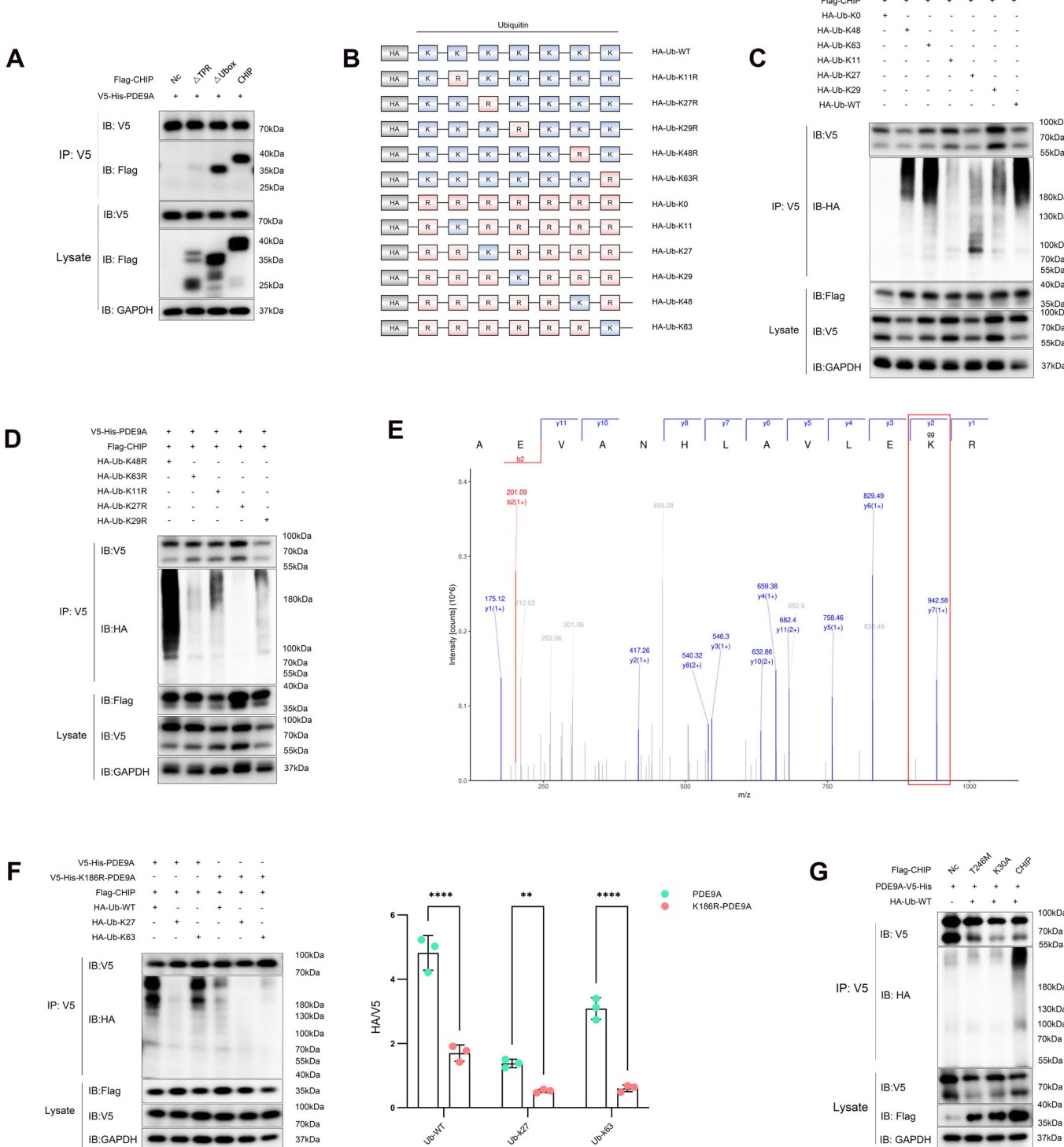

mutations in the TPR domain (K30A) and the T246M mutation corroborated the role of CHIP in PDE9A polyubiquitination (Fig. 2G; Appendix Fig. S3I).

These results suggest that, in the recruitment of K63 and K27 ubiquitin chains to PDE9A, CHIP triggers its sequential ubiquitination and subsequent degradation via the autophagy–lysosomal pathway.

## PDE9A inhibitors and exogenous expression of CHIP alleviates symptoms in CHIP-inactive mutation animal models

Preliminary findings from our recent research show striking similarities between the *CHIP* mutant rodent model and human *CHIP* mutation phenotypes. Both exhibit a progressive ataxia

**Figure 2. CHIP directly mediates K63- and K27-linked polyubiquitination of PDE9A.**

(A) Coimmunoprecipitation assays to identify interacting domains between PDE9A and CHIP. Antibodies: V5-rabbit and Flag-mouse. Experiment replicated × 3. (B) Schematic of ubiquitin variants. (C, D) In vivo ubiquitination assays validating the role of different ubiquitin chain mutants in the CHIP-mediated ubiquitination of PDE9A. HEK293T cells were cotransfected with 6 μg of *V5-His-PDE9A* plasmid, 6 μg of *Flag-CHIP* plasmid, and 6 μg of ubiquitin variants in 10 cm diameter Petri dishes. Polyubiquitin chains are generated with K48, K63, K11, K27 and K29 linkages, whereas synthesis is impaired for the K63R and K27R chains. Experiment replicated × 3. Antibodies: V5-rabbit, HA-mouse, and Flag-mouse. (E) LS/MS detection of ubiquitination at position K186 in the PDE9A sequence AEVANHLAVLEKR (indicated within the red box). Experiment replicated × 2. (F) Similar to in vivo ubiquitination assays, HEK293T cells were cotransfected with 6 μg of *V5-His-PDE9A* plasmid or 6 μg of *V5-His-PDE9A-K186R* plasmid, 6 μg of *Flag-CHIP* plasmid, and 6 μg of ubiquitin mutant variant plasmid. Experiment replicated × 3. Antibodies: V5-rabbit, HA-mouse, and Flag-mouse. Paired t test, ****$P < 0.0001$ for all comparisons, **$P = 0.0098$. (G) Similar to the methodology used for in vivo ubiquitination assays, HEK293T cells were cotransfected with 6 μg of the *V5-His-PDE9A* plasmid, 6 μg of the site-specific *CHIP* mutation plasmid, and 6 μg of the *HA-Ub* plasmid. Site-specific mutations p.T246M and p.K30A in *CHIP* result in impaired PDE9A ubiquitination in vivo. Experiment replicated × 3. Each summary panel shows the means ± SDs and summary plot (G). The *P*-value are shown from left to right. Source data are available online for this figure.

phenotype accompanied by cerebellar neuronal degeneration and death. Concurrently, overt upregulation of PDE9A levels within the rodent brain, which progressively intensifies with disease progression, has been observed (Shi et al, 2018b). These intriguing observations suggest that the increase in PDE9A protein levels is pivotal to the pathogenesis of *CHIP* mutations, although the underlying mechanisms remain unclear. Bay 73-6691, an inaugural phosphodiesterase inhibitor selective for the PDE9A subtype, has previously been employed in studies to ameliorate memory deficits and cognitive impairments in Alzheimer's disease. Its efficacy is attributed primarily to its capacity to enhance the retention of long-term potentiation, bolster synaptic plasticity in the hippocampus and mitigate the detrimental impact of toxic protein aggregation on hippocampal neurons (Miguel et al, 2009; Rombaut et al, 2021; Zheng and Zhou, 2023).

Drawing inspiration from our preliminary findings, we investigated whether the PDE9A inhibitor is therapeutic for *CHIP* homozygous mutation models. Our previous studies established that mutant rats accurately reproduce the human phenotypic manifestations associated with *CHIP* monogenetic mutations, developing overt symptoms of ataxia and memory impairments by six months of age, accompanied by the degeneration of neurons in the cerebellum (Shi et al, 2018a; Shi et al, 2014). We hypothesized that inhibiting PDE9A activity or decreasing PDE9A steady-state protein levels via exogenous CHIP expression could facilitate the loss of sensory–motor and cognitive function. We applied a pharmacological and genetic approach with behaviour assays to test our hypothesis (Fig. 3A).

Consequently, we used the tool compound Bay 73-6691, a phosphodiesterase inhibitor that is selective for the PDE9A subtype. We administered Bay 73-6691 (2 mg/kg/day i.p.) to 5.5-month-old homozygous p.T246M mutant male rats for a 15-day treatment period (Hom(6M⁺)+Bay(+)). Behavioural assessments were performed at 6 months of age and compared with those of age-matched untreated homozygous mutants (Hom(6M⁺)), wild-type littermates (WT), and presymptomatic 4-month-old homozygous rats, which were used as controls for behavioural alterations at the age of six months (Hom(6M⁻)). By six months of age, Hom(6M⁺) rats tolerate less speed on an accelerating rotarod (Fig. 3B) and take more time to traverse a balance beam (Fig. 3C), which is consistent with a decrease in motor coordination. Hom(6M⁺) rats treated with Bay 73-6691 performed similarly to age-matched wild-type and presymptomatic Hom(6M⁻) rats, which was consistent with improved motor function (Fig. 3B,C). Compared with age-matched wild-type rats, Hom(6M⁺) rats presented

decreased stride length, rear-base width, and right front-rear overlap by six months of age. Decreases in stride length, base width, and front-rear overlap during gait analysis can indicate various underlying conditions. A shorter stride length suggests that muscular or neurological issues affect movement. At the same time, a reduced base width could point to decreased stability or balance, and less front rear overlap might reflect alterations in coordination. Compared with those of age-matched wild-type or presymptomatic 4-month-old homozygous mutant rats, the decreases in gait parameters were reversed by treatment with Bay 73-6691 (Fig. 3D,E). These data suggest that Bay 73-6691 treatment improved motor coordination. In addition, memory enhancements were noted, as evidenced by reduced latency and increased frequency in locating the platform in the Morris water maze test (Fig. 3F–I), indicating that Bay 73-6691 combats the clinical phenotype of ARCA by inhibiting PDE9A protein accumulation-mediated toxic effects. To further assess the symptomatic improvement in homozygous mutant rats treated with Bay 73-6691, we administered the compound in three treatment cycles, each separated by a two-month interval. Compared with untreated *CHIP* homozygous mutant littermates (Hom), rats treated with Bay 73-6691 (Hom+Bay(+)) presented increased body weight and improved performance, as evidenced by decreased time on the balance beam and increased endurance in the rotarod test (Appendix Fig. S4A–C).

The inactivating mutation at the p.T246M locus of the *STUB1* gene culminates in CHIP U-box inactivation and reduced cerebral CHIP levels, thereby presenting typical clinical manifestations of ARCA-induced cerebellar ataxia. Consequently, we concurrently employed exogenous CHIP expression as a positive therapeutic control. For genetic experiments, we administered a blood–brain barrier-permeable adeno-associated virus carrying a transgene encoding the CHIP protein (AAV–CHIP) or a control virus (AAV–NC) previously developed by our group (Zhang et al, 2020a). We intravenously administered AAVs (1.2 × 10¹¹ vg/rat HBAAV2/BBB-CHIP virus/i.v.) to 5.5-month-old homozygous mutant rats. Additional cohorts included age-matched untreated homozygous mutant littermates (Hom(7M⁺)), and wild-type (WT) rats. Upon complete cerebral overexpression of exogenous CHIP in mutant rats at 7 months (Hom(7M⁺) + AAV–CHIP), the virus efficiently penetrated the brain tissue of homozygous mutant rats, increasing CHIP protein levels (Appendix Fig. S4D,E). Rats in the gene therapy study underwent the same battery of behaviour tests approximately six weeks after AAV injection. Compared with those of the homozygous mutant (Hom(7M⁺)), behavioural observations

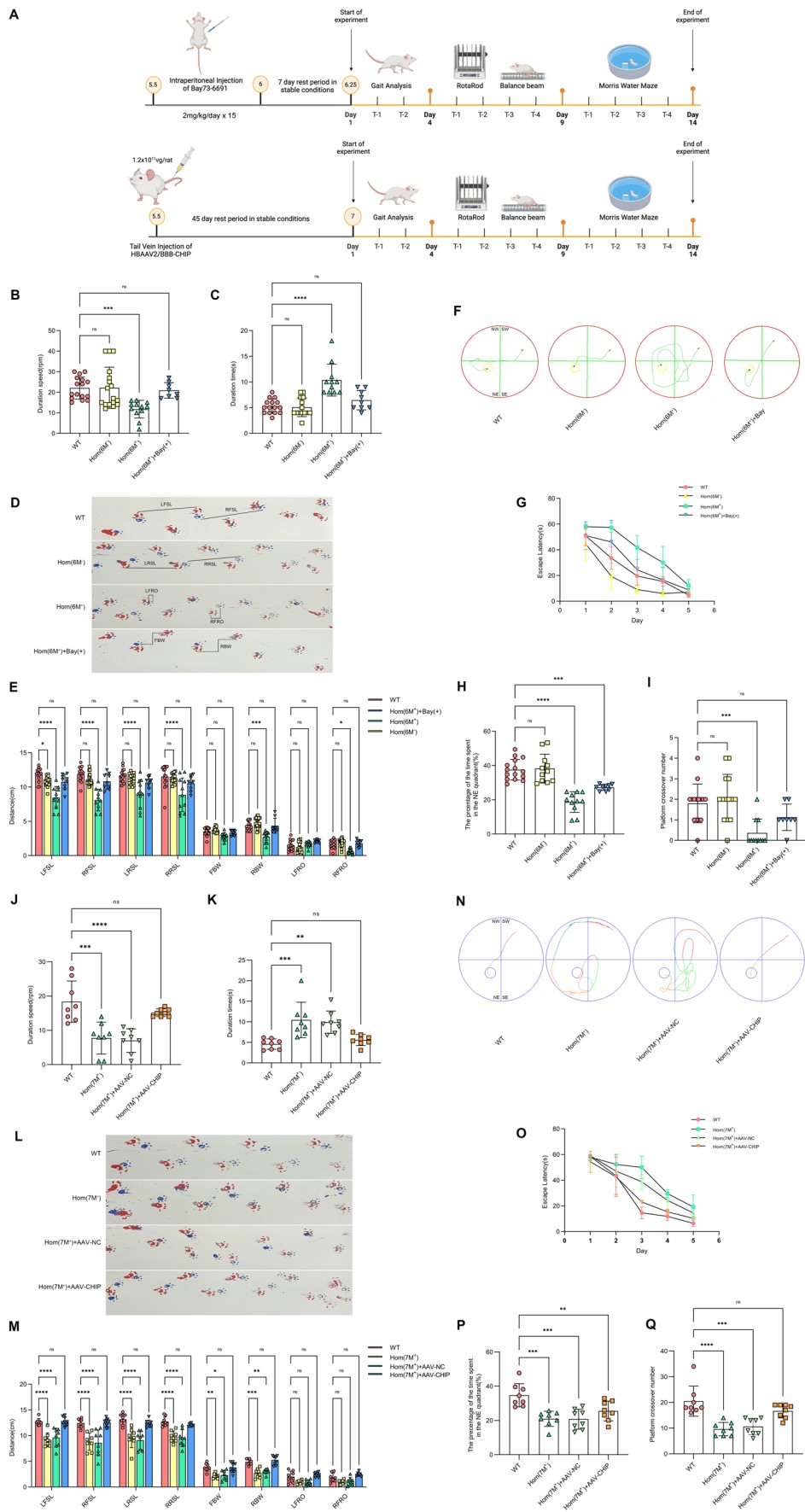

**Figure 3.   Inhibition of PDE9A activity and exogenous supplementation with CHIP ameliorate the ARCA clinical phenotype.**

(A) Timeline for behavioural assessments in the Bay 73-6691 treatment and AAV-CHIP injection cohorts. The Bay 73-6691 treatment cohort included Hom(6M$^+$) (6.25-month-old homozygous rats harbouring the p.T246M mutation), Hom(6M$^+$)+Bay(+) (Bay 73-6691 solubilized in corn oil administered intraperitoneally at 2 mg/kg to homozygous rats), Hom(6M$^-$) (presymptomatic 4-month-old homozygous rats with the p.T246M mutation), and WT (wild-type littermates). The AAV-CHIP injection cohort included Hom(7M$^+$) ($n = 8$, homozygous rats with the p.T246M mutation), Hom(7M$^+$) + AAV-CHIP ($n = 8$, tail vein injection of $1.2 \times 10^{11}$ vg/mL HBAAV2/BBB-CHIP virus to increase CHIP protein levels in the brains of homozygous rats), Hom(7M$^+$) + AAV-NC ($n = 8$, homozygous rats injected with an equivalent dose of control empty HBAAV2/BBB-blank virus) and WT ($n = 8$, wild-type littermates). (B) Tolerance to maximal velocity in the rodent rotarod test. Summary data: The Bay 73-6691 treatment cohort included WT ($n = 16$), Hom(6M$^-$) ($n = 17$), Hom(6M$^+$) ($n = 12$), and Hom(6M$^+$)+Bay(+) ($n = 9$) individuals. Each rat was tested in triplicate. 1WANOVA, Tukey mct, $^{ns}P > 0.9999$, $^{***}P = 0.0009$, $^{ns}P = 0.9631$. (C) Time required for the rats to traverse the balance beam in its entirety during the balance beam. Summary data: The Bay 73-6691 treatment cohort included WT ($n = 15$), Hom(6M$^+$) ($n = 11$), Hom(6M$^-$) ($n = 14$), and Hom(6M$^+$)+Bay(+) ($n = 9$) individuals. Each rat was tested in triplicate. 1WANOVA, Tukey mct, $^{ns}P = 0.9922$, $^{****}P < 0.0001$, $^{ns}P > 0.4606$. (D) Gait analysis illustrating rodent footprints, where red represents the forelimbs and blue represents the hindlimbs. The parameters measured included LFSL (left front stride length), RFSL (right front stride length), LRSL (left rear stride length), RRSL (right rear stride length), LFRO (left front rear overlap), RFRO (right front rear overlap), FBW (front-base width), and RBW (rear-base width). The Bay 73-6691 treatment cohort included WT ($n = 15$), Hom(6M$^+$) ($n = 11$), Hom(6M$^-$) ($n = 14$), and Hom(6M$^+$)+Bay(+) ($n = 9$) individuals. (E) Summary of gait analysis data. Each rat was tested in triplicate. 1WANOVA, Tukey mct, LFSL: $^*P = 0.0185$, $^{****}P < 0.0001$; RFSL: $^{****}P < 0.0001$; LRSL: $^{****}P < 0.0001$; RRSL: $^{****}P < 0.0001$; RBW: $^{***}P = 0.0003$; RFRO: $^*P = 0.0336$. $^{ns}P > 0.05$ for all comparisons. (F) Morris water maze trajectory depicting the exploration patterns of the rats in Bay 73-6691 treatment cohort after the introduction of the platform in the third quadrant. The Bay 73-6691 treatment cohort included WT ($n = 15$), Hom(6M$^+$) ($n = 11$), Hom(6M$^-$) ($n = 11$), and Hom(6M$^+$)+Bay(+) ($n = 9$) individuals. (G) Latency period for rats to locate the hidden platform in the third quadrant, monitored over a consecutive 5-day period. WT ($n = 15$), Hom(6M$^+$) ($n = 11$), Hom(6M$^-$) ($n = 11$), and Hom(6M$^+$)+Bay(+) ($n = 9$). Summary data: 2WANOVA. (H) On the sixth day after platform removal, the proportion of time the rats spent in the NE quadrant after water entry was determined. WT ($n = 15$), Hom(6M$^+$) ($n = 11$), Hom(6M$^-$) ($n = 11$), and Hom(6M$^+$)+Bay(+) ($n = 9$). Summary data: 1WANOVA, Tukey mct, $^{ns}P = 0.9882$, $^{****}P < 0.0001$, $^{***}P = 0.0009$. (I) Number of platform crossings within 60 s and 5 min after water entered the NE quadrant after platform removal. WT ($n = 15$), Hom(6M$^+$) ($n = 11$), Hom(6M$^-$) ($n = 11$), and Hom(6M$^+$)+Bay(+) ($n = 9$). Summary data: 1WANOVA, Tukey mct, $^{ns}P = 0.7787$, $^{***}P = 0.0008$, $^{ns}P = 0.2410$. (J) Tolerance to maximal velocity in the rodent rotarod test in AAV-CHIP injection cohort. Each rat was tested in triplicate. 1WANOVA, Tukey mct, $^{***}P = 0.0001$, $^{****}P < 0.0001$, $^{ns}P = 0.4259$. (K) Time required for the rats to traverse the balance beam in its entirety during the balance beam in AAV-CHIP injection cohort. Each rat was tested in triplicate. 1WANOVA, Tukey mct, $^{***}P = 0.0010$, $^{**}P = 0.0028$, $^{ns}P = 0.8879$. (L) Gait analysis illustrating rodent footprints in AAV-CHIP injection cohort. (M) Summary of gait analysis data. Each rat was tested in triplicate. 1WANOVA, Tukey mct, LFSL: $^{****}P < 0.0001$ for all comparisons; RFSL: $^{****}P < 0.0001$ for all comparisons; LRSL: $^{****}P < 0.0001$ for all comparisons; RRSL: $^{****}P < 0.0001$ for all comparisons; FBW: $^*P = 0.0237$; RBW: $^{***}P = 0.0003$, $^{**}P = 0.0039$. $^{ns}P > 0.05$ for all comparisons. (N) Morris water maze trajectory depicting the exploration patterns of the rats in the AAV-CHIP injection cohort after the introduction of the platform in the third quadrant. Hom(7M$^+$) ($n = 8$), Hom(7M$^+$) + AAV-CHIP ($n = 8$), Hom(7M$^+$) + AAV-NC ($n = 8$), and WT ($n = 8$). (O) Latency period for rats to locate the hidden platform in the third quadrant, monitored over a consecutive 5-day period. Hom(7M$^+$) ($n = 8$), Hom(7M$^+$) + AAV-CHIP ($n = 8$), Hom(7M$^+$) + AAV-NC ($n = 8$), and WT ($n = 8$). Summary data: 2WANOVA. (P) On the sixth day after platform removal, the proportion of time the rats spent in the NE quadrant after water entry was determined. Hom(7M$^+$) ($n = 8$), Hom(7M$^+$) + AAV-CHIP ($n = 8$), Hom(7M$^+$) + AAV-NC ($n = 8$), and WT ($n = 8$). Summary data: 1WANOVA, Tukey mct, $^{***}P = 0.0002$, $^{***}P = 0.0001$, $^{**}P = 0.0094$. (Q) Number of platform crossings within 60 s and 5 min after water entered the NE quadrant after platform removal. Hom(7M$^+$) ($n = 8$), Hom(7M$^+$) + AAV-CHIP ($n = 8$), Hom(7M$^+$) + AAV-NC ($n = 8$), and WT ($n = 8$). Summary data: 1WANOVA, Tukey mct, $^{****}P < 0.0001$, $^{***}P = 0.0001$, $^{ns}P = 0.2134$. Each summary panel shows the means ± SDs, summary plot (B, C, E, H–K, M, P, Q) and regressions (F, O). The $P$-value are shown from left to right. Source data are available online for this figure.

revealed improved balance, coordination and gait in the homozygous mutant rats (Fig. 3J–M) and a significant increase in memory and cognitive abilities after AAV-CHIP treatment (Fig. 3N–Q). These findings underscore the significant ameliorative impact of CHIP on ataxic gait and cognitive memory impairment.

Overall, both PDE9A inhibitor treatment and exogenous CHIP expression therapy have therapeutic efficacy in the treatment of *CHIP* mutation models.

## PDE9A inhibits PKG-mediated CHIP phosphorylation, and Bay 73-6691 elevates phosphorylated CHIP

Recent discoveries have shown that PKG, the principal effector of the second messenger cGMP, directly phosphorylates CHIP at serine 19, a highly conserved residue. CHIP-pS19 increases the stability of CHIP and enhances its affinity for HSP70. Consequently, PKG-mediated CHIP-pS19 (human, S20 mouse) is more proficient in protein clearance (Ranek et al, 2020). Two pivotal questions arise. (1) Can PDE9A influence the endogenous expression of CHIP by modulating its phosphorylation, thereby shortening the CHIP half-life? (2) Can Bay 73-6691 enhance CHIP phosphorylation by suppressing PDE9A levels, further strengthening the binding of CHIP to HSP70 and facilitating the ubiquitination and degradation of PDE9A in tandem with CHIP-HSP70?

We used a CHIP-S20-specific phosphorylation antibody (Antibody-rabbit Pho-CHIP Ser20) (Ranek et al, 2020) for probing p-CHIP decreases in the cerebellum and hippocampus of our rodent model. After the inhibition of PDE9A with Bay 73-6691, a conspicuous increase in CHIP S20 phosphorylation was detected (Fig. 4A). Moreover, AAV-mediated exogenous CHIP expression increased p-CHIP levels (Fig. 4B). In vitro cellular experiments also confirmed these observations, demonstrating a decrease in p-CHIP levels following CHIP inactivation mutations and a notable decrease in p-CHIP levels as PDE9A increased or CHIP decreased (Appendix Fig. S5A). Inhibition of PDE9A elevates cGMP protein levels, which in turn increases PKG protein levels, thereby increasing the phosphorylation of downstream proteins (Van Staveren et al, 2002).

Moreover, we injected the same dose of Bay 73-6691 into wild-type rats (WT+Bay(+)) and found that p-CHIP levels increased in these rats. Interestingly, in both normal HEK293T cells (NC) and those overexpressing CHIP (CHIP), treatment with Bay 73-6691 (NC+Bay(+) and CHIP+Bay(+)) also resulted in an increase in the intracellular p-CHIP level, further confirming that PDE9A reduces the phosphorylation level of the CHIP protein (Appendix Fig. S5B,C). Hence, PDE9A attenuates the PKG content and thus indirectly curtails phosphorylated CHIP, truncating its half-life and accelerating its hydrolysis.

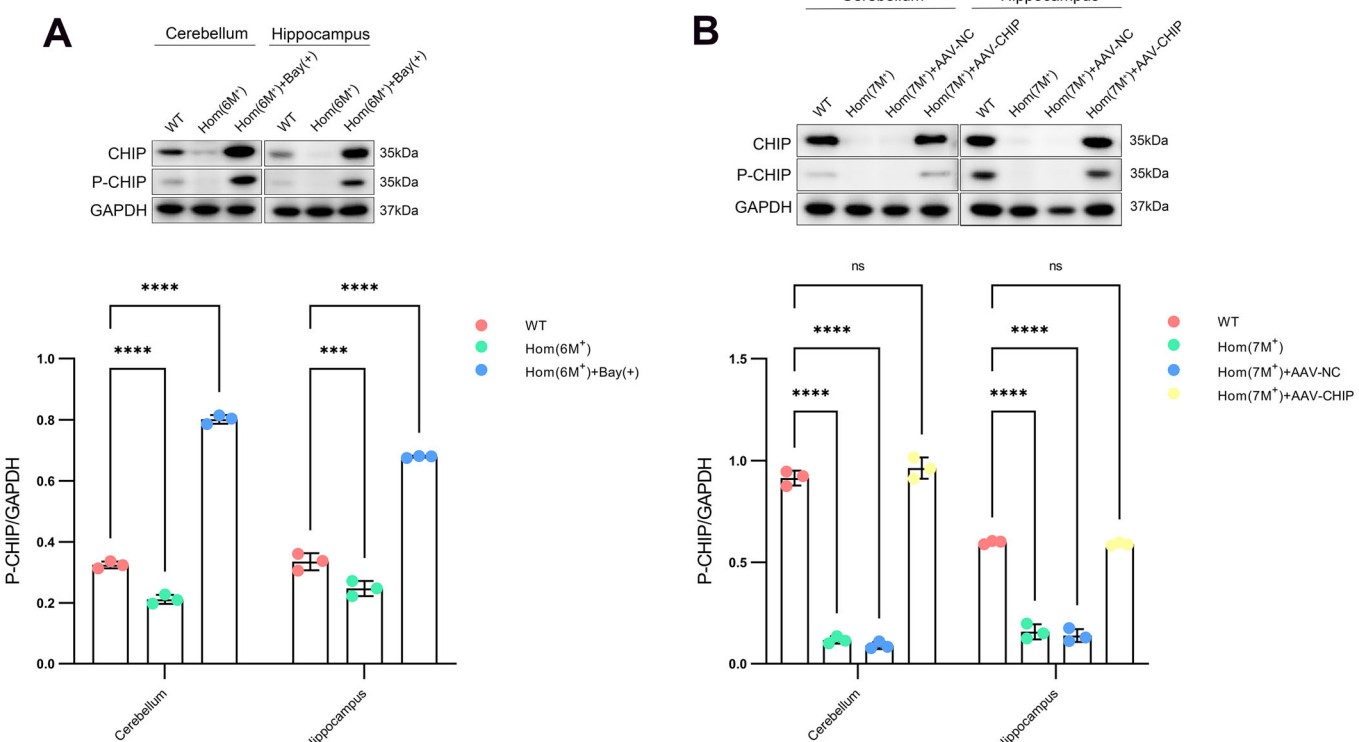

**Figure 4. PDE9A inhibits CHIP phosphorylation.**

(**A**) The phosphorylation state of CHIP at serine 20 in cerebellar and hippocampal tissues across the Bay 73-6691 treatment cohort was assessed via a phospho-specific S20 antibody. Upper: Blots showing the CHIP and p-CHIP (phosphorylated CHIP) levels in Hom(6M+) (6.25-month-old homozygous rats harbouring the p.T246M mutation), Hom(6M+)+Bay(+) (Bay 73-6691-treated homozygous rats), and WT (wild-type littermates), with each lane containing a mixed-tissue protein sample from three rats. Lower: summary data, $n = 3$ biological replicates/group, 2WANOVA, Tukey mct, Cerebellum: ****$P < 0.0001$ for all comparisons; Hippocampus: ****$P < 0.0001$, ***$P = 0.0002$. (**B**) CHIP phosphorylation levels in the cerebellum and hippocampus were quantified for the AAV-CHIP injection cohorts. Upper: Blots showing the CHIP and p-CHIP levels among Hom(7M+) (7-month-old homozygous rats harbouring the p.T246M mutation), Hom(7M+) + AAV-CHIP (AAV-CHIP-injected homozygous rats), Hom(7M+) + AAV-NC (AAV-NC-injected homozygous rats), and WT (wild-type littermates), with each lane containing a mixed tissue protein sample from three rats. Lower: summary data, $n = 3$ biological replicates/group, 2WANOVA, Tukey mct, Cerebellum: ****$P < 0.0001$ for all comparisons, ns$P = 0.2375$; Hippocampus: ****$P < 0.0001$ for all comparisons, ns$P = 0.9858$. Each summary panel shows the means ± SDs and summary plot (**A**, **B**). The $P$-value are shown from left to right. Source data are available online for this figure.

## Treatment with Bay 73-6691 improves mitophagy dysfunction and cell apoptosis in *CHIP* mutations

PDE9A reduces the levels of intracellular CHIP by decreases in CHIP phosphorylation through the cGMP-PKG signalling cascade. Depletion of CHIP decreases the degree of K63- and K27-linked polyubiquitination of PDE9A, inhibiting its lysosomal degradation and thus promoting intracellular PDE9A accumulation. Previously, our investigations revealed significant neuropathological injuries in the cerebellar Purkinje cells of *CHIP* p.T246M mutant rats, characterized by nuclear condensation, dendritic swelling and a notable reduction in Purkinje cell count (Shi et al, 2018b). We hypothesized that increased PDE9A accumulation coupled with decreased CHIP promotes cellular damage and apoptosis.

Annexin V/PI flow cytometry in HEK293T cells revealed that CHIP-T246M (T246M) expression caused 41% of the cells to undergo programmed cell death, in contrast to the 8% of the cells expressing CHIP-WT (CHIP). Cotreatment with Bay 73-6691 (T246M+Bay) reduced T246M-dependent cell death to 28%. Similarly, an increase in PDE9A led to 48% cell death, which was

mitigated by either increased CHIP-WT expression (21%) or Bay 73-6691 treatment (31%) (Appendix Fig. S6A). Similarly, orthogonal CCK8 assays revealed that elevated PDE9A or CHIP-T246M levels significantly hindered cell proliferation. This adverse effect was alleviated by Bay-66791 treatment or increased CHIP-WT expression (Appendix Fig. S6B). Hence, PDE9A accumulation and CHIP mutations lead to decreased CHIP levels, detrimentally affecting cellular functions and hindering normal proliferation. Moreover, compared with age-matched wild-type control (WT) rats, symptomatic homozygous mutation (Hom) rats presented Purkinje cell loss ranging from 45 to 38% in cerebellar sections. Notably, this loss was reduced to 18% in homozygous mutation rats treated with Bay 73-66791 (Hom+Bay(+)) (Fig. 5A). Furthermore, the administration of AAV-CHIP to homozygous mutant rats (Hom+AAV-CHIP) resulted in the complete restoration of Purkinje cell density to levels comparable to those of wild-type rats (Fig. 5B). These data demonstrate that CHIP-mediated mutation results in significant Purkinje cell loss, a hallmark of cerebellar dysfunction. The therapeutic interventions with Bay 73-66791 and AAV-CHIP not only attenuated deficits in motor

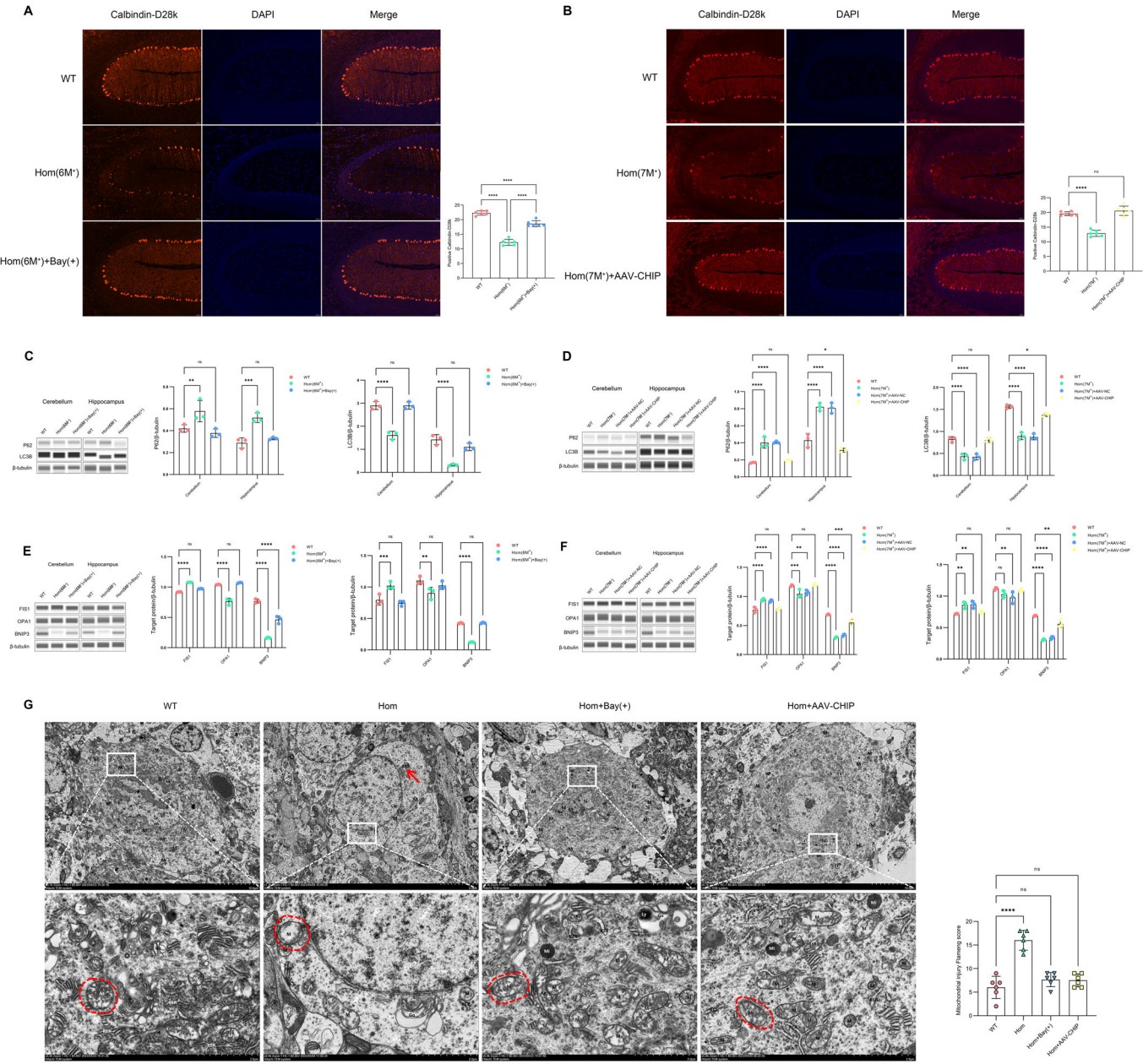

function but also restored Purkinje cell density, highlighting the potential for reversing mutation-induced damage. Additional Nissl staining of the hippocampus of *CHIP* homozygous mutant rats revealed a reduction in the number of neurons and disordered arrangement, which were somewhat improved upon treatment with Bay 73-66791 and AAV-mediated exogenous CHIP expression (Appendix Fig. S6C).

The reduction in CHIP levels and the concurrent increase in PDE9A levels elicit cellular toxicity, culminating in neuronal necrosis and a decrease in the neuronal population. This phenomenon can be attributed to the inactivation of CHIP, which impedes the formation of K63 and K27 ubiquitin chains on PDE9A, thus diminishing the autophagic–lysosomal degradation of PDE9A and fostering its abnormal accumulation within the cell. It has been reported that the concatenation of K63 and K27 ubiquitin chains primarily initiates a signalling cascade that activates mitophagy, targeting aberrantly aggregated substrate proteins for degradation. The equilibrium of mitophagy is pivotal in maintaining the homeostasis of intracellular proteins and mitochondria (Pickles et al, 2018; Tsakiri et al, 2019). Dysregulation of mitophagy results in the accumulation of cytotoxic proteins. Notably, PDE9A can localize to mitochondria, where it regulates mitophagy via the second messenger cGMP. cGMP promotes autophagosome formation and the initiation of mitophagy through pathways involving PKG signalling, oxidative stress responses, and BNIP3. However, overaccumulation of PDE9A decreases cGMP levels, inhibiting mitochondrial respiration and leading to mitochondrial swelling and impaired mitophagy function (Mishra et al, 2021). In addition,

**Figure 5.  PDE9A aggregation and CHIP reduction disrupt mitophagy.**

(A, B) Left: Quantitative immunofluorescence of the cerebellar Purkinje cell marker protein calbindin-D28K to determine alterations in the number of cerebellar Purkinje neurons in the Bay 73-6691-treated and AAV-CHIP-injected cohorts. The Bay 73-6691 treatment cohort included Hom(6M$^+$) (6.25-month-old homozygous rats harbouring the p.T246M mutation), Hom(6M$^+$)+Bay($+$) (Bay 73-6691 solubilized in corn oil administered intraperitoneally at 2 mg/kg to homozygous rats), and WT (wild-type littermates). The AAV-CHIP injection cohort included Hom(7M$^+$) (homozygous rats with the p.T246M mutation), Hom(7M$^+$) + AAV-CHIP (tail vein injection of 1.2 × 10$^{11}$ vg/mL HBAAV2/BBB-CHIP virus to increase CHIP protein levels in the brains of homozygous rats) and WT (wild-type littermates). Calbindin-D28k (red), DAPI (blue). Right: summary data, $n = 6$ biological replicates/group, 1WANOVA, Tukey mct, Bay 73-6691 treatment cohort: ****$P < 0.0001$ for all comparisons; AAV-CHIP injection cohort: ****$P < 0.0001$, $^{ns}P = 0.3712$. (C) Left: Automated capillary western blot assays for the expression of autophagic proteins in cerebellar and hippocampal tissues across the Bay 73-6691 treatment cohort. Tissue protein samples from three rats were mixed for each lane. Right: summary data, $n = 3$ biological replicates/group, 2WANOVA, Tukey mct, P62: **$P = 0.0076$, $^{ns}P = 0.5963$, ***$P = 0.0004$, $^{ns}P = 0.6285$; LC3B: ****$P < 0.0001$, $^{ns}P = 0.9998$, ****$P < 0.0001$, $^{ns}P = 0.0766$. (D) Left: Similar methodology in AAV-CHIP injection cohort. Tissue protein samples from three rats were mixed for each lane. Right: summary data, $n = 3$ biological replicates/group, 2WANOVA, Tukey mct, P62: ****$P < 0.0001$ for all comparisons, $^{ns}P = 0.8687$, *$P = 0.0315$; LC3B: ****$P < 0.0001$ for all comparisons, $^{ns}P = 0.9886$, *$P = 0.0106$. (E) Left: Automated capillary western blot assays for the expression of autophagic proteins in cerebellar and hippocampal tissues from the Bay 73-6691 treatment cohort. Tissue protein samples from three rats were mixed for each lane. Right: summary data, cerebellar tissue followed by the hippocampus. $n = 3$ biological replicates/group, 2WANOVA, Tukey mct, Cerebellum: FIS1: ****$P < 0.0001$, $^{ns}P = 0.1018$; OPA1: ****$P < 0.0001$, $^{ns}P = 0.4467$; BNIP3: ****$P < 0.0001$ for all comparisons. Hippocampus: FIS1: ***$P = 0.0003$, $^{ns}P = 0.5225$; OPA1: **$P = 0.0012$, $^{ns}P = 0.2388$; BNIP3: ****$P < 0.0001$. $^{ns}P = 0.9928$. (F) Left: Similar methodology in AAV-CHIP injection cohort. Right: summary data, cerebellar tissue followed by the hippocampus. $n = 3$ biological replicates/group, 2WANOVA, Tukey mct, Cerebellum: FIS1: ****$P < 0.0001$ for all comparisons, $^{ns}P = 0.9207$; OPA1: ***$P = 0.0003$, **$P = 0.0022$, $^{ns}P = 0.9778$; BNIP3: ****$P < 0.0001$ for all comparisons, ***$P = 0.0003$. Hippocampus: FIS1: **$P = 0.0026$, **$P = 0.0016$, $^{ns}P = 0.8312$; OPA1: $^{ns}P = 0.0774$, **$P = 0.0027$, $^{ns}P = 0.8654$; BNIP3: ****$P < 0.0001$ for all comparisons, **$P = 0.0054$. (G) TEM observations of mitochondrial alterations in rat cerebellar tissue. Left: Ultrastructure of the cerebellum under TEM, Hom (*CHIP* p.T246M homozygous mutant rats), Hom+Bay($+$) (Bay 73-6691 treated the homozygous mutant rats), Hom+AAV-CHIP (AAV-CHIP injected the homozygous mutant rats), and WT (wild-type littermates). In the Hom group, mitochondria (M, indicated by the red dashed circle) are less common and exhibit localized membrane blurriness and rupture, abundant crista disintegration and dissolution and matrix vacuolisation. Autophagolysosomes (ASSs, red arrows) are also visible. N: Nucleus, Go: Golgi apparatus, Lip: lipofuscin, RER: rough endoplasmic reticulum, Mi: microbody, AP: autophagosomes. Right: Mitochondrial injury Flameng score; five different microscopic fields were selected for each sample, and 20 mitochondria were randomly selected for each field of view. Summary data: $n = 6$ rat samples/group, 1WANOVA, Tukey mct, ****$P < 0.0001$, $^{ns}P = 0.4370$, $^{ns}P = 0.5253$. Each summary panel shows the means ± SDs and summary plot (A–G). The $P$-value are shown from left to right. Source data are available online for this figure.

CHIP plays a crucial role within mitochondria, regulating mitophagy (Lizama et al, 2018). CHIP maintains the normal quantity of mitochondria within the cell by activating LC3B-mediated autophagy. A decrease in CHIP levels triggers mitochondrial oxidative stress, resulting in an increased number of aberrant mitochondria. Similarly, ubiquitin chains linked through K27, which are mediated by CHIP, are recognized by the autophagy receptor P62, triggering mitophagy and maintaining the balance of mitophagy within the cell (Deretic and Levine, 2018; Lizama et al, 2018; Sparrer et al, 2017). Thus, we investigated whether the mutation of CHIP and the increase in PDE9A levels could lead to mitochondrial abnormalities and disrupt the equilibrium of mitophagy. Coincidentally, in *CHIP* homozygous mutation rats (Hom), increased P62 and decreased LC3B levels were detected in both the hippocampus and cerebellum. Nonetheless, Bay 73-6691-treated (Hom+Bay($+$)) and AAV-mediated exogenous CHIP expression (Hom+AAV-CHIP) resulted in a decrease in P62 and a surge in LC3B levels (Fig. 5C,D). Concurrently, a decrease in the mitochondrial inner membrane fusion protein OPA1 and the mitochondrial autophagy protein BNIP3, which are located on the outer membrane, alongside an increase in the mitochondrial fission protein FIS1, was noted in the *CHIP* homozygous mutant rats. These changes imply a disruption in mitochondrial fusion function and an imbalance in mitochondrial autophagy, attributed to CHIP-mediated cerebellar ataxia. However, normalization of these proteins was observed following treatment with Bay 73-6691 or AAV-CHIP, demonstrating that enhancing CHIP levels and reducing PDE9A levels can maintain mitochondrial autophagy equilibrium (Fig. 5E,F). In addition, TEM analysis revealed a reduction in normal mitochondria, an increase in abnormally swollen mitochondria with compromised membrane integrity, extensive cristae disruption, matrix dissolution or vacuolisation, and an increased number of autolysosomes and autophagosomes in the cerebellum of *CHIP* homozygous mutant rats (Hom).

Conversely, treatment with Bay 73-6691 (Hom+Bay($+$)) and AAV-CHIP (Hom+AAV-CHIP) increased the number of mitochondria, size variability, and membrane integrity in the cerebellar tissues of mutant rats, resulting in healthier mitochondria (Fig. 5G). Mitochondrial assessment via the Flameng method revealed significantly higher scores in the Hom group than in the other three groups, with decreases in the mitochondrial perimeter and area (Flameng et al, 1980; Shaw et al, 2020). Treatment with Bay 73-6691 or AAV-CHIP resulted in a reduction in scores, indicating mitigated mitochondrial damage and morphological improvements in the mutated rats (Fig. 5G; Appendix Fig. S6D, Appendix Table S2). These findings suggest that reduced CHIP and loss of function, accompanied by increased PDE9A accumulation, damage mitochondrial structure and function, inducing mitophagy impairment. While Bay 73-66791 inhibits PDE9A aggregation and increases CHIP levels, it ameliorates mitochondrial damage and maintains the mitophagy balance.

## Single-cell sequencing reveals cGMP and cAMP signalling pathway dysregulation as the predominant pathway alterations in CHIP mutations

Research suggests that *CHIP* mutations manifest predominantly as ARCA and cognitive impairments, highlighting the close association between the cerebellum and the hippocampus (Shi et al, 2018b). Single-cell RNA sequencing (scRNA-seq) technology offers insights into the pathophysiological changes underlying this condition at the single-cell level. Therefore, to elucidate the regulatory relationship between PDE9A and CHIP at the single-cell level, this study conducted scRNA-seq of the hippocampus and cerebellum from a rat model.

We obtained four sets of scRNA-seq results, encompassing a cohort of wild-type rats as the control group (WT), *CHIP* homozygous mutation group (Hom), AAV-mediated exogenous

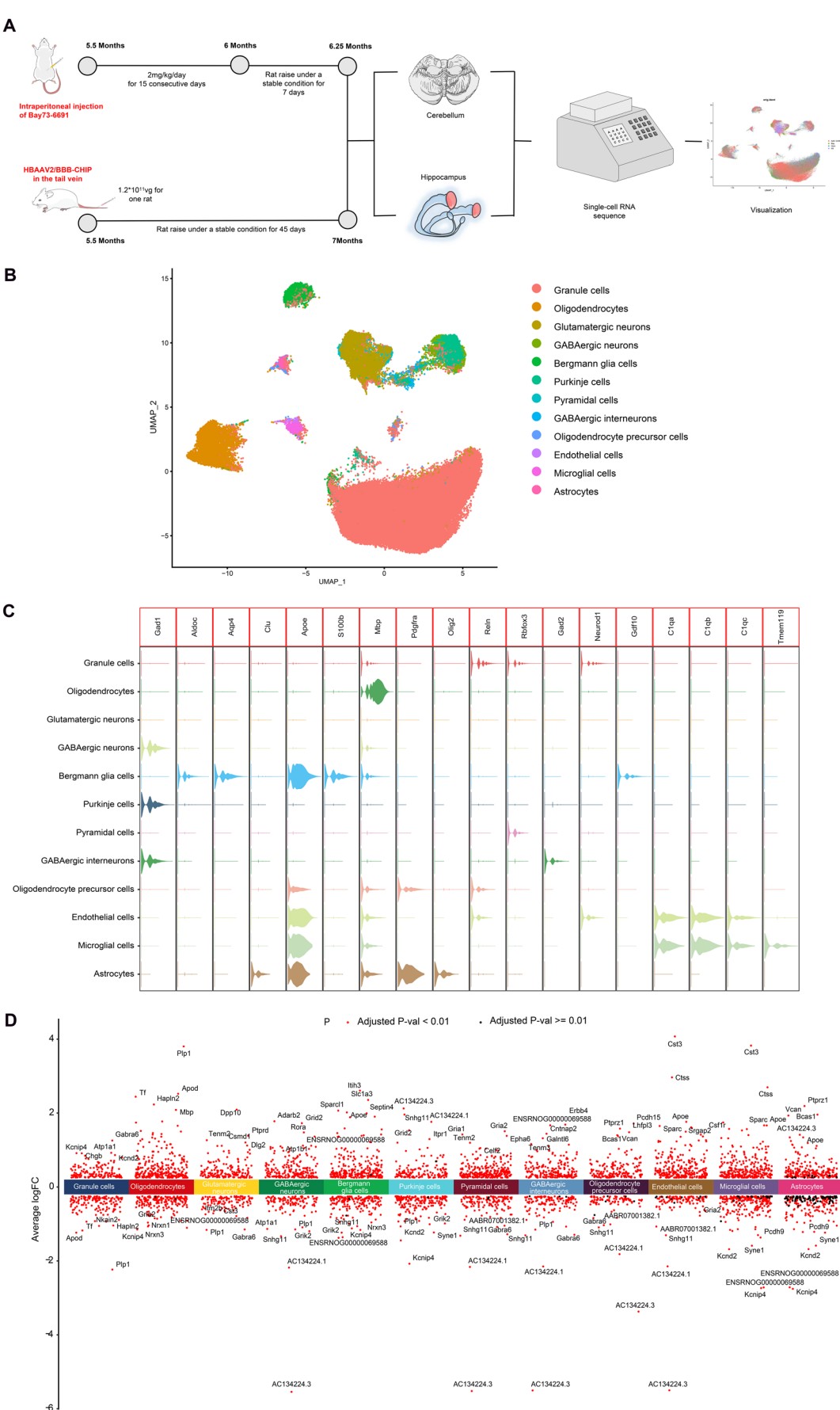

◀ **Figure 6.  Single-cell transcriptomic cell clustering.**

(A) Schematic of the experimental strategy applied for single-cell transcriptome analysis. Four experimental groups were assessed: Hom ($n = 3$, *CHIP* p.T246M homozygous mutant rats), Hom+Bay(+) ($n = 3$, Bay 73-6691-treated homozygous mutant rats), Hom+AAV-CHIP ($n = 3$, AAV-CHIP-injected homozygous mutant rats), and WT ($n = 3$, wild-type littermates). Hippocampal and cerebellum tissues were mixed 1:1 for each rat, with three rats per group. (B, C) Cell clustering diagram: Cells from the rat hippocampus and cerebellum were classified into 12 distinct clusters on the basis of the following marker genes: granule cells (Reln, Rbfox3, Neurod1), oligodendrocytes (Mbp), glutamatergic neurons (Aldoc), GABAergic neurons (Gad2, Pnoc, Tcf7l2, Lhx9), GABAergic interneurons (Pax2, Lbx1), pyramidal cells (Nell1, Npy2r, Slc4a8, B3gat2), oligodendrocyte precursor cells (Pdgfra, Olig1, Olig2), endothelial cells (C1qa, C1qb, C1qc), Bergmann glia cells (Gdf10, S100b), microglial cells (Tmem119), Purkinje cells (Gad1, Car8, Calb1, Pcp2, Necab2, Pcp4) and astrocytes (Clu, Apoe). (D) Volcano plot illustrating the expression of cluster-specific marker genes across the 12 identified cell clusters. granule cells (58,325 cells), oligodendrocytes (6395 cells), glutamatergic neurons (4576 cells), GABAergic neurons (2982 cells), GABAergic interneurons (840 cells), pyramidal cells (1765 cells), oligodendrocyte precursor cells (1107 cells), endothelial cells (613 cells), Bergmann glia cells (2498 cells), microglial cells (383 cells), Purkinje cells (1492 cells) and astrocytes (183 cells). Wilcoxon rank sum test.

CHIP expression group (Hom+AAV-CHIP), and Bay 73-6691 treatment group (Hom+Bay(+)), capturing 13,473, 19,279, 25,238, and 23,177 individual cells, respectively (Fig. 6A; Appendix Fig. S7A). On the basis of the expression of marker genes (Carter et al, 2018; Gupta et al, 2018; Shah et al, 2016), we classified 81,167 single cells into 12 distinct cell subtypes, including granule cells, oligodendrocytes, glutamatergic neurons, GABAergic neurons, GABAergic interneurons, pyramidal cells, oligodendrocyte precursor cells, endothelial cells, Bergmann glia cells, microglial cells, Purkinje cells and astrocytes, and the heatmaps revealed the expression of specific genes within each cell subtype (Fig. 6B–D; Appendix Fig. S7B).

Differential gene enrichment analysis, using KEGG pathway analysis, across total cells revealed that the Hom group exhibited significant enrichment within the cAMP and cGMP-PKG signalling pathways relative to the Wt group (Fig. 7A). AUCell and GSEA further revealed that, compared with the Wt group, the Hom group displayed notable suppression of the cGMP-PKG signalling pathway and significant activation of the cAMP signalling pathway (Fig. 7A–E).

Previous research has indicated that CHIP exerts regulatory control over the cAMP signalling pathway. cAMP and PRKACA are critical components of the cAMP signalling pathway. cAMP stimulates the protein kinase A (PKA) holoenzyme, which consists of two catalytic subunits (PKAc) and two regulatory subunits (PKAr). The PKA holoenzyme is inherently inactive and becomes active when stimulated by the second messenger cAMP, leading to a conformational change in the regulatory subunit that releases the active subunit PKAc. The catalytic subunit alpha of protein kinase A (PRKACA) is the predominant isoform of PKAc, is expressed in most tissues, and is primarily used to represent the activity status of PKAc within tissues (Yang et al, 2014). Prior studies have reported that CHIP can mediate the ubiquitination and degradation of PKAc (Rinaldi et al, 2019). To validate the scRNA-seq results, we observed an increase in cAMP and PRKACA levels in the cerebellum and hippocampus of *CHIP* mutant rats (Fig. 8A,B). Furthermore, we introduced forskolin, which elevates cAMP levels by activating adenylate cyclase, into an in vivo ubiquitination reaction involving PDE9A and CHIP. The results revealed that an increase in cAMP concentration was associated with a reduction in the CHIP-mediated K63- and K27-linked ubiquitination of PDE9A. In addition, both PKAc and PDE9A interact with HSP70 to target CHIP for ubiquitination. These findings suggest that the cAMP-induced upregulation of PKAc is associated with competitive binding to CHIP for the ubiquitination and inhibition of PDE9A degradation (Fig. 8C; Appendix Fig. S3J). cGMP, PKG1 and PKG2

are the key molecules in the cGMP-PKG signalling pathway. To validate the accuracy of the aforementioned conclusions, we further assessed the levels of these key molecules in the cerebellum and hippocampus. Compared with those in the Wt group, the levels of CHIP, cGMP, PKG1 and PKG2 were diminished in the Hom group, and the PDE9A level was increased (Fig. 8D–G). These results effectively demonstrate that CHIP mutation results in cGMP and cAMP signalling imbalance.

## AAV-CHIP and Bay 73-6691 participate in the bidirectional regulation of cGMP and cAMP signalling for therapeutic effects

To determine whether AAV-mediated exogenous CHIP expression (Hom+AAV-CHIP) rescues the pathological changes induced by CHIP mutation, we employed scRNA-seq and revealed that treatment with AAV-CHIP eliminates the abnormal signalling pathways caused by CHIP mutation. Compared with the Hom +AAV-CHIP group, the *CHIP* homozygous mutation group (Hom) presented significantly lower activity of the cGMP-PKG signalling pathway, concomitant with pronounced activation of the cAMP signalling pathway (Fig. 7F–H; Appendix Fig. S8). Considering the above roles of PDE9A in *CHIP* mutations and the inhibitory effect of Bay 73-6691 on PDE9A, we hypothesized that Bay 73-6691 has therapeutic potential for ACAR in the context of *CHIP* mutations. Therefore, we performed scRNA-seq on *CHIP* mutants treated with Bay 73-6691 (Hom+Bay(+)) to assess their therapeutic effects. These results were in line with expectations, as Bay 73-6691 had a pronounced therapeutic effect on *CHIP* mutations (Fig. 7I–K; Appendix Fig. S8). The levels of these pathway signalling molecules in the cerebellum and hippocampus further confirmed the therapeutic effects of AAV-CHIP treatment and Bay 73-6691 treatment on the imbalance of cGMP and cAMP signalling in the CHIP mutation group. To validate these conclusions, we further assessed the levels of key molecules of signalling pathways in the cerebellum and hippocampus after exogenous elevation of CHIP levels and Bay 73-6691 treatment in a CHIP mutation animal model. cAMP and PRKACA considerably increased in the Hom group, but this increase was reversed with the upregulation of CHIP levels and the downregulation of PDE9A levels. CHIP, cGMP, PKG1 and PKG2 levels were decreased in the Hom group, accompanied by an increase in PDE9A; however, the exogenous elevation of CHIP levels and Bay 73-6691 treatment reversed these protein levels (Fig. 8A,B,D–G).

Purkinje neurons are the primary neuronal subtype affected by *CHIP* mutations, and we further explored changes in Purkinje

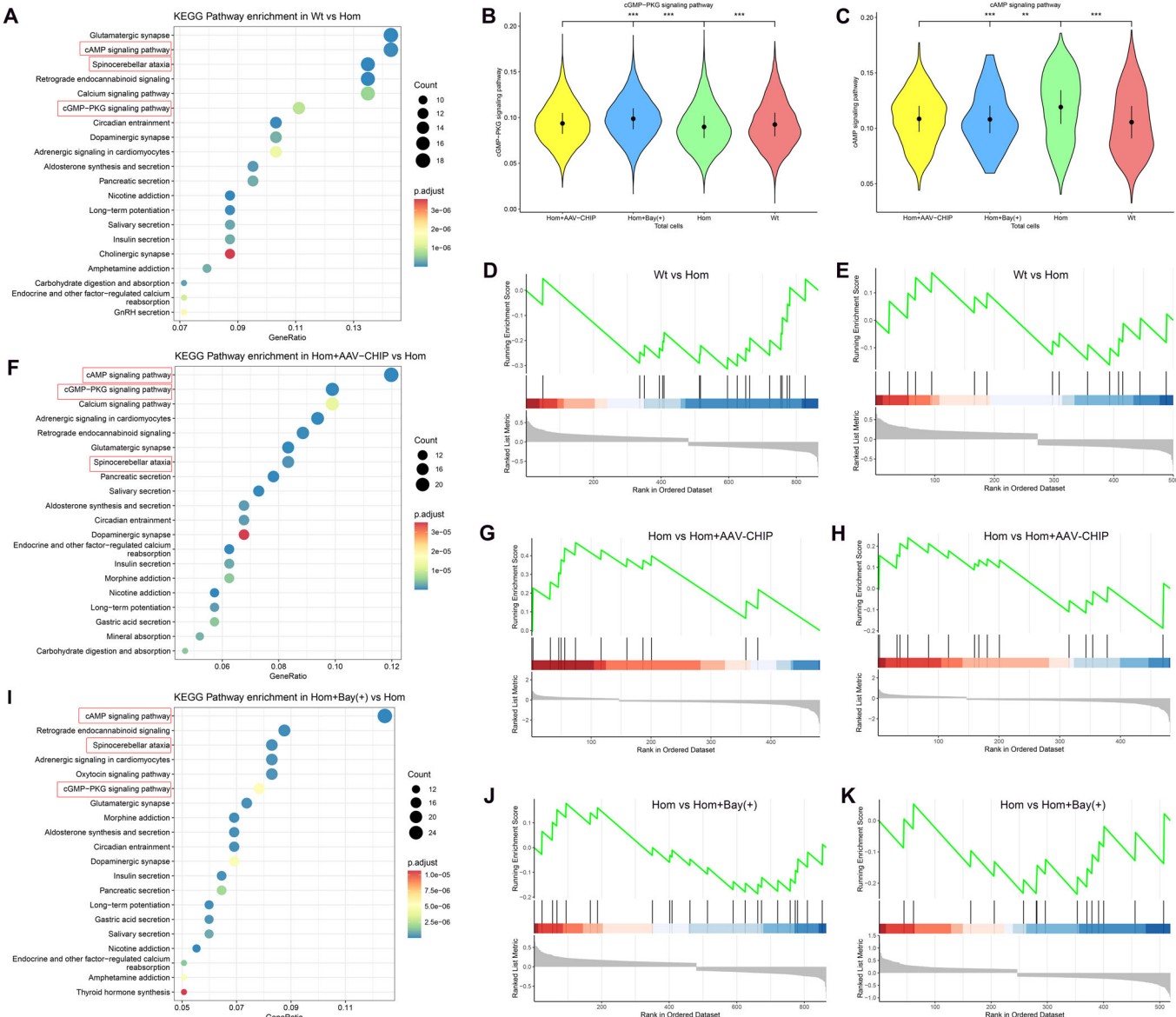

**Figure 7. Involvement of PDE9A in CHIP-mediated cGMP/cAMP signalling crosstalk.**

(A) KEGG enrichment analysis of differentially expressed genes in whole cells comparing Wt vs. Hom. Hom (*CHIP* p.T246M homozygous mutant rats) and Wt (wild-type littermates). The most relevant signalling pathways are highlighted in red boxes. Hypergeometric test. (B) AUCell pathway activity analysis showing the expression of the cGMP–PKG signalling pathway in whole cells across the four groups. Hom (n = 3, *CHIP* p.T246M homozygous mutant rats. 19,279 cells), Wt (n = 3, wild-type littermates. 13,473 cells), Hom+Bay(+) (n = 3, Bay 73-6691-treated homozygous mutant rats. 23,177 cells), and Hom+AAV-CHIP (n = 3, AAV–CHIP-injected homozygous mutant rats. 25,238 cells). T test, ***P < 2.22E−16, ***P < 2.22E−16, ***P < 2.22E−16. (C) AUCell pathway activity analysis showing the expression of the cAMP signalling pathway in whole cells across the four groups. Hom (n = 3. 19,279 cells), Wt (n = 3. 13,473 cells), Hom+Bay(+) (n = 3. 23,177 cells), and Hom+AAV-CHIP (n = 3. 25,238 cells). T test, ***P < 2.22E−16, **P = 0.0018, ***P < 2.22E−16. (D, E) GSEA pathway activity analysis indicating alterations in the cGMP-PKG signalling pathway and cAMP signalling pathway in total cells when comparing Wt vs. Hom. (F) KEGG enrichment analysis of differentially expressed genes in whole cells comparing Hom+AAV-CHIP vs. Hom. Hypergeometric test. (G, H) GSEA pathway activity analysis indicating alterations in the cGMP-PKG signalling pathway and cAMP signalling pathway in total cells when comparing Hom+AAV-CHIP vs. Hom. (I) KEGG enrichment analysis of differentially expressed genes in whole cells comparing Hom+Bay(+) vs. Hom. Hypergeometric test. (J, K) GSEA pathway activity analysis indicating alterations in the cGMP-PKG signalling pathway and cAMP signalling pathway in total cells when comparing Hom+Bay(+) vs. Hom. Each summary panel shows the means ± SDs and summary plot (B, C). The *P*-value are shown from left to right.

neurons through scRNA-seq. Through cell proportion analysis, we found that the proportion of Purkinje neurons in the Hom group was significantly lower than that in the WT group. Interestingly, treatment with Bay 73-6691 and AAV-CHIP resulted in an increase in the number of Purkinje neurons in the Hom group, suggesting

that Bay 73-6691 and AAV-CHIP treatments could partially restore the number of Purkinje neurons, thereby ameliorating the associated symptoms in *CHIP*-mutant rats (Wt: 0.87%, Hom: 0.24%, Hom+AAV-CHIP: 4.54%, Hom+Bay(+): 0.78%). Similar to the total cell analysis, the KEGG enrichment analysis of Purkinje

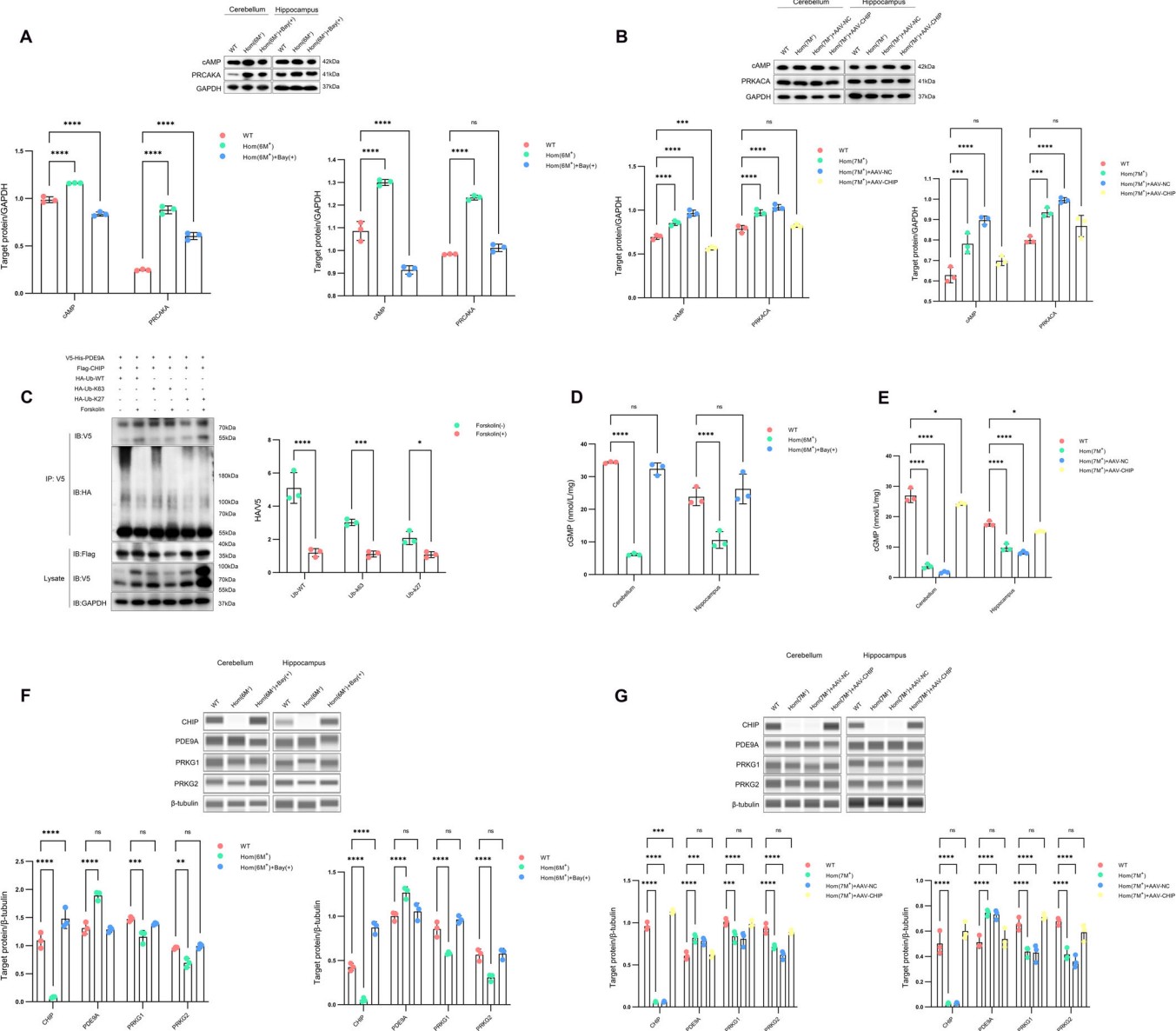

neurons revealed that the genes differentially expressed in the Hom group were enriched in ubiquitin-mediated proteolysis, the cAMP signalling pathway, spinocerebellar ataxia and the cGMP-PKG signalling pathway (Fig. 9A). The analysis of Purkinje cell pathway activity supported these observations. Overall, these results reiterate the ability of CHIP to activate cGMP/PKG signalling while concurrently inhibiting cAMP signalling, thus maintaining intra-cellular cGMP/cAMP homeostasis (Fig. 9A,B,D,E). Notably, *CHIP* mutations hinder the formation of K63 and K27 polyubiquitin chains on PDE9A, thereby affecting its degradation in autolyso-somes. Moreover, elevated levels of PDE9A result in a reduction in PKG levels. This decrease in PKG leads to a concomitant reduction in p-CHIP and a decrease in the half-life of CHIP. Therefore, we also performed scRNA-seq on the *CHIP* mutants treated with Bay 73-6691 to assess the aforementioned therapeutic effect on Purkinje neurons. Compared with those in the Hom group, the genes

differentially expressed in the Bay 73-6691-treated group were predominantly enriched in oxidative phosphorylation, the cAMP signalling pathway and ubiquitin-mediated proteolysis (Fig. 9C), supporting the involvement of PDE9A in CHIP-mediated ubiqui-tination reactions, with CHIP potentially participating in PDE9A-mediated phosphorylations.

Importantly, AUCell analysis revealed that CHIP E3 ubiquitin ligase activity deficiency affected ubiquitin-mediated proteolysis, ultimately increasing the level of intracellular cAMP. The activation of ubiquitin-mediated proteolysis resulted in PDE9A degradation, thus promoting autophagy and mitophagy (Fig. 9F–H). These results collectively support the notion that CHIP promotes the degradation of PDE9A within autolysosomes through K63- and K27-linked ubiquitination, whereas cAMP competitively inhibits the ubiquitination of PDE9A. In addition, following Bay 73-6691 treatment, the cGMP–PKG signalling pathway and oxidative

**Figure 8.  Expression levels of pathway proteins in the cerebellum and hippocampus.**

(**A**) Upper: Blotting was used to evaluate cAMP–PKA signalling protein expression in the cerebellum and hippocampus tissues of Bay 73-6691 treatment cohorts. Bay 73-6691 treatment cohorts: Hom(6M$^+$) (6.25-month-old homozygous rats harbouring the p.T246M mutation), Hom(6M$^+$)+Bay(+) (Bay 73-6691 solubilized in corn oil administered intraperitoneally at 2 mg/kg to homozygous rats), and WT (wild-type littermates). The protein samples from three rats were mixed for each lane. The GAPDH blots selected for the repeated experiments were consistent, and statistical analysis was performed by calculating the ratio of each target protein to its corresponding GAPDH. Lower: summary data, $n = 3$ biological replicates/group, 2WANOVA, Tukey mct, Cerebellum: cAMP: ****$P < 0.0001$ for all comparisons, PRCAKA: ****$P < 0.0001$ for all comparisons; Hippocampus: cAMP: ****$P < 0.0001$ for all comparisons, PRCAKA: ****$P < 0.0001$, $^{ns}P = 0.2607$. (**B**) Upper: Blotting was used to evaluate cAMP–PKA signalling protein expression in the cerebellum and hippocampus tissues of AAV-CHIP injection cohort. The AAV-CHIP injection cohort included Hom(7M$^+$) (homozygous rats with the p.T246M mutation), Hom(7M$^+$) + AAV-CHIP (tail vein injection of 1.2 × 10$^{11}$ vg/mL HBAAV2/BBB-CHIP virus to increase CHIP protein levels in the brains of homozygous rats) and WT (wild-type littermates). The protein samples from three rats were mixed for each lane. The GAPDH blots selected for the repeated experiments were consistent, and statistical analysis was performed by calculating the ratio of each target protein to its corresponding GAPDH. Lower: summary data, $n = 3$ biological replicates/group, 2WANOVA, Tukey mct, Cerebellum: cAMP: ****$P < 0.0001$ for all comparisons, ***$P = 0.0004$, PRCAKA: ****$P < 0.0001$ for all comparisons, $^{ns}P = 0.4916$; Hippocampus: cAMP: ***$P = 0.0002$, ****$P < 0.0001$, $^{ns}P = 0.0783$, PRCAKA: ***$P = 0.0006$, ****$P < 0.0001$, $^{ns}P = 0.0867$. (**C**) The impact of the cAMP agonist forskolin (50 μM) on CHIP-mediated PDE9A ubiquitination with K63 and K27 linkages. In vivo ubiquitination assays revealed changes in CHIP-mediated ubiquitination of PDE9A upon cAMP suppression. Experiment replicated × 3. Antibodies: V5-rabbit, HA-mouse, and Flag-mouse. Paired t test, ****$P < 0.0001$, ***$P = 0.0006$, *$P = 0.0496$. (**D**) ELISA assays were used to measure cGMP concentrations in the cerebellum and hippocampus in the Bay 73-6691 treatment cohort. Tissue protein samples from three rats from each group were mixed. Summary data, $n = 3$ biological replicates/group, 2WANOVA, Tukey mct, Cerebellum: ****$P < 0.0001$, $^{ns}P = 0.6151$; Hippocampus: ****$P < 0.0001$, $^{ns}P = 0.5007$. (**E**) ELISA assays were used to measure cGMP concentrations in the cerebellum and hippocampus in the AAV-CHIP injection cohort. Tissue protein samples from three rats from each group were mixed. Summary data, $n = 3$ biological replicates/group, 2WANOVA, Tukey mct, Cerebellum: ****$P < 0.0001$ for all comparisons, *$P = 0.0233$; Hippocampus: ****$P < 0.0001$ for all comparisons, *$P = 0.0314$. (**F**) Upper: Automated capillary western blot assays were used to assess PKG signalling protein expression in cerebellar and hippocampal tissues across the Bay 73-6691 treatment cohort. Tissue protein samples from three rats were mixed for each lane. Lower: summary data, $n = 3$ biological replicates/group, 2WANOVA, Tukey mct, Cerebellum: CHIP: ****$P < 0.0001$ for all comparisons, PDE9A: ****$P < 0.0001$, $^{ns}P = 0.9301$, PRKG1: ***$P = 0.0005$, $^{ns}P = 0.4655$, PRKG2: **$P = 0.0030$, $^{ns}P = 0.8901$; Hippocampus: CHIP: ****$P < 0.0001$ for all comparisons, PDE9A: ****$P < 0.0001$, $^{ns}P = 0.5242$, PRKG1: ****$P < 0.0001$, $^{ns}P = 0.0856$, PRKG2: ****$P < 0.0001$, $^{ns}P = 0.9677$. (**G**) Upper: Automated capillary western blot assays were used to assess PKG signalling protein expression in cerebellar and hippocampal tissues across the AAV-CHIP injection cohort. Tissue protein samples from three rats were mixed for each lane. Lower: summary data, $n = 3$ biological replicates/group, 2WANOVA, Tukey mct, Cerebellum: CHIP: ****$P < 0.0001$ for all comparisons, ***$P = 0.0003$, PDE9A: ****$P < 0.0001$, ***$P = 0.0002$, $^{ns}P = 0.9801$, PRKG1: ***$P = 0.0003$, ****$P < 0.0001$, $^{ns}P = 0.9045$, PRKG2: ****$P < 0.0001$ for all comparisons, $^{ns}P = 0.5738$; Hippocampus: CHIP: ****$P < 0.0001$ for all comparisons, $^{ns}P = 0.1177$, PDE9A: ****$P < 0.0001$ for all comparisons, $^{ns}P = 9289$, PRKG1: ****$P < 0.0001$ for all comparisons, $^{ns}P = 0.5815$, PRKG2: ****$P < 0.0001$ for all comparisons, $^{ns}P = 0.1878$. Each summary panel shows the means ± SDs and summary plot (**A–G**). The P-value are shown from left to right. Source data are available online for this figure.

phosphorylation were significantly activated (Fig. 9I), which corroborates the inhibitory effect of PDE9A on CHIP-S20 phosphorylation, subsequently reducing CHIP and p-CHIP levels in Purkinje cells.

In addition, to provide a comprehensive assessment of the effects of AAV–CHIP and Bay 73-6691 treatment on the animal model of CHIP mutation, we used a heatmap to highlight the prominent changes in gene expression in pathways such as cAMP signalling, cGMP-PKG signalling, oxidative phosphorylation, ubiquitin-mediated proteolysis, mitophagy and autophagy. Moreover, the Wt, AAV–CHIP-treated and Bay 73-6691-treated groups exhibited greater consistency than the Hom group did. These results align closely with previous conclusions and again confirm the potential therapeutic targets of PDE9A and CHIP for ARCA (Appendix Fig. S9). These findings confirmed that CHIP mutations modulate PKAc ubiquitination and the competitive increase in PDE9A. Moreover, after treatment with Bay 73-6691, normal wild-type rats exhibited activation of cGMP signalling proteins, whereas cAMP signalling proteins were inhibited (Appendix Fig. S10). These results effectively demonstrate that CHIP downregulates the affinity of PDE9A for cGMP hydrolysis, thereby mediating cGMP and cAMP signalling crosstalk.

Simultaneously, we compared the differences in the pathogenic pathways between other cell subtypes and Purkinje cells. The differentially expressed genes in five subgroups of cells, including granule cells, Bergmann glia cells, glutamatergic neurons, oligo-dendrocyte precursor cells and oligodendrocytes, were significantly enriched in the cAMP signalling pathway and cGMP-PKG signalling pathway, indicating similar pathway disturbances to those in Purkinje cells (Appendix Fig. S11). Although Bay 73-6691

and CHIP overexpression could eliminate some abnormal changes in certain cell subgroups, AUCell analysis only revealed changes in granule cells in these pathways, which were similar to the aforementioned changes in Purkinje cells, whereas such changes were not significant in other cell subtypes. This finding suggests the possible involvement of different biological mechanisms (Appendix Fig. S12).

The scRNA-seq results not only support the notion that CHIP mutations reduce the formation of K63 and K27 polyubiquitin chains on PDE9A, impacting PDE9A degradation in autophago-lysosomes but also indicate that PKG mediates the reduction in p-CHIP and the shortened half-life of CHIP resulting from increased PDE9A levels. The results also confirmed that the application of Bay 73-6691 and AAV-mediated exogenous CHIP expression could ameliorate the imbalance of the intracellular cGMP/cAMP pathways, indicating the potential for ARCA therapy and neuroprotection.

## Discussion

CHIP enhances the clearance of damaged/misfolded proteins and suppresses protein toxicity, impacting immunity, ageing, metabolic stress and a myriad of human diseases, including cardiovascular and neurodegenerative diseases. Homozygous mutations in CHIP have been directly linked to ARCA. Our study revealed the upregulation of PDE9A expression as a critical pathological mechanism associated with CHIP mutations. CHIP and PDE9A mutually regulate their expression levels and interact through the TPR domain of CHIP. The TPR domain potentially facilitates the

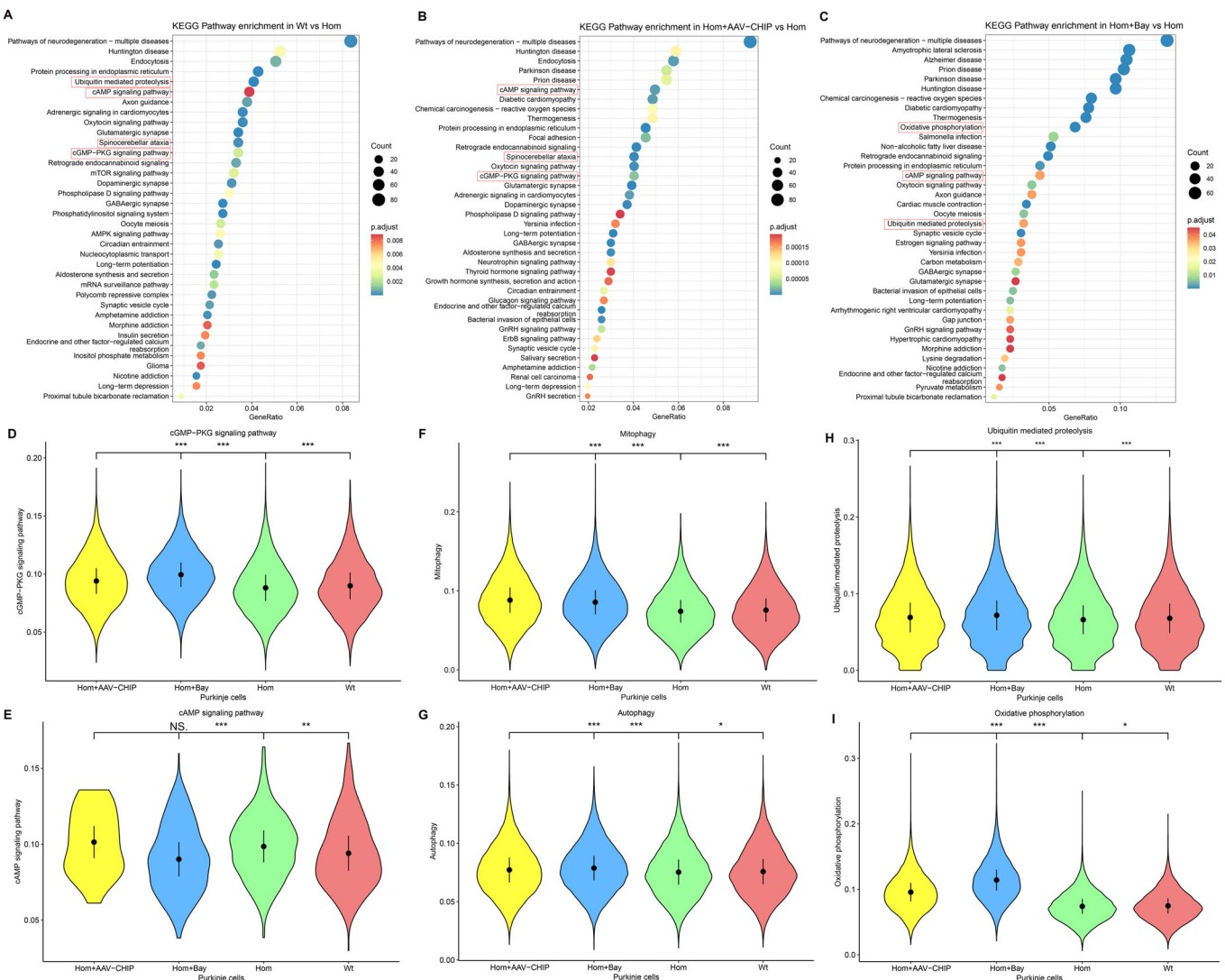

**Figure 9. Bidirectional posttranslational modifications of PDE9A and CHIP.**

(A–C) Differential gene KEGG enrichment analysis in Purkinje cells for the following comparisons: Wt vs. Hom, Hom+AAV-CHIP vs. Hom, and Hom+Bay(+) vs. Hom, Hom (*CHIP* p.T246M homozygous mutant rats), Wt (wild-type littermates), Hom+AAV-CHIP (AAV-CHIP-injected homozygous mutant rats), and Hom+Bay(+) (Bay 73-6691-treated homozygous mutant rats). The most relevant signalling pathways are highlighted in red boxes. Hypergeometric test. (D–I) AUCell pathway activity analysis revealed alterations in Purkinje cells across the four groups in multiple signalling pathways. Hom (*n* = 3. 47 cells), Wt (*n* = 3. 117 cells), Hom+AAV-CHIP (*n* = 3. 1148 cells), and Hom+Bay(+) (*n* = 3. 180 cells). T test, cGMP-PKG signalling pathway: \*\*\**P* < 2.22E−16, \*\*\**P* < 2.22E−16, \*\*\**P* = 1.8E−07; cAMP signalling pathway: [NS]*P* = 0.5500, \*\*\**P* = 3E−07, \*\**P* = 0.0066; Mitophagy: \*\*\**P* < 2.22E−16, \*\*\**P* < 2.22E−16, \*\*\**P* = 2E−09; Autophagy: \*\*\**P* < 2.22E−16, \*\*\**P* < 2.22E−16, \**P* = 0.0257; Ubiquitin-mediated proteolysis: \*\*\**P* < 2.22E−16, \*\*\**P* < 2.22E−16, \*\*\**P* = 8E−06; Oxidative phosphorylation: \*\*\**P* < 2.22E−16, \*\*\**P* < 2.22E−16, \**P* = 0.0190. Each summary panel shows the means ± SDs and summary plot (D–I). The *P*-value are shown from left to right.

bridging of Hsp70 to PDE9A, and the U-box domain of CHIP recruits K63- and K27-linked ubiquitination to modify PDE9A, steering its degradation via autolysosomes. Upon *CHIP* mutation, PDE9A is hindered from undergoing autophagic degradation and instead accumulates aberrantly. This increases PDE9A levels, resulting in cGMP hydrolysis, inhibition of CHIP S19 phosphorylation and decreased CHIP stability. A feedback loop ensues, where cAMP signalling intensifies and PKAc competitively inhibits the ubiquitin-mediated degradation of PDE9A. This perturbation of augmented aberrant PDE9A and depleted CHIP suppresses intracellular cGMP signalling, thereby disrupting mitophagy

homeostasis and triggering cell death. However, by inhibiting PDE9A activity through Bay 73-6691, the aforementioned pathological outcomes can be ameliorated. Given the crucial role of CHIP in cardiac, skeletal muscle, neurogenesis and oncologic diseases, our findings hold therapeutic significance and reveal that Bay 73-6691 is a promising treatment for *CHIP* mutation-associated ARCA. Our research revealed the crucial role of CHIP in maintaining the cellular balance of cAMP and cGMP. Using scRNA-seq technology, we found that both the cGMP-PKG and cAMP signalling pathways exhibited distinct patterns in the *CHIP* mutation model, with the cGMP-PKG signalling pathway being

**Purkinje cell**

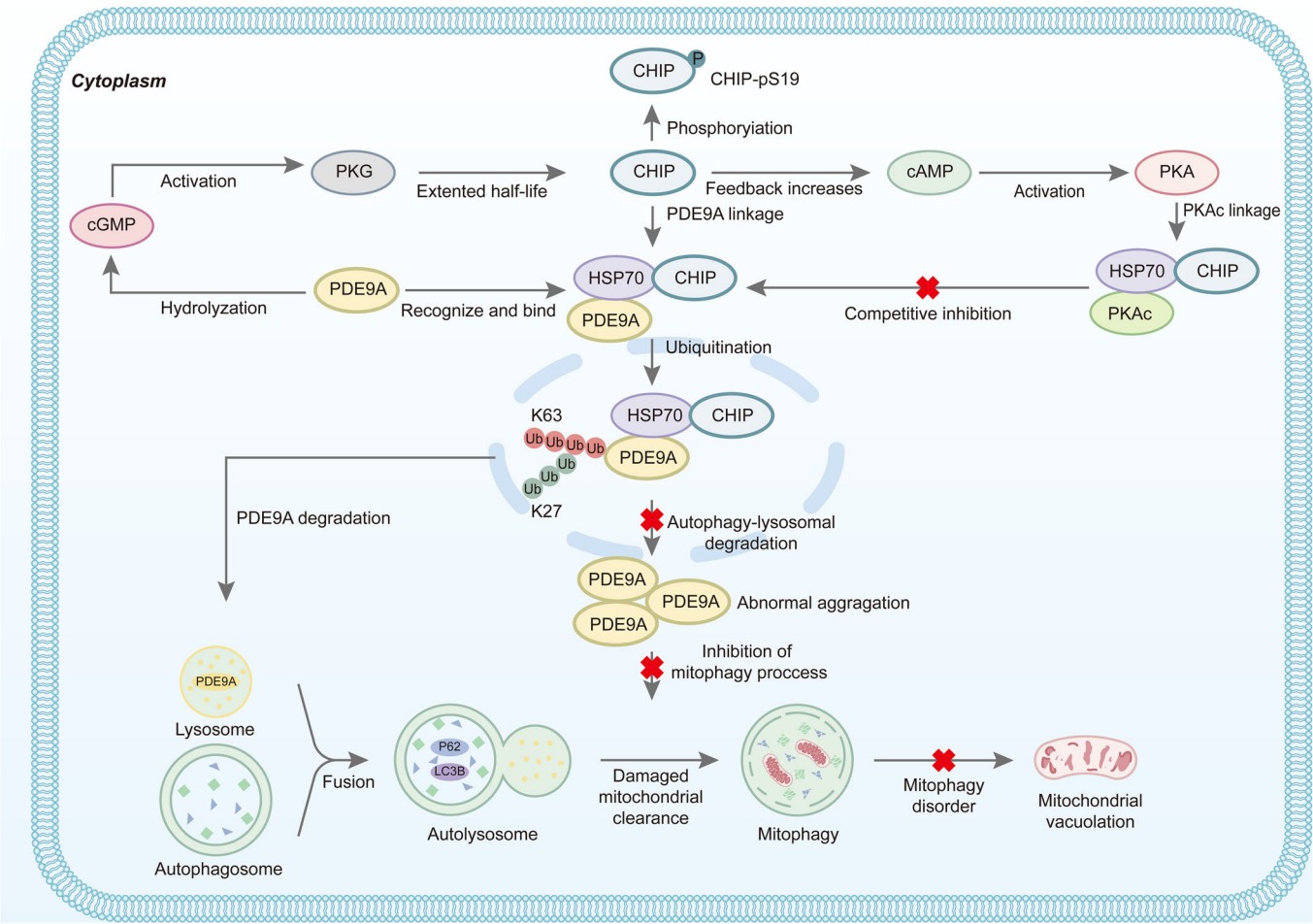

**Figure 10. Schematic of the Regulatory Mechanism Between CHIP and PDE9A.**

The diagram fully illustrates the regulatory network of CHIP and PDE9A within Purkinje cells, demonstrating the involvement of cGMP and cAMP signaling pathways and their implications for mitophagy and proteostasis in neuronal apoptosis. Arrows indicate activation or progression of processes, while red crosses denote inhibition.

inhibited and the cAMP signalling pathway being activated. Exogenous overexpression of CHIP or application of Bay 73-6691 upregulated CHIP, cGMP, PKG1 and PKG2 while reducing PDE9A. As the level of CHIP increased and PDE9A activity was inhibited, the cAMP and PKA levels significantly decreased. Our results suggest that the interaction between CHIP and PDE9A is a potential bidirectional regulator of the cGMP and cAMP signalling pathways (Fig. 10).

The intracellular crosstalk of the second messengers cAMP and cGMP plays a vital role in maintaining normal biological functions in cellular signal transduction. PKA opens calcium ion channels, activates cAMP-activated exchange proteins and initiates downstream pathways. As a second messenger, PKG is limited primarily to vascular, pulmonary and brain tissues. In neuronal cells, cAMP and cGMP mediate antagonistic cellular behaviours, such as ion channel activity, cell volume regulation and axon guidance, in response to external factors. Local cAMP and cGMP within the undifferentiated axons of neuronal cells promote or inhibit axon formation and play contrasting roles during dendritic growth. These two signalling pathways antagonise each other and mutually

regulate and collectively govern cellular physiological functions (Bos, 2006; Ricciarelli and Fedele, 2018; Stork and Schmitt, 2002). In cardiac myocytes, cAMP hydrolytic activity is regulated by cGMP, where cGMP allosterically stimulates the hydrolysis of cAMP in PDE2. In PDE3, cGMP competitively inhibits the hydrolysis of cAMP (Zaccolo and Movsesian, 2007). However, PDE9A acts as a selective inhibitor of cGMP, exclusively hydrolysing only cGMP and inhibiting the cGMP-PKG signalling pathway. In neurons, the bidirectional regulation of PDE9A by CHIP is essential for mediating the crosstalk between cAMP and cGMP. The PDE9A inhibitor Bay 73-6691 maintains the balance of CHIP-mediated cGMP/cAMP signalling crosstalk, ameliorating Purkinje cell mitophagy disturbance-induced ARCA neuropathological alterations.

Moreover, the pathological manifestations of ARCA are multifaceted, with common occurrences encompassing neuronal atrophy and degeneration, demyelination and the disappearance of Purkinje cells, coupled with glial cell proliferation, causing widespread degeneration in the cerebellar hemispheres and vermis (Synofzik et al, 2019). Thus, our initial validation in an ARCA animal model

revealed CHIP and PDE9A in Purkinje cells as therapeutic targets for ARCA. To further explore the pathological alterations mediated by CHIP, scRNA-seq was used to observe the molecular alterations across distinct cell types within the cerebellum and hippocampus. Although granule cells and Purkinje cells possess analogous pathogenic mechanisms, we observed that pathogenicity could be ameliorated by both Bay 73-6691 treatment and CHIP overexpression. Granule cells, which are pervasively distributed within the hippocampus and cerebellum and are instrumental in maintaining hippocampal cognitive function and cerebellar equilibrium, predominantly contribute, alongside Purkinje cells and other neurons, to neurological aberrations in CHIP mutation-induced ARCA, thereby providing new insights into pre-existing research findings. In addition, the observation of disparate pathway activities in various glial cells, such as Bergmann glial cells, oligodendrocyte precursor cells and oligodendrocytes, as well as in glutamatergic neurons juxtaposed with Purkinje cells, suggests disparate pathological mechanisms, findings that might be relevant to the pathogenesis of other CHIP-mediated diseases, as previously reported.

Numerous NDDs are caused by the aggregation of misfolded and aggregation-prone proteins, which typically reside in ubiquitin-associated intraneuronal inclusion bodies, resulting in protein toxicity that disrupts the normal function of neuronal cells through impairments in protein quality control mechanisms (Wilson et al, 2023). Recent research has confirmed that alterations in the CHIP gene are implicated in the pathogenesis of various neurodegenerative disorders, including Alzheimer's disease (AD), Parkinson's disease (PD), amyotrophic lateral sclerosis (ALS), polyglutamine diseases (PolyQ diseases), intracerebral haemorrhage (ICH), and ischaemic stroke, all of which culminate in the loss of neuronal function and manifest relevant clinical phenotypes (Kaushik et al, 2024; Kim et al, 2023; Liu et al, 2023; Zhang et al, 2020b; Zhang et al, 2024). The cAMP and cGMP signalling systems play crucial roles in synaptic transmission, neuronal excitability, neural plasticity, and neuroprotection in hippocampal cells (Argyrousi et al, 2020). Dysfunctions in the cascading signalling of cGMP and cAMP can produce a broad spectrum of neuropsychiatric symptoms, such as those observed in AD and PD (VerPlank et al, 2020; Xiang et al, 2024). Disruption of cAMP response element-binding protein (CREB) affects the transcription of brain-derived neurotrophic factor (BDNF), leading to the loss of normal function of the specific protein Huntington in Huntington's disease, where a reduction in cGMP is also observed in the context of highly expanded PolyQ proteins (Myeku and Duff, 2018). In ALS, cAMP activity is significantly associated with the pathological aggregation of TDP-43. In mouse models of cerebral haemorrhage, enhancing cAMP/PKA signalling can attenuate microglial activation, reduce cytokine/chemokine expression, and improve local microcirculation (Myeku and Duff, 2018). Although PKA and PKG possess very distinct physiological functions, they share a critical common feature: enhancing the degradation of short-lived misfolded proteins by the ubiquitin–proteasome system (UPS), making the balance of cAMP/cGMP signalling critically important in neuronal death and morphological abnormalities in these NDDs (Cuinat et al, 2023; Lee and Goldberg, 2022; Xiang et al, 2024).

In addition, our results revealed that mitophagy was impaired in the CHIP-mutated ataxia rodent model. Mitophagy is a highly conserved cellular process that effectively clears damaged mitochondria, playing an important role in energy supply, metabolic homeostasis, and neuronal survival (Stavoe and Holzbaur, 2019). When pathogenic or mutated proteins appear in cells, metabolic homeostasis is affected, resulting in damaged mitochondria (Jetto et al, 2022). Dysfunctional mitophagy is involved in many NDDs, such as PD, AD, hereditary ataxia (HA), and traumatic brain injury. Targeting mitophagy may be suitable for the treatment of NDDs and acute brain injury (Zhang et al, 2021). Inhibiting PDE9A or increasing CHIP levels improved mitochondrial morphology, reduced the incidence of autophagy disorders, and rescued neuronal death in NDDs.

Given that, in the early stages of NDDs, targeting PDEs is a logical strategy for preventing or halting the progression of these disorders. PDE2 inhibitors are used to treat heart failure and pulmonary hypertension (Bubb et al, 2014), PDE5 inhibitors are known for their ability to treat erectile dysfunction and are thought to have benefits in Alzheimer's disease (Adesuyan et al, 2024), and PDE10 inhibitors are used to explore treatments for Huntington's disease and schizophrenia (Halene and Siegel, 2007). This study confirms the bidirectional regulation between CHIP and PDE9A, which mediate cGMP/cAMP signal crosstalk and could become a common therapeutic direction for these NDDs, making PDE9A and CHIP a promising strategy for exploring NDD therapeutics. Nonetheless, the PDE9A inhibitor Bay 73-6691 is currently in the early stages of preclinical research (Fedele and Ricciarelli, 2021). It remains unclear whether Bay 73-6691 is suitable for human trials or whether it is still solely a laboratory research tool. The present study revealed that Bay 73-6691 could ameliorate neurological damage in the CHIP mutation model through a dual ubiquitination–phosphorylation pathway, potentially serving as a candidate drug for treating CHIP-related diseases and thereby offering a new approach for the treatment of related diseases, especially CHIP mutation-related ataxias. Research has also suggested that Bay 73-6691 regulates the cGMP/cAMP signalling in the CNS and participates in the control of CHIP-related PQCs, elucidating potential new drug mechanisms and target points of Bay 73-6691. Our study revealed mechanistic insights into the mutual regulatory relationship between CHIP and PDE9A, suggesting that both may become potential targets for the treatment of NDDs. Therefore, in future diagnostics and treatments of NDDs, we could focus more research on the interaction between CHIP and PDE9A, which holds great promise for providing a new strategy for the treatment of NDDs.

# Methods

**Reagents and tools table**

| Reagent/Resource | Reference or Source | Identifier or Catalog Number |
|---|---|---|
| **Experimental models** | | |
| Human embryonic kidney (HEK) 293 (HEK293T) cells | Chinese Academy of Sciences Cell Bank | |
| CHIP p.T246M homozygous mutant Rattus norvegicus | https://doi.org/10.1093/hmg/ddt497 | |
| **Recombinant DNA** | | |
| HBAAV2/BBB-CHIP virus | https://doi.org/10.1007/s12975-019-00715-w | |

| Reagent/Resource | Reference or Source | Identifier or Catalog Number |
|---|---|---|
| pCMV-PDE9A-3xflag | Shanghai GeneChem Co., Ltd. | GOSE0321011 |
| pCMV-V5-6his-PDE9A | Shanghai GeneChem Co., Ltd. | GOSE0321374 |
| pCMV-V5-6his-PDE9A(K186R) | Shanghai GeneChem Co., Ltd. | GOSE0420844 |
| pCMV-3xflag-STUB1 | Shanghai GeneChem Co., Ltd. | GOSE0377369 |
| pCMV-STUB1-HA | Shanghai GeneChem Co., Ltd. | GOSE0377351 |
| pCMV-3xflag-STUB1 (del226-300aa) | Shanghai GeneChem Co., Ltd. | GOSE0377371 |
| pCMV-3xflag-STUB1 (del26-aa127) | Shanghai GeneChem Co., Ltd. | GOSE0377370 |
| pCMV-3xflag-STUB1(T246M) | Shanghai GeneChem Co., Ltd. | GOSE0377352 |
| pCMV-3xflag-STUB1(K30A) | Shanghai GeneChem Co., Ltd. | GOSE0377353 |
| pCMV-HA-ubiquitin-WT | Shanghai GeneChem Co., Ltd. | GOSE0321371 |
| pCMV-HA-ubiquitin-K48 | Shanghai GeneChem Co., Ltd. | GOSE0378320 |
| pCMV-HA-ubiquitin-K48R | Shanghai GeneChem Co., Ltd. | GOSE0378319 |
| pCMV-HA-ubiquitin-K63 | Shanghai GeneChem Co., Ltd. | GOSE0378322 |
| pCMV-HA-ubiquitin-K63R | Shanghai GeneChem Co., Ltd. | GOSE0378321 |
| pCMV-HA-ubiquitin-K11 | Shanghai GeneChem Co., Ltd. | GOSE0382801 |
| pCMV-HA-ubiquitin-K11R | Shanghai GeneChem Co., Ltd. | GOSE0389030 |
| pCMV-HA-ubiquitin-K27 | Shanghai GeneChem Co., Ltd. | GOSE0382802 |
| pCMV-HA-ubiquitin-K27R | Shanghai GeneChem Co., Ltd. | GOSE0389031 |
| pCMV-HA-ubiquitin-K29 | Shanghai GeneChem Co., Ltd. | GOSE0389032 |
| pCMV-HA-ubiquitin-K29R | Shanghai GeneChem Co., Ltd. | GOSE0389033 |
| pCMV-HA-ubiquitin-K0 | Shanghai GeneChem Co., Ltd. | GOSE0389135 |
| pCMV-myc-HSP70 | Shanghai GeneChem Co., Ltd. | GOSE0418622 |
| shRNA-PDE9A | Shanghai GeneChem Co., Ltd. | GIEE0321372 |
| shRNA-STUB1 | Shanghai GeneChem Co., Ltd. | GIEE0321370 |
| **Antibodies** | | |
| Anti-PDE9A-rabbit | ProteinTech, China | 12648-1-AP |
| Anti-PDE9A-mouse | Santa Cruz, USA | Sc-166375 |
| Anti-PDE9A-rabbit | Abcam, UK | Ab168432 |
| Anti-STUB1-rabbit | Abcam, UK | Ab134064 |
| Anti-STUB1-mouse | Santa Cruz, USA | Sc-133083 |

| Reagent/Resource | Reference or Source | Identifier or Catalog Number |
|---|---|---|
| Anti-HA-tag-mouse | Abcam, UK | Ab49969 |
| Anti-DYKDDDDK-mouse | ProteinTech, China | 66008-4-Ig |
| Anti-V5-rabbit | ProteinTech, China | 14440-1-AP |
| Anti-V5-mouse | Abcam, UK | Ab206571 |
| Anti-ubiquitin-mouse | Abcam, UK | Ab303664 |
| Anti-PRKG1-rabbit | ProteinTech, China | 21646-1-AP |
| Anti-PRKG2-rabbit | ProteinTech, China | 55138-1-AP |
| Anti-cAMP-rabbit | Abcam, UK | Ab76238 |
| Anti-PRKACA-rabbit | ProteinTech, China | 27398-1-AP |
| Anti-calbindin-rabbit | Abcam, UK | Ab108404 |
| Anti-β-tubulin-rabbit | ProteinTech, China | 10094-1-AP |
| Anti-GAPDH-mouse | ProteinTech, China | 60004-1-Ig |
| Anti-pho-CHIP | Abmart, China | P26709M |
| IPKine™ HRP, Mouse Anti-Rabbit IgG LCS | Abbkine, China | A25022 |
| IPKine™ HRP, Goat Anti-Mouse IgG HCS | Abbkine, China | A25112 |
| HRP-conjugated AffiniPure goat anti-rabbit IgG(H + L) | ProteinTech, China | SA00001-2 |
| HRP-conjugated AffiniPure Goat Anti-mouse IgG(H + L) | ProteinTech, China | SA00001-1 |
| Goat anti-rabbit IgG H&L (Alexa Fluor®488) | Abcam, UK | Ab150077 |
| goat anti-mouse IgG H&L (Alexa Fluor®594) | Abcam, UK | Ab150116 |
| **Oligonucleotides and other sequence-based reagents** | | |
| PCR Primers-*CHIP* | Shangya, China | |
| PCR Primers-*PDE9A* | Shangya, China | |
| PCR Primers-*GAPDH* | Shangya, China | |
| **Chemicals, Enzymes and other reagents** | | |
| Bafilomycin A1 | MCE, USA | HY-100558 |
| Bortezomib | MCE, USA | HY-10227 |
| Forskolin | MCE, USA | HY-15371 |
| Bay 73-6691 | MCE, USA | HY-104028 |
| Cycloheximide | MCE, USA | HY-12320 |
| **Software** | | |
| GraphPad Prism 9 software | GraphPad Software, USA | |
| RStudio (R 4.2.2) | RStudio, USA | |
| **Other** | | |
| MGISEQ-2000 platform | BGI Genomics, China | |

## Plasmids and antibodies

All the plasmids used in this study were purchased from Shanghai GeneChem Co., Ltd.: pCMV-PDE9A-3xflag, pCMV-V5-6his-PDE9A, pCMV-3xflag-STUB1, pCMV-STUB1-HA, pCMV-3xflag-STUB1(del226-300aa), pCMV-3xflag-STUB1(del26-aa127), pCMV-3xflag-

STUB1(T246M), pCMV-3xflag-STUB1(K30A), pCMV-HA-ubiqui-tin-WT, pCMV-HA-ubiquitin-K48, pCMV-HA-ubiquitin-K48R, pCMV-HA-ubiquitin-K63, pCMV-HA-ubiquitin-K63R, pCMV-HA-ubiquitin-K11, pCMV-HA-ubiquitin-K11R, pCMV-HA-ubiquitin-K27, pCMV-HA-ubiquitin-K27R, pCMV-HA-ubiquitin-K29, pCMV-HA-ubiquitin-K29R, pCMV-HA-ubiquitin-K0, pCMV-myc-HSP70, shRNA-PDE9A, shRNA-STUB1-1, shRNA-STUB1-2 and shRNA-STUB1-3.

The antibodies used were as follows: anti-PDE9A-rabbit (ProteinTech, China), anti-PDE9A-mouse (Santa Cruz, USA), anti-STUB1-rabbit (Abcam, UK), anti-STUB1-mouse (Santa Cruz, USA), anti-HA-tag-mouse (Abcam, UK), anti-DYKDDDDK-mouse (ProteinTech, China), anti-V5-rabbit (ProteinTech, China), anti-GFP-tag-rabbit (Abcam, UK), anti-ubiquitin-mouse (Abcam, UK), anti-PRKG1-rabbit (ProteinTech, China), anti-PRKG2-rabbit (ProteinTech, China), anti-CREB1-mouse (ProteinTech, China), anti-cAMP-rabbit (ProteinTech, China), anti-PRKACA-rabbit (ProteinTech, China), anti-calbindin-rabbit (Abcam, UK), anti-GAPDH-mouse (ProteinTech, China), anti-pho-CHIP (Abmart, China), HRP-conjugated AffiniPure goat anti-rabbit IgG(H + L) (ProteinTech, China), HRP-conjugated AffiniPure Goat Anti-mouse IgG(H + L) (ProteinTech, China), Goat anti-rabbit IgG H&L (Alexa Fluor®488) (Abcam, UK) and goat anti-mouse IgG H&L (Alexa Fluor®594) (Abcam, UK).

## Cell culture and plasmid transfection

Human embryonic kidney (HEK) 293 (HEK293T) cells obtained from the Chinese Academy of Sciences Cell Bank were cultured in high-glucose DMEM (HyClone, China) supplemented with 10% FBS (BI, China). All the plasmids were transfected into 293T cells via the Lipofectamine®3000 transfection kit (Invitrogen, USA) according to the manufacturer's protocol. Six hours posttransfection, the medium was replaced with medium containing bafilomy-cin A1 (50 nM, MCE, USA), bortezomib protease inhibitors (100 nM, MCE, USA), the cAMP agonist forskolin (50 μM, MCE, USA) or the PDE9A inhibitor Bay 73-6691 (200 μg/mL, MCE, USA) for specified durations prior to further analysis.

## Animal husbandry and treatment

All animal experiments were approved by the Ethics Committee of Zhengzhou University Animal Research Centre and were con-ducted in a specific pathogen-free facility. The animals used in this study were the progeny of the previously characterized CHIP p.T246M homozygous mutant *Rattus norvegicus* (Shi et al, 2014). Genotyping was performed via tail DNA extraction and amplifica-tion with specific primers, followed by Sanger sequencing. To prepare the Bay 73-6691 injectable solution, we dissolved 1 ml of a 100 mg/ml Bay 73-6691 solvent in 50 ml of corn oil (Solarbio, China). This mixture was then thoroughly homogenized using an orbital shaker to achieve a final concentration of 2 mg/ml Bay 73-6691. Rats aged 5.5 months received intraperitoneal injections of Bay 73-6691 (2 mg/kg/day) for 15 consecutive days, followed by a 7-day interval prior to further experiments. In addition, 5.5-month-old male homozygous *Rattus norvegicus* mutants were subjected to intracranial CHIP overexpression via 1.8% isoflurane anaesthesia and intravenous injection of HBAAV2/BBB-CHIP virus (1.2 × 10^12 vg/mL; HanBio Biotechnology, China).

The viral construct employed in this study was fundamentally derived from the architecture utilized in our preceding work. For the current experimental setup, the GFP tag previously incorpo-rated into the virus was excised, while the remaining structural components were preserved (Zhang et al, 2020a). The AAV vector sequence was constructed according to the work of Deverman and named AAV/BBB (Deverman et al, 2016). The vector used a CMV promoter, and the connection of T2A enables CHIP and Flag to be expressed separately under the same promoter.

## Single-cell RNA sequencing and bioinformatics analysis

Groups of four *Rattus norvegicus*, each treated with Bay 73-6691, littermate wild-type males, p.T246M homozygous mutants, and those injected with HBAAV2/BBB-CHIP were anaesthetized with 1.8% isoflurane for 20 min. Fresh hippocampal and cerebellar tissues were rapidly harvested and processed for single-cell capture (Sin, China). The mRNA libraries were constructed and sequenced on the MGISEQ-2000 platform by BGI Genomics (China). Raw sequencing data in Fastq format were processed via the CeleScope pipeline to obtain 10× single-cell gene expression matrices. Subsequent data analysis was carried out in RStudio (R 4.2.2) using the Seurat package for dimensionality reduction, clustering and visualisation; Harmony for batch effect removal; and ClusterProfiler for Kyoto Encyclopaedia of Genes and Genomes (KEGG) enrichment analysis. Further statistical tests were con-ducted for differential gene expression and pathway activity.

## RT–qPCR

Total RNA was isolated via an RNA extraction kit, and cDNA was synthesized via the PrimeScript™ RT reagent kit with gDNA eraser according to the manufacturer's instructions (Takara, Japan). A 20-fold dilution of cDNA was utilized in combination with CHIP and PDE9A primers to quantify their mRNA expressions across different cellular models using RT-qPCR. The reaction was executed using an RT-qPCR kit according to the manufacturer's protocol, with triplicate controls for each group. The following primer pairs were used: *CHIP* forward: 5′-ATCCCCGACTACCT GTGTGGC-3′ and reverse: 5′-TGCTCCTCGATGTCCTTGCG-3′; *PDE9A* forward: 5′-TTGACTCCTCGACGCGATGTTC-3′ and reverse: 5′-CTGAGGGTGACAGGGTTGATGC-3′; and *GAPDH* forward: 5′-GGCGCTGAGTACGTCGTGGAGT-3′ and reverse: 5′-GGGCAGAGATGATGACCCTTTTG-3′. Each sample was mea-sured in triplicate.

## Immunoblotting

To prepare protein samples for western blot analysis, we lysed designated quantities of *Rattus norvegicus* cerebellar tissue, hippocampal tissue and harvested cells in RIPA buffer containing protease and phosphatase inhibitors at a ratio of 1000:10:10:10 (Solarbio, China). Subsequent to sonication for effective cellular disruption, protein concentrations were normalised across samples using the BCA protein assay. The proteins were then diluted in 5× SDS sample buffer (Beyotime, China) and denatured at 100 °C for 5 min. Electrophoresis was performed on a 4–20% BeyoGel™ Plus Precast PAGE gel for the Tris-Gly system (Beyotime, China), followed by transfer to a membrane. Protein markers were selected

for reference. Primary antibodies were incubated overnight at 4 °C at specified dilutions (Flag-mouse, 1:5000; PDE9A-mouse, 1:5000; CHIP-rabbit, 1:10,000; V5, 1:5000; HA, 1:20,000; ubiquitin, 1:1000; cAMP, 1:2000; PRCAKA, 1:1000; Myc, 1:2000; GAPDH, 1:5000; and β-tubulin, 1:5000). The membranes were subsequently washed and incubated with HRP-conjugated secondary antibodies (Affini-Pure goat anti-rabbit, 1:5000; AffiniPure goat anti-mouse, 1:5000). Detection was performed using an enhanced chemiluminescence (ECL) substrate and captured using an Amersham Imager 680 (GE Healthcare Bio-Sciences AB, Sweden).

For automated capillary western blot assays, protein samples were prepared at a final concentration of 2–3 μg/μL. Assays were conducted according to the standard operating procedures of the chemiluminescence training kit (PS-T001, ProteinSimple) and executed on a simple western Jess system (ProteinSimple). Protein cartridges with a size range of 12–230 kDa were selected. The primary antibodies used were diluted to the following concentrations: PDE9A-rabbit, 1:50; CHIP-rabbit, 1:100; PRKG1-rabbit, 1:30; PRKG2-rabbit, 1:25; P62-rabbit, 1:25; LC3B-rabbit, 1:40; and β-tubulin-rabbit, 1:100.

## ELISA

Cerebellar and hippocampal tissue cleavage by 200 μL of lysate was analysed for cGMP content via a rat cyclic guanosine monophosphate (cGMP) ELISA kit (OmnimAbs, USA). Each sample had six replicate wells.

## Protein half-life assay

HEK293T cells transfected with CHIP-HA, PDE9A-flag or cotransfected with both were treated with cycloheximide (40 μM, Beyotime, China) at multiple time points. The protein levels were assessed by western blotting.

## Coimmunoprecipitation (COIP)

Following plasmid transfection, the HEK293T cells were harvested and lysed using a 1000:10:10:10 ratio mixture of NP-40 lysis buffer, PMSF, protease inhibitors and phosphatase inhibitors to yield protein samples with a concentration of 5 μg/μL. The lysates were preincubated with BeyoMag™ Protein A + G beads (Beyotime, China) for 1 h at 4 °C to remove nonspecifically bound proteins. The supernatant was then discarded, and the samples were incubated with specific antibodies (V5-rabbit, 10 μg; GFP-rabbit, 10 μg; Myc-Mouse, 10 μg) on an orbital shaker at 4 °C for 12 h. Subsequently, 100 μL of fresh magnetic beads was added, and the incubation was continued overnight at 4 °C. After incubation, the magnetic beads were washed four times for 5 min each with NP-40 lysis buffer (Beyotime, China). The samples were then resuspended in 20 μL of 5× SDS sample buffer and 30 μL of NP-40 lysis buffer, followed by heating at 100 °C for 5 min. The supernatants were collected for further analysis via western blotting to detect the target protein.

## Immunostaining

For cellular immunofluorescence, the cells were fixed, permeabilised and incubated with primary antibodies overnight at 4 °C, followed by incubation with fluorescent secondary antibodies. For cerebellar tissue immunofluorescence, the rats were anaesthetised and perfused with 4% paraformaldehyde (Solarbio, China). The tissues were then processed and stained using similar protocols. Fluorescence imaging was conducted via an automated fluorescence microscope (3DHISTECH, Hungary) or multiphoto laser scanning microscopy (Carl Zeiss, Germany).

## Cellular ubiquitination assays

For in vivo ubiquitination, the plasmids Flag-CHIP, V5-His-PDE9A and HA-Ubiquitin (both wild-type and mutant forms) were transfected into HEK293T cells. Coimmunoprecipitation was conducted using an anti-V5 antibody (10 μg V5-rabbit), and subsequent western blot analyses were performed using primary antibodies targeting Flag (1:5000, mouse), HA (1:2500, mouse), and ubiquitin (1:1000, mouse).

## LC–MS/MS identification of ubiquitination sites

The samples subjected to COIP with V5 antibodies were separated via gel electrophoresis. The gel slices were digested with trypsin at a concentration of 10 ng/μL. The resulting peptides were dissolved in liquid chromatography mobile phase A and separated via an EASY-nLC 1200 ultrahigh-performance liquid chromatography system. Subsequent analysis was conducted with an Orbitrap Exploris 480 mass spectrometer (Thermo, MA, USA). Data were collected using a data-dependent acquisition (DDA) program. The secondary mass spectrometry data were searched using Proteome Discoverer 2.4. The database was set with the PDE9A protein sequence referencing the UniProt database (PDE9A_HUMAN ID: O76083-1). The settings included a maximum of four missed cleavages, a precursor ion mass tolerance of 10 ppm, and a fragment ion mass tolerance of 0.02 Da. Fixed modifications were set as cysteine alkylation; variable modifications included methionine oxidation, protein N-terminal acetylation, and lysine ubiquitination. The peptide ion score threshold was set above 20. Data were filtered using a false discovery rate (FDR) of 1%, and peak areas were mean normalized.

## Cell viability assay (CCK-8)

The cells were seeded at a density of 1000 cells per well in a 96-well plate and subjected to cell counting kit-8 assays (Dojindo, Japan) according to the manufacturer's instructions. Cell viability was assessed at 24, 48, 72, 96 and 120 h postseeding.

## Flow cytometry

The cells were harvested to obtain $1 \times 10^6$ cells per sample, filtered through a cell strainer to yield single-cell suspensions of $1 \times 10^5$ cells and then stained with the FITC Annexin V Apoptosis Detection Kit I (BD, USA). Subsequent analysis was conducted via flow cytometry (Beckman, USA).

## TEM

Cerebellar tissue samples were collected from 6.5-month-old rats treated with PDE9A, age-matched CHIP T246M homozygous mutant rats, wild-type littermates and 7.3-month-old CHIP-

overexpressing rats. The tissues were initially fixed in electron microscopy fixative solution (Servicebio, China) and further postfixed in 1% osmium tetroxide (Ted Pella Inc., China) for 2 h. After washing, the tissues were dehydrated through a graded ethanol series, infiltrated and embedded. Ultrathin sections were prepared and mounted on copper grids. Staining was performed using a 2% uranyl acetate solution (Servicebio, China) in the dark for 8 min, followed by three washes with 70% ethanol and ultrapure water. The sections were subsequently counterstained with 2.6% lead citrate for 8 min under $CO_2$-free conditions and washed three times with ultrapure water. Observations and image capture were performed using TEM (Hitachi, Japan).

### Nissl staining

Following the deparaffinisation and antigen retrieval processes of the tissue paraffin sections, staining was conducted using a Nissl stain kit (cresyl violet method) (Solarbio, China).

### Behavioural studies

#### Morris water maze

The Morris water maze test was conducted using the SANS Animal Behaviour Analysis System software (SansBio, China). The apparatus was divided into four quadrants: northeast (NE), southeast (SE), southwest (SW), and northwest (NW). From Day 1 to Day 5, the platform was submerged to a depth of 1.5 cm below the water surface, and the same procedure was repeated. On Day 6, the platform was removed, and exploratory behaviours were recorded for 60 s. Metrics, such as platform quadrant exploration time and its ratio to total exploration time. were analysed to assess cognitive and memory functions.

#### Rotarod test

Motor coordination and balance were assessed using an accelerating rotarod apparatus (Rot-Rod, Softmaze). The rats underwent a four-day training regimen consisting of three daily trials separated by 30-min intervals. On the fifth day, testing was carried out. The animals were placed on the rotarod at an initial speed of 4 rpm, which was gradually increased to 40 rpm over 5 min. The speed of the rotarod at the time of the second fall was recorded. Instances in which the animal clung to the rod were counted as falls after two occurrences.

#### Balance beam

Following established protocols, a wooden beam with a length of 150 cm and diameter of 18 mm was employed. The beam was elevated to a height of 40 cm. The rats were trained to traverse the beam from the starting point to the endpoint three times daily with 2-min intervals over four consecutive days. On the fifth day, the time taken to complete the traversal was recorded.

#### Gait analysis

A horizontal runway, 2 m in length and 20 cm in width, with 20-cm high sidewalls, was set up. Clean white paper was laid at the base of the runway. The forepaws were marked with blue nontoxic dye, and the hind paws were marked with red dye. The rats were released from the end of the runway and allowed to walk towards the opposite end for two consecutive days. On the third day, the paw prints were collected and analysed to assess gait patterns.

### Statistical analysis

The data are presented as the means ± SDs of individual experiments, and Microsoft Excel Version 2022, Prism 9 software and RStudio (R 4.2.2) were used to process the data. Sample sizes and individual statistical results for the details of all analyses are provided in the figures or legends.

## Data availability

The authors declare that the data supporting the findings of this study are available within the paper and its supplementary information files, all relevant data of the present paper are available from the corresponding author on reasonable request. Single-cell sequencing: PRJCA033500 (https://ngdc.cncb.ac.cn/omix/release/OMIX008238).

The source data of this paper are collected in the following database record: biostudies:S-SCDT-10_1038-S44318-024-00351-7.

## Peer review information

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

## Acknowledgements

Our research was supported by the National Natural Science Foundation of China (No. 81974211, No. 82171247 and No. 82371433), and Scientific Research and Innovation Team of The First Affiliated Hospital of Zhengzhou University (No. ZYCXTD2023011). Thanks to all authors for their efforts and support.

## Author contributions

**Xiaoyan Hao**: Validation; Visualization; Methodology; Writing—original draft. **Zhengwei Hu**: Validation; Methodology. **Mengjie Li**: Software; Methodology. **Shuo Zhang**: Resources; Methodology. **Mibo Tang**: Methodology. **Chenwei Hao**: Methodology. **Shasha Qi**: Methodology. **Yuanyuan Liang**: Visualization; Methodology. **Michael F Almeida**: Visualization; Methodology. **Kaitlan Smith**: Visualization; Methodology. **Chunyan Zuo**: Methodology. **Yanmei Feng**: Methodology. **Mengnan Guo**: Methodology. **Dongrui Ma**: Methodology. **Shuangjie Li**: Methodology. **Zhiyun Wang**: Methodology. **Yuemeng Sun**: Methodology. **Zhifen Deng**: Methodology. **Chengyuan Mao**: Methodology. **Zongping Xia**: Methodology. **Yong Jiang**: Data curation; Formal analysis; Supervision. **Yanxia Gao**: Data curation; Formal analysis; Supervision. **Yuming Xu**: Resources; Data curation; Formal analysis; Supervision; Writing—review and editing. **Jonathan C Schisler**: Resources; Data curation; Formal analysis; Supervision; Writing—review and editing. **Changhe Shi**: Conceptualization; Project administration; Writing—review and editing.

In addition to the CRediT author contributions listed above, the contributions in detail are:

XYH was responsible for the detailed design of the whole experiment, specific operation and manuscript written. ZWH performed rat breeding, genotype identification, and immunostaining. MJL performed Rstudio for analysing the sequencing data. SZ, MBT, CWH, and SSQ performed and assisted in immunoblotting assay. YYL, MFA, and KS contributed to the drawing of the schematic. CYZ, YMF, and MNG played crucial role in the cell culture. DRM, SJL, ZYW, and YMS performed in behavioural studies. ZFD, CYM, and ZPX provided guidance of in vivo and in vitro ubiquitination experiments. YJ and YXG mainly completed guidance of statistical analysis and article layout. YMX and JCS mainly completed guidance of statistical analysis, article layout, and review and editing. CHS contributed to the conception of this research, polished the article in written, improved the logical rationality of the article and ensured the final manuscript to submit.

Source data underlying figure panels in this paper may have individual authorship assigned. Where available, figure panel/source data authorship is listed in the following database record: biostudies:S-SCDT-10_1038-S44318-024-00351-7.

## Disclosure and competing interests statement

The authors declare no competing interests.

