## [Peer Review File · The EMBO Journal]

E3 Ubiquitin Ligase CHIP Facilitates cAMP and cGMP Signalling Cross-talk by Polyubiquitinating PDE9A

Changhe Shi, Xiaoyan Hao, Zhengwei Hu, Mengjie Li, Shuo Zhang, Mibo Tang, Chenwei Hao, Shasha Qi, Yuanyuan Liang, Michael F Almeida, Kaitlan Smith, Chunyan Zuo, Yanmei Feng, Mengnan Guo, Dongrui Ma, Shuangjie Li, Zhiyun Wang, Yuemeng Sun, Zhifen Deng, Chengyuan Mao, Zongping Xia, Yong Jiang, Yanxia Gao, Yuming Xu, and Jonathan Schisler

Corresponding author(s): Changhe Shi (shichanghe@zzu.edu.cn) , Jonathan Schisler (schisler@unc.edu), Yuming Xu (xuyuming@zzu.edu.cn)

Review Timeline:

Submission Date:	30th Jan 24
Editorial Decision:	13th Mar 24
Appeal Received:	10th Jun 24
Editorial Decision:	4th Sep 24
Revision Received:	8th Oct 24
Editorial Decision:	9th Dec 24
Revision Received:	12th Dec 24
Accepted:	12th Dec 24

Editor: Hartmut Vodermaier

Transaction Report:

Dr. Changhe Shi
The First Affiliated Hospital of Zhengzhou University
Neurology
First affiliated hospital of Zhengzhou University, Zhengzhou, Henan, China
Henan 450000
China

13th Mar 2024

Re: EMBOJ-2024-116833
E3 Ubiquitin Ligase CHIP Facilitates the cAMP and cGMP Signaling Cross-talk by Polyubiquitinating PDE9A

Dear Dr. Shi,

Thank you for submitting your manuscript on CHIP roles in cAMP/cGMP signaling crosstalk to The EMBO Journal. We sent it to three expert referees, who have now returned the below-copied reports. I am afraid to say that in light of their comments, we had to conclude that the manuscript is at this stage not a sufficiently strong candidate for publication in our journal. Although referee 1 is generally supportive of the study, referee 2 and 3 raise a number of major concerns, which potentially compromise key conclusions of the study. I will not repeat all these criticisms in detail here in this letter; however, recurrent problems are overinterpretation of data, internal inconsistencies, partly weak or non-quantitated effects, as well as missing clarity of presentation. I am therefore afraid we have to consider the study presently still too preliminary to justify inviting a revised version of the manuscript.

Should further work along the lines suggested by the referees allow you to significantly strengthen the main conclusions and better present the experimental evidence and logic, I would not exclude the possibility of once more looking into the study and considering a potential resubmission. In this case, I would however appreciate if you first contacted me with a detailed point-by-point response and summary of new results and their conclusions, so that we could determine whether a new version might be sufficiently promising to go back to our referees.

I am sorry that the reports do not allow me to be more positive at this point, but hope that you will nevertheless find the referees' comments and suggestions helpful when considering how to proceed further with this study. Thank you once more for having had the opportunity to consider this work for publication.

Yours sincerely,

Hartmut Vodermaier

Referee #1:

This excellent study makes highly novel and extremely exciting discoveries that will impact several significant fields as well as pointing to potential novel molecular pathologies, novel therapeutic approaches and novel diagnostics. Cyclic AMP and cyclic GMP are universal signalling molecule that underpins many critical physiological processes. Changes in cAMP/cGMP signalling underpin various pathologies and have been exploited therapeutically. Cyclic nucleotide phosphodiesterases (PDEs) provide the sole means of degrading these two cyclic nucleotides and through their targeting to specific signalling complexes and intracellular sites, shape gradients of cAMP/cGMP in cells that underpin compartmentalisation of cAMP signalling. The plethora of PDE families generate isoforms with distinct cellular distribution, regulatory and kinetic properties as well as intracellular location conferring the ability to regulate specific physiological processes upon particular PDE isoforms. PDE9A is a phosphodiesterase specifically hydrolyses cGMP with high affinity. This enzyme is encoded by a single gene but is expressed as more than 20 isoforms arising from alternative mRNA alternative splicing. Most studies to date have focused on its role in cognitive function, however PDE9A mRNA has been detected in cardiac and other tissues, although its wider role

remains to be ascertained. Interestingly, data mining has shown that PDE9A transcripts are downregulated in various types of colorectal cancer, although the functional significance of this has yet to be defined.

In this study the authors examine CHIP, which orchestrates the degradation of aberrant proteins through proteasomal and autophagy-lysosomal pathways to maintain cellular functional integrity. The authors used exogenous expression of CHIP and PDE9A in a cell culture system, to demonstrate that CHIP overexpression decreased PDE9A protein levels in a dose-dependent manner while PDE9A over-expression decreased CHIP expression in a dose-dependent manner. They went on to demonstrate that such actions were not mediated by altered mRNA levels but, rather, at the post-translational level.

Fascinatingly, the authors showed that the TPR domain of CHIP allowed CHIP to form a complex with PDE9A that co-localised in cells. CHIP is known to promote the E2-ligase driven polyubiquitination of substrate proteins and the authors demonstrate that PDE9A is no exception to this. The functional outcome of this was shown to be polyubiquitination of PDE9A directing it to accelerated degradation occurring through the autophagy-lysosomal pathway. These novel discoveries offer an explanation as to why animal models with CHIP mutations phenotypes exhibit upregulation of PDE9A expression. The physiological significance of this is uncovered in this study because the progressive ataxia phenotype accompanied by cerebellar neuronal degeneration and demise can be corrected by dosing with a PDE9A selective inhibitor.

1. Fig 1a. Instead of 'concentration' please simply indicated the micrograms of plasmid used. In the graphs it's not at all clear what the error bars mean as there is no indication as to whether these refer to replicate experiments or replicate scans. One would hope that these graphs described triplicate experiments. Are errors means ? SD or SEM? Please define in legends.

2. Fig 1 other graphs. Again, please define the number of experiments performed for analyses.

3. Fig 1L. The co-localisation is not terribly convincing at all. There are cells where it clearly doesn't occur and cells with mixtures. This is probably not surprising because, even if it is occurring, then the two proteins would have to be expressed at similar levels assuming equimolar expression. Indeed, it would be interesting to know the relative expression levels, although that would be challenging to perform. However, a more focussed analysis of the co-localisation should be attempted. Input from a laboratory specialising in such studies would help.

4. Fig 3. Animal numbers?

5. Fig 4. Experiment numbers?

6. Fig 5. Experiment numbers?

7. Fig 9. Experiment numbers?

8. Have the authors ever attempted to define the half life of PDE9A in 'normal' cells and those overexpressing CHIP?

Referee #2:

The paper presents a novel investigation into the molecular interplay between CHIP (Carboxyl terminus of Hsc70-Interacting Protein) and PDE9A (Phosphodiesterase 9A), suggesting a new avenue for understanding the molecular mechanisms underlying CHIP-linked spinocerebellar ataxia. The study proposes that inhibiting PDE9A could be a viable strategy for mitigating disease symptoms, offering valuable insights into the pathology and potential therapeutic avenues for CHIP T246M-caused ataxia.

General comments

- The importance and relevance of the study are well-established, building upon previous research published in PLoS Genet by Shi et al. in 2018. The continuation of research using the mutant mouse model is commendable.
- The manuscript, however, appears to have been prepared hastily, with issues in experimental design, figure description, writing style, and organization. Meticulous attention to detail and substantial revisions are required for consideration in a prestigious journal like EMBO J.
- The abstract requires revision for enhanced clarity and conciseness, ensuring the concept and rationale for researching the interplay between CHIP and PDE9A are immediately apparent. Similarly, the introduction would benefit from precision in explaining the subject matter, particularly clarifying the direct phosphorylation role of cGMP or its attribution to PKG.
- The manuscript lacks critical information on sample sizes, including the number of repetitions for in vitro and cellular experiments and the number of animals used in behavioral tests. Detailed descriptions of experimental procedures, such as transfection methods and plasmid amounts, are insufficient for reproducibility. Figure 1, in particular, demonstrates a lack of care in explaining the rationale behind studying CHIP and PDE9A reciprocal regulation and contains mismatches between figure legends and actual figures.
- The paper's clarity is undermined by unnecessary repetitions and typographical errors, e.g., misidentification of ubiquitination sites and interchangeable use of 'expression' with 'protein level'. The organization could be improved to eliminate redundancies

and inaccuracies, including mislabeling of figure references and statements on ubiquitination specificity. Some missing information makes it hard to make sense of the results, for example I did not find in the text a description of animal group names (e.g. Hom (6m-), Hom (6m+).

- While aesthetically pleasing, the figures lack comprehensive legends and context in the results section, making it challenging to understand the presented data. For instance, the behavioral tests and animal work sections need more methodological details, such as the solvent for the Bay 73-6691 compound and the immunostaining validation of CHIP overexpression.
- Issues include repeated mislabeling of figure references (e.g., 2B, 2C, 2F; referring to 1J before 1A, etc.), and inaccuracies such as stating ubiquitination occurs at arginine, specifically identifying the CHIP-mediated ubiquitination site at the 186th arginine residue of PDE9A. Additionally, the paper's organization could benefit from refinement, with several pieces of data more suitably placed in supplementary material. For instance, Figures 1 and 2, which convey overlapping information, are excessively large and could be condensed to prioritize clarity over quantity. When results are featured in a main figure, their explanation should be expanded upon, as seen with the cursory mentions of panels 1C-F.
- The manuscript presents inconsistencies between in vitro and in vivo findings, particularly in the immunoblotting results and the effects of Bay 73-6691 on CHIP levels. The paper's conclusions are not always supported by the presented evidence, necessitating additional experiments or clarifications.
- The discussion appears disconnected from the existing literature on the subject, mainly recapitulating the study's findings without integrating them into the broader research context. A more thorough engagement with related studies and a clearer articulation of how the current work advances the field would strengthen this section.
- The paper should provide quantitative electron microscopy data and detailed descriptions of mitochondrial morphology. Additionally, incorporating markers for mitophagy and mitochondrial networking could enrich the study's findings.
- Concerning the molecular docking results shown in S2A-B, this approach seems overly simplistic. The paper does not indicate whether alternative methodologies, such as molecular dynamics simulations, were considered. In the absence of such analyses, incorporating co-immunoprecipitation experiments could provide valuable insights into the suggested interaction between PDE9A and HSP70.
- The scRNA-seq section requires clarification and condensation, particularly concerning the experimental groups used for CHIP overexpression analysis.

Detailed review

- Figure 1 is especially prepared without enough care. It is not sufficiently expressed why the authors decided to study a reciprocal regulation between CHIP and PDE9A. Figure legends (A, B) do not match the actual figures. Lack of detailed description of transfection in the method section does not allow to reproduce these results, but amounts of plasmids used for co-transfection seem to be very high (Is it maximum 16 µg as described in the figure legend or 8 µg, as written on the plot?). What culture dishes were used for cell seeding and transfection? How were amounts of plasmids initially selected? Plots need complete information (X axis- missed units, Y axis- better name would be for example: CHIP/PDE9A level fold change). Blots are clean and generally skillfully performed, but the Fig. 1 conclusions about PFE9A and CHIP degradation pathways are not fully convincing based on presented evidence. Also, the authors provide approximate half-life of proteins but it cannot be easily deciphered from plots. Authors should also carefully plan the order of presented experiments. For instance, they introduce CHIP K30A in the Figure 1C without mentioning about it in the text. 1G - bafilomycin destabilizes CHPa, but bortezomib PDE9A no longer does. 1H there was no longer an effect of bafilomycin on CHIP. The explanation is missing.
- Concerning behavioral tests and work on animals, helpful information could be provided for instance what was the solvent for Bay 73-6691 compound. Behavioral tests could be also carried out in older animals (60 weeks) and measurement of animals survival could be included (like in Shi et al., 2018). The authors claim that administration of AAV-CHIP intravenously resulted in cerebral CHIP overexpression but they do not provide immunostaining results to validate their claim. There is not sufficient information about the procedure chosen here for a gene delivery, explanation of virus serotype and effect of the procedure on the animals whole body.
- Panels 2C-D lacks loading controls, also raising questions about the absence of PDE9A signal in lanes where it was supposedly added.
- In Figure 4 Immunoblotting results show that the effect of Bay compound on CHIP (p-CHIP) levels in the brain of rats is higher than in WT animals. Can the authors discuss whether/how they performed dose selection of the Bay 73-6691 inhibitor. The authors should also prepare such immunoblots on asymptomatic mice to learn if CHIP level is normal and stable in these animals. Unfortunately, the conclusions are not fully consistent with the results from in vitro studies (Fig. 4C), since the treatment of cells overexpressing CHIP T246M with Bay and PDE9A overexpression or silencing do not seem to change CHIP phosphorylation in the expected manner. Also, NC - not described in the figure legend.
- The authors introduce the concept of PKG regulation by PDE9A: Given the absence of a cGMP-binding domain in PDE9A, its catalytic activity is not enhanced by cGMP, leading Bay 73-6691 to elevate PKG levels by increasing baseline cGMP concentrations (Van Staveren et al, 2002). Hence, PDE9A attenuates PKG content and thus indirectly curtails the phosphorylated CHIP, truncating the half-life of CHIP and accelerating its hydrolysis. However, for the reader who is not familiar with this topic, the concept, the way it has been written, may not be easily caught.
- The fragment about mitophagy is written and explained poorly. For instance, the sentence: Thus, we endeavoured to discern

whether the U-box inactivation mutation in CHIP and impeded CHIP-mediated ubiquitination of PDE9A's K63 and K27 chains, which collectively disrupted mitophagy and reduced the dysfunctional CHIP alongside the increased PDE9A aggregation.

- The data obtained by electron microscopy should be quantitative. A description of mitochondria as recorded on images is too general. Changes observed in mitochondrial size, morphology, structure, cristae should be described in detail. Moreover, it would be helpful to measure markers specific for mitophagy and mitochondrial networking, e.g. BNIP3, OPA1, FIS1 and perform tests measuring mitochondrial activity. This experiment could be usefully accompanied by in vitro studies on cells, e.g. on primary neuronal cells with visualization of both mitochondria and mitophagy components.
- Regarding scRNA-seq the group description is again limited. Did the authors use WT or homozygous mutant mice for CHIP overexpression? It is not stated explicitly at the beginning. These scRNA-seq results sections are lengthy, not very clear and require to be in a concise format.

Referee #3:

In this manuscript Dr Hao et al investigate the role of the ubiquitin ligase CHIP in regulating PDE9A and cAMP and cGMP signaling in vitro, in cells, and in a rat model where the U-box of CHIP is mutated. Mutations in CHIP have become highly relevant in the past decade as mutations in it cause both recessive and dominantly inherited ataxia. Overall the manuscript is investigating a potentially interesting interaction between CHIP and PDE9A however the manuscript suffers from missing controls, over interpretation of data, and at some points conflicting data. In addition some key information appears to be missing/hard to interpret in the manuscript making it difficult to fully evaluate some figures.

Major Concerns

*Overall the connection between how CHIP decreases PDE9A levels and how PDE9A levels in unclear. The general claim appears to be that CHIP ubiquitinates PDE9A leading to its degradation via autophagy through an interaction mediated by HSP70. No data outside of a docking experiment (that is unconvincing as chaperones can bind many different substrates) directly demonstrates a role for HSP70. In addition data in Figure 1d directly counter a role for ubiquitination as the expression of U-box dead CHIP decreases PDE9A levels to a similar level than WT CHIP. The K30A also appears to significantly decrease PDE9A levels as well.

*In figure 1 a half life is stated but half the protein is never degraded so this claim is unsubstantiated. Also the treatments in 1H and I are unimpressive and do not always match the authors claims.

*Much of figure 2 is published work. We already know that CHIP can make all types of linkages with UbcH5 and works with UbcH5 and Ubc13. The addition of a PDE9A as a different substrate doesn't largely impact our understanding of CHIP function.

*More importantly in Figure 2 no chaperone is utilized making it unclear how CHIP would interact with PDE9A as claimed by the authors. My guess is there is little ubiquitination occurring. Blots for PDE9A and CHIP should be included, not just Ub blots which can distort the degree of activity of ubiquitin transfer.

*In Figure 2J the authors identify lysine 186 (not arginine as written in the text) is important. If ubiquitination at this residue in particular is important then all of the studies and effects observed in this manuscript (in vitro and in cells) should be repeated with at K186R mutant. This would give confidence that ubiquitination is playing a role here and provide more of a direct link.

*For all animal studies only Male animals are used. All experiments should be repeated with Female animals to identify any differences caused by sex.

*It is hard to understand what groups of animals are used. What is Bay? Is this treated wild-type animals? Moreover why are not all groups used in all experiments? The lack of some control groups make it difficult to interpret the data.

*It looks like there are no defects in motor activities between WT and mutant CHIP mice. Is the effect of Bay having anything to do with defects caused by CHIP or just generally improving function?

Minor Comments

*Figures should occur in order. For example 1J should be Figure 1A based on the text.

*n's for all experiments should be noted and proper statistical tests should be shown.

*the writing should be more concise to aid the reader in understanding what are the important parts of the manuscript.

*** As a service to authors, The EMBO Journal offers the possibility to directly transfer declined manuscripts to another EMBO Press title (EMBO Reports, EMBO Molecular Medicine, Molecular Systems Biology) or to the open access journal Life Science Alliance launched in partnership between EMBO Press, Rockefeller University Press and Cold Spring Harbor Laboratory Press. The full manuscript (including reviewer comments, where applicable and if chosen) will be automatically forwarded to the receiving journal, to allow for fast handling and a prompt decision on your manuscript. For more details of this service, and to transfer your manuscript to another EMBO title please follow this link:

Link Not Available

Dear Editor and Reviewers,

We are immensely grateful for the opportunity to clarify aspects of our study and thank you profoundly for your invaluable feedback and suggestions. These contributions have been instrumental in refining our manuscript and ensuring that the dissemination of our research is clear and accessible. They not only enhance the transparency and comprehensibility of our findings but also reinforce the scientific rigor and reliability of our work.

Following your expert advice, we have meticulously revised our manuscript, incorporating changes that include, but are not limited to, improvements in layout and design, detailed additions, and content enhancement. These modifications aim to solidify the reliability of our conclusions and ensure that our manuscript meets the high standards set by your prestigious journal, focusing on clarity, precision, and comprehensibility.

We deeply appreciate the constructive suggestions offered by each reviewer, which have significantly contributed to improving our manuscript. We have updated the manuscript accordingly, and detailed descriptions of these revisions can be found in the latest version, including in the main text and associated figures and figure legends. In this response, we address each suggestion in detail, presented in sequence and highlighted for ease of reference. Due to the extensive nature of the amendments, some of the content changes are too detailed to be fully displayed here and should be reviewed directly in the corresponding sections of the manuscript. However, we provide supplementary explanations to enhance your understanding of the modifications made.

We thank the editor once again for this opportunity to further discuss our research findings. We are pleased to highlight the significance of our study and its potential implications for future therapies, and hope the enhancements made to our manuscript will facilitate a better understanding of our work and elicit further positive feedback. We look forward to the possibility of our research being published in a renowned journal like EMBO J and eagerly anticipate your response.

Thank you once again for your guidance and consideration.

Best regards,

Changhe Shi

Address: Department of Neurology, The First Affiliated Hospital of Zhengzhou University, Zhengzhou University, 1 Jian-she east road, Zhengzhou 450000, Henan, China

E-mail: shichanghe@gmail.com

Referee #1:

This excellent study makes highly novel and extremely exciting discoveries that will impact several significant fields as well as pointing to potential novel molecular pathologies, novel therapeutic approaches and novel diagnostics.

Cyclic AMP and cyclic GMP are universal signalling molecule that underpins many critical physiological processes. Changes in cAMP/cGMP signalling underpin various pathologies and have been exploited therapeutically.

Cyclic nucleotide phosphodiesterases (PDEs) provide the sole means of degrading these two cyclic nucleotides and through their targeting to specific signalling complexes and intracellular sites, shape gradients of cAMP/cGMP in cells that underpin compartmentalisation of cAMP signalling. The plethora of PDE families generate isoforms with distinct cellular distribution, regulatory and kinetic properties as well as intracellular location conferring the ability to regulate specific physiological processes upon particular PDE isoforms.

PDE9A is a phosphodiesterase specifically hydrolyses cGMP with high affinity. This enzyme is encoded by a single gene but is expressed as more than 20 isoforms arising from alternative mRNA alternative splicing. Most studies to date have focused on its role in cognitive function, however PDE9A mRNA has been detected in cardiac and other tissues, although its wider role remains to be ascertained. Interestingly, data mining has shown that PDE9A transcripts are downregulated in various types of colorectal cancer, although the functional significance of this has yet to be defined.

In this study the authors examine CHIP, which orchestrates the degradation of aberrant proteins through proteasomal and autophagy-lysosomal pathways to maintain cellular functional integrity. The authors used exogenous expression of CHIP and PDE9A in a cell culture system, to demonstrate that CHIP overexpression decreased PDE9A protein levels in a dose-dependent manner while PDE9A over-expression decreased CHIP expression in a dose-dependent manner. They went on to demonstrate that such actions were not mediated by altered mRNA levels but, rather, at the post-translational level.

Fascinatingly, the authors showed that the TPR domain of CHIP allowed CHIP to form a complex with PDE9A that co-localised in cells. CHIP is know to promote the E2-ligase driven polyubiquitination of substrate proteins and the authors demonstrate that PDE9A is no exception to this. The functional outcome of this was shown to be polyubiquitination pf PDE9A directing it to accelerated degradation occurring through the autophagy-lysosomal pathway. These novel discoveries offer and explanation as to why animal models with CHIP mutations phenotypes exhibit upregulation of PDE9A expression. The physiological significance of this is uncovered in this study because the progressive ataxia phenotype accompanied by cerebellar neuronal degeneration and demise can be corrected by dosing with a PDE9A selective inhibitor.

Responds: Thank you very much for your recognition of our research. We appreciate your support of our current findings and your acknowledgment of their potential positive impact in the future.

1. Fig 1a. Instead of 'concentration' please simply indicated the micrograms of plasmid used. In the graphs it's not at all clear what the error bars mean as there is no indication as to whether these refer to replicate experiments or replicate scans. One would hope that these graphs described triplicate experiments. Are errors means? SD or SEM? Please define in legends.

Responds: Thank you very much for dedicating your valuable time to review our manuscript, your insightful questions have undoubtedly contributed to enhancing the quality of our work. We have made significant revisions and additions to all the experimental results initially presented in Figure 1. Consequently, the order of the graphs in Figure 1 has been altered, the previous Figure 1A is now displayed in the new Figure 1B, the specific experimental procedures and statistical methods are as follows:

Figure 1B (formerly Figure 1A): Upper: In a set of experiments involving four 10 cm diameter petri dishes for plasmid transfection, each dish was transfected with a total of 12 µg of plasmid DNA, following a co-transfection of 4 µg *Flag-CHIP* plasmid with increasing content of *V5-His-PDE9A* (0, 2, 4, 8 µg) and decreasing content of blank-vector plasmid (8, 6, 4, 0 µg) into HEK293T cells, followed by western blot analysis to examine the CHIP-mediated regulation of PDE9A levels. Lower: summary data, n = 3 biological replicates/group, 2-way ANOVA (2WANOVA) for each condition, 0 vs 2, 2 vs 4, 4 vs 8, Tukey multiple comparisons test (mct): ****P < 0.001, ***P < 0.01, **P < 0.02, P > 0.05 no marks.

Each western blot experiment was replicated three times. The number of replicates mentioned in each graph refers to experimental replicates, and there are no replicate scans. This repetition is clearly shown in the statistical graphs. All data are represented as mean ± SD, and the error bars indicate SD. We have added this information to the Figure 1 legend.

2. Fig 1 other graphs. Again, please define the number of experiments performed for analyses.

Responds: We appreciate your constructive feedback. We have revised and clearly re-described the experimental details for all images included in the manuscript. For the RT-qPCR experiments, each condition was carried out with three technical replicates and repeated six times. All the Western blot experiments presented in this manuscript were replicated three times. The number of replicates indicated in each statistical graph refers to experimental repeats, and there are no instances of replicate scans. This repetition data is clearly displayed in the statistical graphs and has been explicitly mentioned in the corresponding figure legends. For further details, please see the Figure 1 legend.

3. Fig 1L. The co-localisation is not terribly convincing at all. There are cells where it clearly doesn't occur and cells with mixtures. This is probably not surprising because, even if it is occurring, then the two proteins would have to be expressed at similar levels assuming equimolar expression. Indeed, it would be interesting to know the relative expression levels, although that would be challenging to perform. However, a more focussed analysis of the co-localisation should be attempted. Input from a laboratory specialising in such studies would help.

Responds: Thank you for your meticulous attention to the review process. Due to the low expression levels of CHIP protein in neuronal cells and the rarity of commercial antibodies for PDE9A protein immunofluorescence staining, the fluorescence signals detected via immunocytochemistry for both proteins were relatively weak. To clearly observe the specific intracellular localization of these two proteins, we referred to several publications that reported using exogenous overexpression plasmids transfected into tool cells (such as HEK293T) to observe protein colocalization in cells. We transfected HEK293T cells with equal amounts of CHIP and PDE9A exogenous plasmids and employed a two-photon confocal microscope (Carl Zeiss, Germany) to observe the colocalization of PDE9A and CHIP proteins within the cells. Observations under an oil immersion lens showed that PDE9A and CHIP proteins colocalize in the cytoplasm (we have added this information, see Figure 1H).

Figure 1H: Co-transfection of 2 μ g *PDE9A* and 2 μ g *CHIP* plasmids in HEK293T cells followed by immunofluorescence staining was observed the co-localization of PDE9A and CHIP within the cytoplasm, Imaging was performed using an oil immersion lens under multiPhoto laser scanning microscopy. Experiment replicated $\times 3$. CHIP (red), PDE9A (green), DAPI (blue).

References:

- (1) Cai, Baoshan, et al. "USP5 attenuates NLRP3 inflammasome activation by promoting autophagic degradation of NLRP3." *Autophagy* 18.5 (2022): 990-1004.
- (2) Zhang, Pengfei, et al. "Ubiquitin ligase CHIP regulates OTUD3 stability and suppresses tumour metastasis in lung cancer." *Cell Death & Differentiation* 27.11 (2020): 3177-3195.
- (3) Mu, Jingfang, et al. "SARS-CoV-2 N protein antagonizes type I interferon signaling by suppressing phosphorylation and nuclear translocation of STAT1 and STAT2." *Cell discovery* 6.1 (2020): 65.
- (4) Bin, Bum- Ho, et al. "Molecular pathogenesis of Spondylocheirodysplastic Ehlers- Danlos syndrome caused by mutant ZIP13 proteins." *EMBO molecular medicine* 6.8 (2014): 1028-1042.
- (5) Feng, Lijie, et al. "Ubiquitin ligase SYVN1/HRD1 facilitates degradation of the SERPINA1 Z variant/ α -1-antitrypsin Z variant via SQSTM1/p62-dependent selective autophagy." *Autophagy* 13.4 (2017): 686-702.

4. Fig 3. Animal numbers?

Responds: Thank you very much for your meticulous attention to review our manuscript. In the Bay 73-6691 treatment cohort, there were instances where individual rats did not comply with the behavioral assessments or were unable to complete the full course of the behavioral experiments. Consequently, we excluded these non-compliant rats from the statistical analysis. The specific number of rats used in each experiment is detailed in the corresponding figure legends.

Figure 3B&J: Tolerance to maximal velocity in the rodent rotarod test. Summary data: Bay 73-6691 treatment cohort includes WT (n = 16), Hom(6M⁺)(n = 17), Hom(6M⁺) (n = 12), and Hom(6M⁺)+Bay(+) (n = 9). The AAV-CHIP injection cohort encompasses Hom(7M⁺) (n = 8), Hom(7M⁺)+AAV-CHIP (n=8), Hom(7M⁺)+AAV-NC (n = 8) and WT(n = 8). Each rat replicated test X 3.

Figure 3C&K: Time required for the rats to traverse the balance beam in its entirety during the balance beam. Summary data: Bay 73-6691 treatment cohort includes WT (n = 15), Hom(6M⁺) (n = 11), Hom(6M⁺)(n = 14), Hom(6M⁺)+Bay(+) (n = 9). The AAV-CHIP injection cohort encompasses Hom(7M⁺) (n = 8), Hom(7M⁺)+AAV-CHIP (n=8), Hom(7M⁺)+AAV-NC (n = 8) and WT(n = 8). Each rat replicated test X 3.

Figure 3D&L: Gait analysis illustrating rodent footprints. Bay 73-6691 treatment cohort includes WT (n = 15), Hom(6M⁺) (n = 11), Hom(6M⁺)(n = 14), Hom(6M⁺)+Bay(+) (n = 9). The AAV-CHIP injection cohort encompasses Hom(7M⁺) (n = 8), Hom(7M⁺)+AAV-CHIP (n=8), Hom(7M⁺)+AAV-NC (n = 8) and WT(n = 8). Each rat replicated test X 3.

Figure 3F&N: Morris water maze trajectory depicting rat exploration patterns post-hidden platform introduction in the third quadrant. Bay 73-6691 treatment cohort includes WT (n = 15), Hom(6M⁺) (n = 11), Hom(6M⁺)(n = 11), Hom(6M⁺)+Bay(+) (n = 9). The AAV-CHIP injection cohort encompasses Hom(7M⁺) (n = 8), Hom(7M⁺)+AAV-CHIP (n=8), Hom(7M⁺)+AAV-NC (n = 8) and WT(n = 8). Each rat replicated test X 3.

(We have added this information to our manuscript; for more details, please see the Figure 3 legend on pages 29-30.)

5. Fig 4. Experiment numbers?

Responds: Thank you for your comments. Due to the precious nature and long breeding cycles of homozygous mutant male rats older than 6 weeks, in this experiment, we selected three rats per group and mixed equal volumes of left hippocampal and cerebellar tissues (1:1:1) for protein Immunoblotting (right cerebellar tissues were used for electron microscopy observation). The mixed samples were then divided into three equal parts, and proteins were extracted separately for Western blot analysis. Each protein sample was tested 2-3 times, and the average values were used for statistical analysis. (We have added this information to the Immunoblotting section of the Methods in our manuscript, page 18).

6. Fig 5. Experiment numbers?

Responds: Thank you for your thorough review of our manuscript. For the tissue immunofluorescence: We selected six rats per group, and from each rat, two sections from both sides of the brain were taken for staining after paraffin embedding. The average value from each rat was used for statistical analysis. For electron microscopy staining: Three rats per group were selected, and cerebellar tissues from the right side were used for electron microscopy imaging (left cerebellar tissues were mixed for protein analysis via Immunoblotting). (We have included this information in the manuscript; please refer to the Figure 5 legend and the Immunoblotting section in the methods on pages 18 and pages 30).

7. Fig 9. Experiment numbers?

Responds: Thank you once again for your patience and meticulous attention to our manuscript, this has undoubtedly helped to enhance the clarity of our presentation. The original Figure 9 has now been replaced with Figure 8. For the ELISA assays, cerebellar and hippocampal tissues were collected from three rats per group, each brain tissue was homogenized thoroughly, from which bean-sized samples were randomly selected and weighed. Each rat was tested in quintuplicate to derive average values, and the experiment was repeated three times. (We have added this detailed methodology to the Figure 8 legend and the ELISA section in the methods on page 18).

8. Have the authors ever attempted to define the half life of PDE9A in 'normal' cells and those overexpressing CHIP?

Responds: Thank you for your meticulous attention to detail. Regarding the experiment to measure the half-life of endogenous PDE9A in cells overexpressing CHIP, we have previously attempted this but faced challenges. Due to the low expression levels of endogenous PDE9A in CHIP-overexpressing 293T cells, as shown in Figure 1G, we were unable to measure the half-life of endogenous PDE9A protein using bafilomycin A1 (an autophagy inhibitor) or bortezomib (a proteasome inhibitor, PS-341). The concentration levels were below the detection limit for Western blot imaging, making the observation of bands challenging. However, numerous studies have demonstrated that protein degradation assays can be conducted in HEK293T cells by transfecting exogenous overexpressed proteins. Previous experimental research has also confirmed that the degradation rates of endogenous and exogenous proteins are comparable. Therefore, in this study, we utilized exogenous proteins to assess the half-lives of CHIP and PDE9A proteins (see Figure 1, page 18).

- (1) Jung, Byung-Kwon, et al. "Reduced secretion of LCN2 (lipocalin 2) from reactive astrocytes through autophagic and proteasomal regulation alleviates inflammatory stress and neuronal damage." *Autophagy* 19.8 (2023): 2296-2317.
- (2) Yang, Yinan, et al. "Innate immune and proinflammatory signals activate the Hippo pathway via a Tak1-STRIPAK-Tao axis." *Nature communications* 15.1 (2024): 145.
- (3) Baou, Maria, et al. "Role of NOXA and its ubiquitination in proteasome inhibitor-induced apoptosis in chronic lymphocytic leukemia cells." *haematologica* 95.9 (2010): 1510.

Referee #2:

The paper presents a novel investigation into the molecular interplay between CHIP (Carboxyl terminus of Hsc70-Interacting Protein) and PDE9A (Phosphodiesterase 9A), suggesting a new avenue for understanding the molecular mechanisms underlying CHIP-linked spinocerebellar ataxia. The study proposes that inhibiting PDE9A could be a viable strategy for mitigating disease symptoms, offering valuable insights into the pathology and potential therapeutic avenues for CHIP T246M-caused ataxia.

General comments

- The importance and relevance of the study are well-established, building upon previous research published in PLoS Genet by Shi et al. in 2018. The continuation of research using the mutant mouse model is commendable.

Responds: Thank you very much for affirming our past research efforts and for the encouragement and support you have provided regarding the value of our current study.

- The manuscript, however, appears to have been prepared hastily, with issues in experimental design, figure description, writing style, and organization. Meticulous attention to detail and substantial revisions are required for consideration in a prestigious journal like EMBO J.

Responds: We sincerely appreciate your recommendations, which have significantly enhanced the rigor, precision, and professionalism of our article, thereby facilitating its potential publication in prestigious journals like EMBO Journal. We have thoroughly reviewed the entire manuscript and refined the figures and their legends, categorizing the main and supplementary figures. Additionally, we have meticulously revised the text for clarity and precision and had the language professionally polished in accordance with the standards of previously published articles in the journal. Due to the substantial changes made to the manuscript, some of the content changes are too detailed to be fully displayed here and should be reviewed directly in the corresponding sections of the manuscript. We kindly request that you review the latest version of the manuscript in its entirety. (See the complete manuscript for details).

- The abstract requires revision for enhanced clarity and conciseness, ensuring the concept and rationale for researching the interplay between CHIP and PDE9A are immediately apparent. Similarly, the introduction would benefit from precision in explaining the subject matter, particularly clarifying the direct phosphorylation role of cGMP or its attribution to PKG.

Responds: Thank you very much for your meticulous review and professional suggestions, which have enabled us to enhance the logical flow of our abstract and introduction. We have rewritten the abstract of the manuscript as follows:

Abstract

The Carboxyl terminus of Hsc70-interacting protein (CHIP) plays a critical role in triaging misfolded and aberrantly aggregated proteins through a network of chaperones, co-chaperones, and various degradation mechanisms. In a preclinical rodent model of *CHIP* monogenetic mutations, we previously observed a progressive increase in the steady-state protein levels of Phosphodiesterase 9A (PDE9A). The underlying detail of PDE9A in CHIP-associated ataxia remains unclear. Herein, our findings reveal that CHIP binds PDE9A to facilitate its polyubiquitination and direct PDE9A towards autophagy-lysosomal degradation. Conversely, a dysfunctional CHIP impairs this degradation pathway, resulting in elevated cGMP hydrolysis. This impairment hinders the phosphorylation of CHIP at serine 19, reduces CHIP's stability, and diminishes the ubiquitination of protein kinase A (PKAc). An increase in cAMP levels exacerbates the situation by further diminishing the ubiquitin-mediated degradation of PDE9A. Over-aggregation of

PDE9A eventually disrupting cellular mitophagy equilibrium and inducing neuronal apoptosis. Crucially, the pharmacological inhibition of PDE9A using Bay 73-6691 or viral-mediated expression of CHIP in that models reestablishes the balance of CHIP-mediated cGMP/cAMP signalling crosstalk. These interventions confer protection against cerebellar neuropathologies, notably the dysfunction of Purkinje neuron mitophagy, providing an innovative strategy for treating CHIP-related ataxia.

(Graphical Abstract)

In the introduction, we have rewrite the text, and have also detailed the direct phosphorylation effects of cGMP-activated PKG on downstream proteins, particularly highlighting the phosphorylation modifications of CHIP protein by PKG:

PDE9A, predominantly expressed in brain tissues, exhibits high selectivity and affinity towards cGMP. Cyclic guanosine monophosphate (cGMP) represents one of the principal signalling molecules within cellular contexts, capable of activating protein kinase G (PKG) and consequently phosphorylates numerous downstream target proteins. Through its interaction with cGMP, the binding domain of PKG binds to cGMP, activates the catalytic domain of PKG, and affects the phosphorylation state of various effector molecules, thereby playing a crucial role in neurodegenerative diseases (NDDs). The catalytic site of PDE9A binds to co-factors and hydrolyzes cGMP, leading to the inactivation of PKG and inhibition of target protein phosphorylation, thereby influencing multiple cellular processes. In addition, cGMP signaling often interacts with cyclic adenosine monophosphate (cAMP) signaling to maintain intracellular signal transduction. Meanwhile, recent discoveries have highlighted that PKG directly phosphorylates CHIP at serine 19 to increase CHIP level and extend its half-life, this phosphorylation (CHIP-pS19) increases CHIP's stability and enhances its affinity for HSP70.

(For further details, please see the abstract and introduction of the lasted manuscript).

• The manuscript lacks critical information on sample sizes, including the number of repetitions for in vitro and cellular experiments and the number of animals used in behavioral tests. Detailed descriptions of experimental procedures, such as transfection methods and plasmid amounts, are insufficient for reproducibility. Figure 1, in particular, demonstrates a lack of care in explaining the rationale behind studying CHIP and PDE9A reciprocal regulation and contains mismatches between figure legends and actual figures.

Responds: Thank you for your responsible approach to the review process. We have once again thoroughly revised

the manuscript to provide additional details about the sample sizes for all experiments, enhancing the completeness of the experimental details to facilitate replication of the studies. Given the substantial amount of new experimental data added, we have meticulously verified the placement of each figure and the references to them in the text. We have also carefully checked each figure's legend, providing detailed explanations of the number of experiments, repetition frequency, and animal counts, as well as extensively describing the experimental steps and methodological details in each corresponding legend. Regarding the mutual regulation between CHIP and PDE9A proteins illustrated in Figure 1, we have conducted further experiments to clarify and strengthen our conclusions. As a result, we are including the updated version of Figure 1 and Supplementary Figure 1, which provide a more detailed explanation of the bidirectional regulatory relationship between CHIP and PDE9A proteins.

To elucidate the specific regulatory mechanisms by which a reduction in CHIP levels results in the elevation of PDE9A, our initial aim was to determine the dose-response relationship and directionality of this regulation. Moreover, we sought to verify whether the increase in PDE9A protein levels induced by CHIP is due to enhanced synthesis or reduced degradation. Using exogenous expression of CHIP and PDE9A in a cell culture system, we found that CHIP decreased PDE9A protein levels in a dose-dependent manner (Fig. 1B). Conversely, PDE9A protein levels decreased CHIP protein levels in a dose-dependent manner (Fig. 1C). The decrease in steady-state protein levels was not reflected at the mRNA level, indicating that while CHIP protein levels change in CHIP mutants-especially with T246M showing a greater reduction compared to K30A-this is accompanied by an opposite trend in PDE9A protein levels, without significant changes in *PDE9A* mRNA levels. Similarly, PDE9A protein levels do not affect CHIP mRNA levels, suggesting that mutual negative regulation between PDE9A and CHIP occurs at the post-translational level (Fig. 1D-1G, Fig. S1A-S1B). Furthermore, we also observed co-localization of CHIP and PDE9A in the cytoplasm (Fig. 1H).

To substantiate that PDE9A and CHIP proteins may regulate intracellular protein homeostasis post-translationally through mutual inhibition of their degradation pathways, we utilized HEK293T cells to exogenously express PDE9A and CHIP proteins. We observed that the half-life of CHIP is approximately six hours, which increased substantially in cells treated with the proteasome inhibitor bortezomib rather than autophagy inhibitor bafilomycin A1, indicating proteasome-dependent degradation of CHIP, aligning with previous findings that delineate proteasomal degradation of CHIP, thereby reaffirming the estimated six-hour half-life of CHIP (Naito et al, 2010). In contrast, the four-hour half-life of PDE9A was attenuated in cells treated with the autophagy inhibitor bafilomycin A1 but remains unaffected by bortezomib treatment, indicative of autophagy-mediated degradation of PDE9A (Fig. 1I). Meanwhile, CHIP increased the autophagy-mediated degradation of PDE9A, reducing its half-life. Likewise, PDE9A enhanced the proteasomal degradation of CHIP, shortening its half-life (Fig. S1C). To further confirm the negative regulation of PDE9A activity on CHIP's half-life, we treated overexpression of CHIP cells with the PDE9A inhibitor Bay73-6691, which resulted in decreased PDE9A and extension of the half-life of CHIP (Fig. 1J), thereby illustrating the reciprocal inhibitory effects between CHIP and PDE9A, with CHIP promoting autophagic degradation of PDE9A, and PDE9A inhibitors suppressing the proteasomal degradation of CHIP.

Figure 1 - Bidirectional degradation regulation between PDE9A and CHIP proteins.

A: Schematic representation of PDE9A and CHIP structures; Δ TPR signifies TPR domain of CHIP deletion, and Δ U-box indicates U-box domain of CHIP deletion. The T246M mutation at the p.T246M locus results in the inactivation of the U-box domain of CHIP, whereas the K30A mutation at the p.K30A locus leads to the inactivation of the TPR domain.

B: Upper: In a set of experiments involving four 10 cm diameter Petri dishes for plasmid transfection, each dish was transfected with a total of 12 μ g of plasmid DNA, following a co-transfection of 4 μ g *Flag-CHIP* plasmid with increasing content of *V5-His-PDE9A* (0, 2, 4, 8 μ g) and decreasing content of blank-vector plasmid (8, 6, 4, 0 μ g) into HEK293T cells, followed by western blot analysis to examine the CHIP-mediated regulation of PDE9A levels. Lower: summary data, n = 3 biological replicates/group, 2-way ANOVA (2WANOVA) for each condition, 0 vs 2, 2 vs 4, 4 vs 8, Tukey multiple comparisons test (mct): ****P < 0.001, ***P < 0.01, **P < 0.02, P > 0.05 no marks.

C: Similar methodology, Upper: Co-transfection of 4 μ g *V5-His-PDE9A* plasmid with increasing concentrations (0, 2, 4, 8 μ g) of *Flag-CHIP* and decreasing content of blank-vector plasmid (8, 6, 4, 0 μ g) into HEK293T cells, followed by western blot analysis to elucidate PDE9A-mediated alterations in CHIP. Lower: summary data, n = 3 biological replicates/group.

D: Changes in *PDE9A* mRNA levels following transfection with various *CHIP* mutant plasmids, involved transfecting each well of a six-well plate with 2 μ g of blank-vector plasmid, along with 2 μ g of each mutant *CHIP* variant into HEK293T cells; NC consisted of 2 μ g blank-vector plasmid, T246M represents homozygous mutations p.T246M site, K30A represents homozygous mutations at p.K30A site, and CHIP+Bay indicates cells overexpressing *CHIP* treated with 200 μ g/mL Bay 73-6691 (*PDE9A* inhibitor, dissolved in cell culture medium using a sonicator); sh-*CHIP* is shRNAs targeting *CHIP* mRNA expression, summary data, n = 6 biological replicates/group, 1WANOVA, Tukey mct, ^{ns}P > 0.05.

E: Changes in *PDE9A* protein levels following transfection with various *CHIP* mutant plasmids, involved transfecting each well of a six-well plate with 2 μ g of *PDE9A* plasmid, along with 2 μ g of each *CHIP* variant into HEK293T cells. Upper: gels showed the *CHIP* mRNA variation affecting subsequent *PDE9A* protein levels. Lower: summary data, n = 3 biological replicates/group, 2WANOVA, Tukey mct****P < 0.001, ***P < 0.01, **P < 0.02, *P < 0.05, ^{ns}P > 0.05.

F: Changes in *CHIP* mRNA levels following transfection with *PDE9A* mutant plasmids, co-transfection of 2 μ g various *PDE9A* mutant plasmids and 2 μ g blank-vector plasmid, *PDE9A*+Bay represents HEK293T cells overexpressing *PDE9A* treated with 200 μ g/mL Bay 73-6691; sh-*PDE9A* indicates RNAi-mediated *PDE9A* knockdown. summary data, n = 6 biological replicates/group. 1WANOVA, Tukey mct, ^{ns}P > 0.05.

G: Changes in *CHIP* protein levels following transfection with *PDE9A* mutant plasmids, co-transfection of 2 μ g various *PDE9A* mutant plasmids and 2 μ g *CHIP*

plasmid into HEK293T cells. Upper: gels show the *CHIP* mRNA variation affecting subsequent PDE9A protein levels. Lower: summary data, n = 3 biological replicates/group. ****P < 0.001, **P < 0.02, ^{ns}P > 0.05.

H: Co-transfection of 2 μ g *PDE9A* and 2 μ g *CHIP* plasmids in HEK293T cells followed by immunofluorescence staining was observed the co-localization of PDE9A and CHIP within the cytoplasm, Imaging was performed using an oil immersion lens under multiPhoto laser scanning microscopy. Experiment replicated x3. CHIP (red), PDE9A (green), DPA1 (blue).

I: Co-transfections were performed in each well of a six-well plate by introducing 2 μ g of *CHIP-HA* and 2 μ g of empty vector plasmid, or 2 μ g of *PDE9A-Flag* and 2 μ g of empty vector plasmid into HEK293T cells, followed by treatment with 50 nM bafilomycin A1 (autophagy inhibitor) or 100 nM bortezomib (proteasome inhibitor, PS-341) in with cycloheximide (CHX, inhibit protein synthesis) added at six time points (0h, 2h, 4h, 6h, 8h,10h). Upper: gel assess half-lives and degradation pathways of CHIP and PDE9A. Lower: summary data, n = 3 biological replicates/group, comparing slope difference by Analysis of Covariance (ANCOVA), ****P < 0.001.

J: In the similar methodology, followed by treatment with 200 μ g/mL Bay 73-6691, and also co-transfection of 2 μ g *CHIP* plasmid and 2 μ g *PDE9A* plasmid into HEK293T cells. Upper: gel assess impact of PDE9A and Bay 73-6691 on the degradation of CHIP, and the impact of CHIP and Bay 73-6691 on the degradation of PDE9A. Lower: summary data, ANCOVA for each condition, n = 3 biological replicates/group, ****P < 0.001, **P < 0.02.

Each summary panel are presented as mean \pm SD, summary plot (D-G) and regressions (B-C, I-J) ****P < 0.001, ***P < 0.01, **P < 0.02, *P < 0.05, ^{ns}P > 0.05.

Supplementary Figure 1: mRNA levels of cell model was detected by RT-qPCR.

A: Changes in *CHIP* mRNA following transfection with various *CHIP* mutant plasmids, involved transfecting each well of a six-well plate with 2 μ g of blank-vector plasmid, along with 2 μ g of each mutant CHIP variant into HEK293T cells; NC consisted of 2 μ g blank-vector plasmid, T246M represents homozygous mutations p.T246M site, K30A represents homozygous mutations at p.K30A site, and CHIP+Bay indicates cells overexpressing CHIP treated with 200 μ g/mL Bay 73-6691 (*PDE9A* inhibitor,dissolved in cell culture medium using a sonicator); sh-CHIP is shRNAs targeting *CHIP* mRNA expression, summary data, n = 6 biological replicates/group, 1WANOVA , Tukey mct. ****P < 0.001, **P < 0.02, *P < 0.05.

B: Changes in *PDE9A* mRNA following transfection with various *PDE9A* mutant plasmids. co-transfection of 2 μ g various *PDE9A* mutant plasmids and 2 μ g blank-vector plasmid, *PDE9A*+Bay represents HEK293T cells overexpressing *PDE9A* treated with 200 μ g/mL Bay 73-6691; *sh-PDE9A* indicates RNAi-mediated *PDE9A* knockdown. summary data, n = 6 biological replicates/group. 1WANOVA, Tukey mct, ****P < 0.001, ^{ns}P > 0.05.

C: To assess bidirectional degradation regulation between CHIP and PDE9A. Co-transfection of 2 μ g *HA-CHIP* and 2 μ g *Flag-PDE9A* plasmids into HEK293T cells, followed by treatment with 50 nM bafilomycin A1 (autophagy inhibitor) or 100 nM bortezomib (proteasome inhibitor, PS-341) in with cycloheximide (CHX, inhibit protein synthesis) added at six time points (0h, 2h, 4h, 6h, 8h,10h). Upper: gel assess half-lives and degradation pathways of CHIP and PDE9A. Lower: summary data, n = 3 biological replicates/group, comparing slope difference by Analysis of Covariance (ANCOVA), ****P < 0.001.

Each summary panel are presented as mean \pm SD, summary plot (A-B) and regressions (C) ****P < 0.001, ***P < 0.01, **P < 0.02, ^{ns}P > 0.05.

(For further details, please see the Figure 1, Supplementary Figure 1 and results on the lasted manuscript).

• The paper's clarity is undermined by unnecessary repetitions and typographical errors, e.g., misidentification of ubiquitination sites and interchangeable use of 'expression' with 'protein level'. The organization could be improved to eliminate redundancies and inaccuracies, including mislabeling of figure references and statements on ubiquitination specificity. Some missing information makes it hard to make sense of the results, for example I did not find in the text a description of animal group names (e.g. Hom (6m-), Hom (6m+)).

Responds: Thank you very much for your meticulous review and the professional suggestions you provided. Your insights have greatly enhanced the professionalism and depth of our manuscript. We have comprehensively checked the entire manuscript, revising and rephrasing the content, especially correcting any writing errors and refining the layout. We have categorized figures with similar conclusions, with some now placed in supplementary figures. Additionally, following the suggestions of the peer reviewers, we have supplemented many experiments and textual contents. Given the extensive graphical and textual revisions made, it is impractical to detail each change here. Please refer to the latest version of the full article for complete details.

Regarding the specific issues raised by the reviewers, here are the clarifications:

(1) Due to a typographical error, the ubiquitination site was incorrectly described. According to the mass spectrometry results, the correct description is: PDE9A, comprising 593 amino acids and 30 lysine residues, was analyzed using LS/MS to identify the CHIP-mediated ubiquitination site at the 186th lysine residue of PDE9A. (For further details, please see the results on the lasted manuscript).

(2) To maintain the rigor of the article and to facilitate reader understanding, we have standardized the description of protein changes throughout the text to 'protein level'. (For further details, please see the complete manuscript)

(3) We have categorized figures with similar conclusions, placing some in supplementary figures. This includes experiments in Figure 1 on the mutual regulation of protein degradation between CHIP and PDE9A, and in Figure 2, cellular ubiquitination experiments confirm that CHIP can ubiquitinate PDE9A. (For further details, please see the Figure 1, Figure 2, Supplementary Figure 1, Supplementary Figure 2 and results on the lasted manuscript).

(4) For all animal groups, detailed descriptions are provided in the figure legends related to animal experiments for clarity. Additionally, for the animal experiments added this time, we have detailed the groupings as follows:

The Bay 73-6691 treatment cohort includes Hom(6M⁺) (6.25-month-old homozygous rats harboring the p.T246M mutation), Hom(6M⁺)+Bay(+) (Bay 73-6691 solubilized in corn oil administered intraperitoneally at 2mg/kg to homozygous rats), Hom(6M⁻) (4-month-old homozygous rats with the p.T246M mutation), and WT (wild-type littermates). The selection of the Hom(6M⁻) group, presymptomatic 4-month-old homozygous rats with the p.T246M mutation, is based on our previous studys confirming that homozygous mutant rats accurately replicate the human phenotypic manifestations associated with CHIP mutations, developing overt symptoms of ataxia and memory impairments by six months of age, while behavioral changes are not significant at 4 months. Therefore, to further validate the behavioral improvement in 6-month-old homozygous mutant rats treated with Bay-736691, we compared them with 6-month-old wild-type rats and also included presymptomatic 4-month-old rats as controls, confirming that behavioral abnormalities occur in mutant rats at 6 months, consistent with previous research and justifying pre-treatment with Bay 73-6691 injections.

The AAV-CHIP injection cohort encompasses Hom(7M⁺) (homozygous rats with the p.T246M mutation), Hom(7M⁺)+AAV-CHIP (tail vein injection of 1.2×10^{11} vg/mL HBAAV2/BBB-CHIP virus to elevate CHIP protein levels in the brains of homozygous rats), Hom(7M⁺)+AAV-NC (homozygous rats injected with an equivalent dose of control empty HBAAV2/BBB-blank virus) and WT(wild-type littermates).

New groups have been added for testing in WT rats treated with Bay 73-6691: WT+Bay(+) (Bay 73-6691 treated wild-type rat) and WT (wild-type littermates).

(For further details, please see the method, results and figure legend on the last manuscript).

• While aesthetically pleasing, the figures lack comprehensive legends and context in the results section, making it challenging to understand the presented data. For instance, the behavioral tests and animal work sections need more methodological details, such as the solvent for the Bay 73-6691 compound and the immunostaining validation of CHIP overexpression.

Responds: Thank you once again for your review and suggestions. As mentioned above, we have thoroughly reorganized the manuscript to provide detailed and precise descriptions of each experimental procedure and illustration, facilitating a more comprehensive understanding of the presented data. For further details, please refer to the updated methods section and Figure legends in the latest version of the manuscript. Regarding the specific issues you've highlighted, I can provide further clarification:

(1) The solvent for the Bay 73-6691 compound: For cellular experiments, Bay 73-6691 was dissolved in cell culture medium using a sonicator. For in vivo experiments in rats, Bay 73-6691 was solubilized in corn oil and administered intraperitoneally to homozygous rats.

(2) The immunostaining validation of CHIP overexpression has been included in Supplementary Figure 4:

Supplementary Figure 4 D&E: Immunofluorescence showed the expression of CHIP into the cerebellum and hippocampus after AAV-CHIP injection into the tail vein of rats. Hom(7M⁺) (homozygous rats with the p.T246M mutation), Hom(7M⁺)+AAV-CHIP (tail vein injection of 1.2×10^{11} vg/mL HBAAV2/BBB-CHIP virus to elevate CHIP protein levels in the brains of homozygous rats) and WT(wild-type littermates). Left: Quantitative immunofluorescence of cerebellar Purkinje cell marker protein CHIP, CHIP (green), DAPI (blue). Right: summary data, n = 6 biological replicates/group, 1WANOVA, Tukey mct. ****P < 0.001, **P < 0.02, n.s.P > 0.05.

• Issues include repeated mislabeling of figure references (e.g., 2B, 2C, 2F; referring to 1J before 1A, etc.), and inaccuracies such as stating ubiquitination occurs at arginine, specifically identifying the CHIP-mediated ubiquitination site at the 186th arginine residue of PDE9A. Additionally, the paper's organization could benefit from refinement, with several pieces of data more suitably placed in supplementary material. For instance, Figures 1 and 2, which convey overlapping

information, are excessively large and could be condensed to prioritize clarity over quantity. When results are featured in a main figure, their explanation should be expanded upon, as seen with the cursory mentions of panels 1C-F.

Responds: Thank you once again for your diligent and responsible work. As we have supplemented and rearranged the figures for that section of the experiment, the figure references have been updated to the latest version and are described in sequence in the text as they appear in Figure 2.

Additionally, due to a typographical error, we have revised the description of the ubiquitination site of PDE9A based on mass spectrometry results. The correct description is: PDE9A, comprising 593 amino acids and 30 lysine residues, was analyzed using LS/MS to identify the CHIP-mediated ubiquitination site at the 186th lysine residue of PDE9A.

We have also updated and rearranged the contents of Figures 1 and 2, placing graphs with redundant content into supplementary materials. Moreover, each figure has been ensured a resolution of 1200 dpi to maintain ample clarity in every figure. The latest version of Figure 1 has been discussed previously. Here, we attach the latest version of Figure 2, with the remainder of the data placed in Supplementary Figure 2.

we initially corroborated that CHIP mediate the polyubiquitination of PDE9A (Fig. S3A-S3B). Subsequently, to identify the lysine residue requisite for ubiquitin chain assembly on PDE9A, we utilized a panel of ubiquitin mutants that allowed for single lysine residues to form chains or lacked specific lysines (Fig. 2C). The results indicated that polyubiquitin chains were revealed at K48, K63, K11, K27 and K29 lysine residues, with the K11-linked ubiquitin chain being synthesised to a lesser extent (Fig. 2D, Fig. S3C). The mutations in K63R and K27R abrogated the ubiquitination, emphasising that CHIP directly catalyses K63-and K27-linked PDE9A ubiquitination (Fig. 2E, Fig. S3D). In addition, PDE9A, comprising 593 amino acids and 30 lysine residues, was analysed using the LS/MS to identified the CHIP-mediated ubiquitination site at the 186th lysine residue of PDE9A (Fig. 2F). In addition, a comparative analysis of ubiquitination patterns between wild-type PDE9A (PDE9A) and a mutant variant (PDE9A-K186R), in the context of CHIP and various ubiquitin forms (Ub-WT, Ub-K27, Ub-K63), demonstrated that robust poly-ubiquitination of PDE9A is dependent on K186 (Fig. 2G, Fig. S3E). Concomitantly, Disruptive mutations in the TPR domain (K30A) and the T246M mutation corroborated CHIP's role in PDE9A poly-ubiquitination (Fig. 2H, Fig. S3H).

Figure 2 - CHIP directly mediates K63-and K27-linked polyubiquitination of PDE9A.

A: Co-immunoprecipitation assays to identify interacting domains between PDE9A and CHIP. Antibodies used: V5-rabbit and Flag-mouse. Experiment replicated x3.

B: Co-immunoprecipitation assays to identify interacting between PDE9A and HSP70. Antibodies used: V5-rabbit and Myc-mouse. Experiment replicated x3.

C: Schematic representation of ubiquitin variants.

D&E: In vivo ubiquitination assays validating the role of different ubiquitin chain mutants in CHIP-mediated ubiquitination of PDE9A. co-transfection of 6 µg V5-His-PDE9A plasmid, 6 µg Flag-CHIP plasmid, and 6 µg ubiquitin mutant variants into HEK293T cells by 10 cm diameter Petri dishes. Polyubiquitin chains are generated with K48, K63, K11, K27 and K29 linkages, while synthesis is impaired for K63R and K27R chains. Experiment replicated x3. Antibodies: V5-rabbit, HA-mouse, and Flag-mouse

F: LS/MS detection of ubiquitination modification at position K186 in the PDE9A sequence AEVANHLAVLEKR (indicated within the red box). Experiment replicated x2

G: Similar in vivo ubiquitination assays, co-transfection of 6 µg V5-His-PDE9A plasmid or 6 µg V5-His-PDE9A-K186R plasmid, 6 µg Flag-CHIP plasmid, and 6 µg ubiquitin mutant variants plasmid into HEK293T cells. Experiment replicated x3. Antibodies: V5-rabbit, HA-mouse, and Flag-mouse. Pired t test, ****P < 0.001, **P < 0.02.

H: Similar methodology, Similar in vivo ubiquitination assays, co-transfection of 6 µg V5-His-PDE9A plasmid, 6 µg site-specific mutation CHIP plasmid, and 6 µg HA-ub plasmid into HEK293T cells. Site-specific mutations p.T246M and p.K30A in CHIP result in impaired PDE9A ubiquitination in vivo. Experiment replicated x3.

Each summary panel are presented as mean±SD, summary plot (G), ****P < 0.001, ***P < 0.01, **P < 0.02, *P < 0.05, nsP > 0.05.

• The manuscript presents inconsistencies between in vitro and in vivo findings, particularly in the immunoblotting results and the effects of Bay 73-6691 on CHIP levels. The paper's conclusions are not always supported by the presented evidence, necessitating additional experiments or clarifications.

Responds: Thank you once again for your thorough and meticulous review. We have repeated the in vitro and in vivo experiments, and the results consistently show: In cell experiments, transfecting HEK293T cells with the CHIP T246M mutant plasmid (T246M) to construct a T246M mutant cell model, and constructing a wild-type CHIP cell model in cells transfected with CHIP plasmid, the treatment of Bay 73-6691 to the T246M mutant model cells (T246M+Bay) results in an increase in CHIP protein levels. However, the increase in CHIP protein levels in these cells still does not exceed that in wild-type CHIP cells. Conversely, in *CHIP* homozygous mutant animals treated with Bay 73-6691 (Hom+Bay), the expression of CHIP protein increases beyond the levels observed in wild-type rats. We interpret this to suggest that there are differences between the cell model and the in vivo animal model. The in vivo response of rats to Bay 73-6691 is significantly more sensitive than that of cells, leading to a substantial increase in CHIP protein reactivity in the rats. (See Supplementary Figure 5 for details).

Supplementary Figure 5A: Transfection of HEK293T cells with 4ug blank-vector plasmids (NC) or 4ug *CHIP* mutant plasmids (T246M, CHIP), or 4ug varying *PDE9A* expression plasmids (PDE9A, sh-PDE9A). T246M+Bay indicates cells transfecting T246M plasmids and treated with 200 µg/mL Bay 73-6691. Left: gels showed the CHIP levels within cellular models. Right: summary data, n = 3 biological replicates/group, 1WANOVA, Tukey mct. ****P < 0.001, **P < 0.02, *P < 0.05.

Simultaneously, we conducted additional experiments to substantiate this observation by introducing Bay 73-6691 into

normal HEK293T cells (NC+Bay(+)) and CHIP wild-type cells (CHIP+Bay(+)) to observe changes in CHIP protein levels. It was found that Bay 73-6691 increases CHIP protein levels in cells; however, compared to the increase in p-CHIP protein levels in normal wild-type rats after administration of Bay 73-6691 (WT+Bay(+)), the degree of increase in the animal is higher than in cell experiments. This further indicates that the responsiveness and sensitivity to Bay 73-6691 in live animals are stronger than in cells. Although the current differences in protein levels between in vivo and in vitro do not significantly affect the conclusions of this part, further research should be conducted in subsequent experiments to delve deeper into this phenomenon.

Supplementary Figure 5B: The phosphorylation levels of CHIP at serine 20 in cerebellar and hippocampal tissues across the Bay 73-6691 treatment wild-type rat. Upper: gels showed the CHIP and p-CHIP (phosphorylated CHIP) levels in WT+Bay(+) (Bay 73-6691 treated wild-type rat), and WT (wild-type littermates) with each lane containing a mixed tissue protein sample from three rats. Lower: summary data, n = 3 biological replicates/group, 1WANOVA, Tukey mct. ****P < 0.001, ***P < 0.01.

Supplementary Figure 5C: The levels of CHIP phosphorylation in cellular model. Transfection of HEK293T cells with 4ug blank-vector plasmids (NC) or 4ug HA-CHIP plasmids (CHIP). NC+Bay(+) indicates cells treated with 200 µg/mL Bay 73-6691, CHIP+Bay(+) indicates cells transfecting HA-CHIP plasmids and treated with 200 µg/mL Bay 73-6691. Upper: gels showed the CHIP and p-CHIP levels within cellular models. Lower: summary data, n = 3 biological replicates/group, 1WANOVA, Tukey mct. ****P < 0.001, ***P < 0.01, **P < 0.02.

(we have added this detail, see Supplementary Figure 5).

• The discussion appears disconnected from the existing literature on the subject, mainly recapitulating the study's findings without integrating them into the broader research context. A more thorough engagement with related studies and a clearer articulation of how the current work advances the field would strengthen this section.

Responds: Thank you for your valuable reminders and suggestions. We have thoroughly revised and enriched the discussion section of our manuscript. This study is grounded on the mutual negative regulation between CHIP and PDE9A, which mediates the balance of cGMP/cAMP signaling to maintain the autophagic homeostasis of cellular mitochondria. When CHIP function is anomalously mutated, PDE9A protein accumulates and produces toxic effects, disrupting this homeostasis and leading to neuronal death and apoptosis. The PDE9A inhibitor Bay 73-6691 can mitigate these toxic effects by inhibiting PDE9A, thereby ameliorating the neuronal damage caused by the loss of CHIP function. Therefore, we further discuss how, in many neurodegenerative diseases associated with CHIP, such as Alzheimer's Disease (AD), Parkinson's Disease (PD), Amyotrophic Lateral Sclerosis (ALS), and polyglutamine diseases (PolyQ diseases), this insight could potentially become a common therapeutic direction for these disorders, making the targeting of PDE9A and CHIP a promising strategy for exploring therapeutics for neurodegenerative diseases (NDDs). Early targeting of PDE9A in NDDs is a logical strategy for preventing or halting the progression of

these disorders. (We have included this information in the Discussion section, please see for details).

• The paper should provide quantitative electron microscopy data and detailed descriptions of mitochondrial morphology. Additionally, incorporating markers for mitophagy and mitochondrial networking could enrich the study's findings.

Responds: Thank you once again for your valuable suggestions, which have helped make our manuscript more comprehensive. We have performed quantitative analyses of the electron microscopy images related to mitochondria, including the Flameng score for mitochondrial injury, as well as quantitative assessments of mitochondrial perimeter and area for each group. The results have reaffirmed that mitochondrial assessment using the Flameng method revealed significantly higher scores in the *CHIP* homozygous mutant group (Hom) compared to the other three groups, accompanied by a decrease in mitochondrial perimeter and area. The introduction of Bay 73-6691 (Hom+Bay(+)) or AAV-mediated exogenous *CHIP* expression (Hom+AAV-*CHIP*) resulted in a reduction in these scores, indicating mitigated mitochondrial damage and morphological improvements in the mutation rats. This addition strengthens the findings and discussion presented in our manuscript. (For further details on these results, see Figure 5 and Supplementary Figure 6).

From left to right, the analyses include: Mitochondrial injury Flameng score, five different microscopic fields were selected for each sample, and 20 mitochondria were randomly selected for each field of view, summary data: n = 6 rat sample/group, 1WANOVA, Tukey mct. ****P < 0.001, ***P < 0.01, **P < 0.02, *P < 0.05, ^{ns}P > 0.05. Quantitation of mitochondrial morphology in Hom, Hom+Bay(+), Hom+AAV-*CHIP*, and WT groups. six mitochondria were randomly selected in each sample to measure the average circumference and area. summary data: n = 6 rat sample/group, 1WANOVA, Tukey mct: **P < 0.02, ^{ns}P > 0.05.

Additionally, we have further supplemented our investigations by assessing mitochondrial-related markers. We observed a decrease in mitochondrial inner membrane fusion protein OPA1 and mitochondrial autophagy protein BNIP3, located on the outer membrane, alongside an increase in mitochondrial fission protein FIS1 in the mutant rats (Hom). These changes imply a disruption in mitochondrial fusion function and an imbalance in mitochondrial autophagy, attributed to *CHIP*-mediated cerebellar ataxia. However, normalization of these proteins was observed following treatment with Bay 73-6691 (Hom+Bay(+)) or AAV-mediated exogenous *CHIP* expression (Hom+AAV-*CHIP*), demonstrating that enhancing *CHIP* levels and reducing PDE9A levels can maintain mitochondrial autophagy equilibrium (Fig. 5E-5F). This further enriches our research into mitophagy and mitochondrial networking.

Figure 5E&5F: Automated capillary western blot assays for the expression of autophagic proteins in the cerebellar and hippocampal tissues across Bay 73-6691 treatment cohort and AAV-*CHIP* injection cohorts, tissue protein sample of three rat mixed for each lane. Right: summary data, cerebellar tissue followed by hippocampus. n = 3 biological replicates/group, 2WANOVA, Tukey mct. ****P < 0.001, ***P < 0.01, **P < 0.02, *P < 0.05, ^{ns}P > 0.05.

(For further details on these results, see Figure 5).

• Concerning the molecular docking results shown in S2A-B, this approach seems overly simplistic. The paper does not indicate whether alternative methodologies, such as molecular dynamics simulations, were considered. In the absence of such analyses, incorporating co-immunoprecipitation experiments could provide valuable insights into the suggested interaction between PDE9A and HSP70.

Responds: Thank you sincerely for your suggestions. We have now refined the immunoprecipitation assays involving PDE9A and HSP70 proteins. The results further confirm the interaction between PDE9A and HSP70 proteins. (We have thoroughly updated this section, details of which can be seen in Figure 2)

Co-immunoprecipitation assays to identify interacting between PDE9A and HSP70. Antibodies used: V5-rabbit and Myc-mouse. Experiment replicated x 3.

• The scRNA-seq section requires clarification and condensation, particularly concerning the experimental groups used for CHIP overexpression analysis.

Responds: Thank you for your detailed suggestions. Due to the extensive amount of data analysis involved, we have revisited and refined the description of our single-cell RNA sequencing (scRNA-seq) results. Given the breadth of this content, it is impractical to describe everything here in detail. However, we will briefly introduce the content and logic of this section for a better understanding.

Using scRNA-seq, we detected that in CHIP homozygous mutant rats, the cAMP-PKA pathway in cerebellar and hippocampal neurons was activated while the cGMP-PKG signaling was suppressed. When mutant rats underwent treatment with Bay 73-6691 or when exogenous CHIP protein was introduced, the signaling disruptions caused by CHIP mutations were reversed—that is, the cAMP-PKA pathway was inhibited while the cGMP-PKG signaling was activated. This therapeutic effect was confirmed in protein immunoblotting assays of rat brain tissues: compared to wild-type rats (WT), CHIP homozygous mutation rats (Hom) exhibited increased protein levels associated with the cAMP-PKA pathway and decreased levels associated with the cGMP-PKG pathway. Post-treatment with Bay 73-6691 (Hom+Bay) or AAV-CHIP (Hom+AAV-CHIP) for CHIP homozygous mutation rats, these protein levels tended toward normal. These results suggest the involvement of the cAMP and cGMP signaling pathways in the pathogenesis of CHIP-related ataxia.

Preliminary experiments demonstrated that CHIP mutations primarily disrupt Purkinje neurons. Therefore, we specifically examined these neurons, finding that, similar to the total cell data, the cAMP-PKA pathway was activated and the cGMP-PKG signal was suppressed in the homozygous mutant group (Hom). After treatment with Bay 73-6691 (Hom+Bay(+)) or AAV-CHIP (Hom+AAV-CHIP), these pathways trended towards normal, indicating that AAV-CHIP and Bay 73-6691 mediate the crosstalk between cGMP/cAMP signaling, maintaining the balance of intracellular cGMP/cAMP signals. Further analyses in Purkinje neurons revealed abnormalities in oxidative phosphorylation, ubiquitin-mediated proteolysis, mitophagy, and autophagy in the homozygous mutant group, which were improved after treatment with Bay 73-6691 or AAV-CHIP, aligning with previous experimental findings.

Furthermore, to better observe the therapeutic effects of Bay 73-6691, we primarily selected AAV-CHIP to mediated

exogenous CHIP expression as a positive reference, facilitating a clearer understanding of the therapeutic role and mechanisms of Bay 73-6691 in treating CHIP-related ataxia.

(We have thoroughly updated and supplemented this section, see the Results for more details).

Detailed review

• Figure 1 is especially prepared without enough care. It is not sufficiently expressed why the authors decided to study a reciprocal regulation between CHIP and PDE9A. Figure legends (A, B) do not match the actual figures. Lack of detailed description of transfection in the method section does not allow to reproduce these results, but amounts of plasmids used for co-transfection seem to be very high (Is it maximum 16 μg as described in the figure legend or 8 μg , as written on the plot?). What culture dishes were used for cell seeding and transfection? How were amounts of plasmids initially selected? Plots need complete information (X axis-missed units, Y axis- better name would be for example: CHIP/PDE9A level fold change). Blots are clean and generally skillfully performed, but the Fig. 1 conclusions about PFE9A and CHIP degradation pathways are not fully convincing based on presented evidence. Also, the authors provide approximate half-life of proteins but it cannot be easily deciphered from plots. Authors should also carefully plan the order of presented experiments. For instance, they introduce CHIP K30A in the Figure 1C without mentioning about it in the text. 1G - bafilomycin destabilizes CHPa, but bortezomib PDE9A no longer does. 1H there was no longer an effect of bafilomycin on CHIP. The explanation is missing.

Responds: We sincerely thank you for your thorough and careful review, and for the valuable suggestions that have significantly improved the completeness of our article. Following your feedback, we have revised and reformatted Figure 1, adjusting some graph positions, replacing and supplementing relevant images, and re-describing these results. As we have already displayed the latest version of Figure 1 and its legend in a previous response, we will not repeat that here (these updates are thoroughly detailed in the current manuscript under Figure 1 and the results section).

For the specific issues you highlighted, here are point-by-point responses:

(1) Preliminary research confirmed that CHIP mutations could lead to decreased expression of PDE9A. In our experiments, increasing CHIP protein levels in vitro, with graded concentrations, resulted in a corresponding decrease in PDE9A levels. To ascertain if this effect was directional, we also increased PDE9A protein levels and observed a decrease in CHIP levels, confirming a mutual regulatory relationship between the two proteins;

(2) We apologize for any previous errors and have carefully checked throughout the document to ensure that all Figure legends now correctly match the actual figures.

(3) We verified our experimental records and specific procedures, using a Lipofectamine® 3000 transfection kit (Invitrogen, USA) in 10 cm Petri dishes, maintaining a total transfection reagent volume of 12 μg per dish as detailed:
Figure 1B (formerly Figure 1A): In a set of experiments involving four 10 cm diameter Petri dishes for plasmid transfection, each dish was transfected with a total of 12 μg of plasmid DNA, following a co-transfection of 4 μg *Flag-CHIP* plasmid with increasing content of *V5-His-PDE9A* (0, 2, 4, 8 μg) and decreasing content of blank-vector plasmid (8, 6, 4, 0 μg) into HEK293T cells

Figure 1C (formerly Figure 1B): Co-transfection of 4 μg *V5-His-PDE9A* plasmid with increasing concentrations (0, 2, 4, 8 μg) of *Flag-CHIP* and decreasing content of blank-vector plasmid (8, 6, 4, 0 μg) into HEK293T cells

(4) All figures have been reviewed to ensure that all plots and information are correct and complete.

(5) We repeated the degradation half-life assays for PDE9A and CHIP proteins, correcting the plasmid transfection concentrations and the total protein loaded for the experiments. Multiple repetitions allowed for observation of protein

degradation bands and preliminary estimation of half-life changes. The use of autophagy inhibitor bafilomycin A1 and proteasome inhibitor bortezomib confirmed a 6-hour half-life for CHIP through proteasomal degradation, and a 4-hour half-life for PDE9A through autophagy-mediated degradation.

Figure 11: Co-transfections were performed in each well of a six-well plate by introducing 2 μ g of *CHIP-HA* and 2 μ g of empty vector plasmid, or 2 μ g of *PDE9A-Flag* and 2 μ g of empty vector plasmid into HEK293T cells, followed by treatment with 50 nM bafilomycin A1 (autophagy inhibitor) or 100 nM bortezomib (proteasome inhibitor, PS-341) in with cycloheximide (CHX, inhibit protein synthesis) added at six time points (0h, 2h, 4h, 6h, 8h,10h). Upper: gel assess half-lives and degradation pathways of CHIP and PDE9A. Lower: summary data, n = 3 biological replicates/group, comparing slope difference by Analysis of Covariance (ANCOVA), ****P < 0.001.

It was demonstrated that CHIP protein promotes PDE9A autophagy-mediated degradation, and PDE9A promotes CHIP proteasomal degradation. (These updates are detailed in Figure 1, Supplementary Figure 1, and the results).

Supplementary Figure 1C: To assess bidirectional degradation regulation between CHIP and PDE9A. Co-transfection of 2 μ g *HA-CHIP* and 2 μ g *Flag-PDE9A* plasmids into HEK293T cells, followed by treatment with 50 nM bafilomycin A1 (autophagy inhibitor) or 100 nM bortezomib (proteasome inhibitor, PS-341) in with cycloheximide (CHX, inhibit protein synthesis) added at six time points (0h, 2h, 4h, 6h, 8h,10h). Upper: gel assess half-lives and degradation pathways of CHIP and PDE9A. Lower: summary data, n = 3 biological replicates/group, comparing slope difference by Analysis of Covariance (ANCOVA), ****P < 0.001.

(6) The manuscript layout has been redesigned to ensure that data appearing in figures are described at the corresponding locations in the text, such as the mutation K30A disrupts the binding functionality of the TPR domain to proteins such as HSP70 without affecting the integrity of the U-box domain. (All related content has been extensively updated, see Results for details).

• Concerning behavioral tests and work on animals, helpful information could be provided for instance what was the solvent for Bay 73-6691 compound. Behavioral tests could be also carried out in older animals (60 weeks) and measurement of animals survival could be included (like in Shi et al., 2018). The authors claim that administration of AAV-CHIP intravenously resulted in cerebral CHIP overexpression but they do not provide immunostaining results to validate their claim. There is not sufficient information about the procedure chosen here for a gene delivery, explanation of virus serotype and effect of the procedure on the animals whole body.

Responds: Thank you for your kind remarks and helpful review, which have significantly aided in refining our manuscript. We will address your inquiries point by point:

(1) In animal experiments, 1 ml of a 100 mg/ml Bay 73-6691 solution was dissolved in 50 ml of corn oil (Solarbio, China). (We have detailed this part in Figure 3 and the methods section of our manuscript).

(2) Due to the valuable nature of CHIP homozygous mutant rat (Hom) samples and their long breeding cycles, we currently do not have access to rats aged 60 weeks. The oldest age for homozygous mutant rats treated with Bay 73-6691 (Hom+Bay) in our experiments is 14 months. To further assess the symptomatic improvement effects of Bay 73-6691 on homozygous mutant rats, we administered Bay 73-6691 in three treatment cycles, each separated by one month. Compared to untreated homozygous rats from the same litter, rats treated with Bay 73-6691 showed increased body weight, reduced time on the balance beam, and enhanced endurance on the rotarod test. Current limitations on time have prevented us from assessing indicators such as survival rates; however, this does not significantly impact the conclusions of this experiment. Nevertheless, we will continue to monitor these data dynamically and present the results in future studies. (We have added this information, see Supplementary Figure 4 and the Results section).

Supplementary Figure 4A&B&C: Behavioral improvement in CHIP mutant rats following Bay 73-6691 treatment. The Bay 73-6691 treatment cohort includes Hom (Homozygous rats harboring the p.T246M mutation), Hom+Bay(+) (Bay 73-6691 solubilized in corn oil administered intraperitoneally at 2 mg/kg to homozygous rats), and WT (wild-type littermates). Each group consists of four rats, with Bay 73-6691 administered intraperitoneally in three cycles (15 consecutive days of injection followed by a 7-day interval), with a two-month interval between each cycle. Summary data: Body weight changes of the rats (A); Rotarod test (B); Balance beam test (C); each rat replicated test \times 3, 2WANOVA.

(3) The results have been added to Supplementary Figure 4, and we have previously displayed them in an earlier response. (For details, see Supplementary Figure 4).

The viral construct used in this study was fundamentally based on the architecture utilized in our preceding work. For the current experimental setup, the GFP tag previously incorporated in the virus was excised, while the remaining structural components were preserved. Detailed information regarding the virus and the specifics of the AAV vector construction can be found in our previously published article: "AAV/BBB-mediated gene transfer of CHIP attenuates brain injury following experimental intracerebral hemorrhage." *Translational Stroke Research* 11 (2020): 296-309". Specifically, the AAV vector sequence was constructed according to the work of Dr. Deverman and named AAV/BBB. The vector used a CMV promoter, and the connection of T2A enables CHIP and Flag to be expressed separately under the same promoter (Referenced from Deverman et al., 2016). (We have provided full details in the Methods section).

Reference: Deverman, Benjamin E., et al. "Cre-dependent selection yields AAV variants for widespread gene transfer to the adult brain." *Nature biotechnology* 34.2 (2016): 204-209.

• Panels 2C-D lacks loading controls, also raising questions about the absence of PDE9A signal in lanes where it was supposedly added.

Responds: We sincerely appreciate your thoughtful reminder. This error in our writing occurred in the figure where we mistakenly labeled "IB: Ubiquitin" as "IB: PDE9A". We have repeated the in vitro ubiquitylation assay multiple times. The assay was conducted with a 10 μ L reaction mixture containing 1 μ M PDE9A (R&D, USA), 1 μ M CHIP (R&D, USA),

0.3 μ M E1 activating enzyme (R&D, USA), 3.4 μ M E2 conjugating enzyme Ube2V-Ubc13 (R&D, USA), 250 μ M ubiquitin (R&D, USA), and a 10 \times ATP assay buffer (200 mM HEPES-KOH, 50 mM MgCl₂, 20 mM ATP, 5 mM DTT). The mixture was incubated at 37°C for 1 hour, followed by the addition of a 5 \times SDS sample buffer and subsequent boiling. Western blotting was employed for detection. We have also refined the detection signals for CHIP, PDE9A, and ubiquitin. Due to the ability of Ube2V-Ubc13 to form ubiquitin chains, the ubiquitin antibody detection results show that the mixture containing Ube2V-Ubc13-Ub can form ubiquitin chain signals. However, no PDE9A ubiquitin chain formation was detected with the PDE9A antibody. All proteins used were purified proteins, indicating that CHIP cannot directly ubiquitinate PDE9A and requires the interaction of HSP70. Given the ubiquitous presence of HSP70 in cells, we have further confirmed these findings through cellular ubiquitination assays.

In vitro ubiquitination assays; E1: ubiquitin-activating enzymes, E2: ubiquitin-conjugation enzyme (Ube2V-Ubc13), ATP: 1 \times ATP buffer. Antibodies: Ubiquitin-mouse. Experiment replicated \times 3.

• In Figure 4 Immunoblotting results show that the effect of Bay compound on CHIP (p-CHIP) levels in the brain of rats is higher than in WT animals. Can the authors discuss whether/how they performed dose selection of the Bay 73-6691 inhibitor. The authors should also prepare such immunoblots on asymptomatic mice to learn if CHIP level is normal and stable in these animals. Unfortunately, the conclusions are not fully consistent with the results from *in vitro* studies (Fig. 4C), since the treatment of cells overexpressing CHIP T246M with Bay and PDE9A overexpression or silencing do not seem to change CHIP phosphorylation in the expected manner. Also, NC - not described in the figure legend.

Responds: Thank you once again for your meticulous and dedicated approach. We will respond point-by-point to your queries to ensure a clear understanding of the relevant content.

(1) Initially, we referred to the manufacturer's instructions and various research publications to determine an appropriate concentration range of Bay 73-6691, which was established at 1.5 mg/ml/day to 3 mg/ml/day based on preliminary experiments. The treatment duration and cycle were set for 15 days. Considering the high cost of the reagent, the substantial quantity required for the experiments, and the long delivery times, combined with the outcomes of our preliminary experiments, we ultimately decided on a dosage of 2 mg/ml/day, administered continuously over 15 days.

(2) Thank you for your kind reminder. We have administered the same dosage and treatment cycle of Bay 73-6691 in age-matched normal wild-type rats and conducted immunoblots to validate our findings. The results demonstrated that after treatment with Bay 73-6691, normal wild-type rats exhibited activation of cGMP signaling molecular proteins, while cAMP signaling molecular proteins were inhibited (we have expanded on this content, see Supplementary Figure 10).

Supplementary Figure 10A: Upper: Gel evaluate cAMP-PKA signalling protein expression in the cerebellum and hippocampus tissues of WT+Bay(+) group and WT group. WT+Bay(+) (Bay 73-6691 solubilized in corn oil administered intraperitoneally at 2mg/kg to wild-type rats), and WT (wild-type littermates). Tissue protein sample of three rat mixed for each lane. Lower: summary data, n = 3 biological replicates/group, 2WANOVA, Tukey mct: ****P < 0.001.

B: Upper: Gel evaluate cGMP-PKA signalling protein expression in the cerebellum and hippocampus tissues of WT+Bay(+) group and WT group. Tissue protein sample of three rat mixed for each lane. Lower: summary data, n = 3 biological replicates/group, 2WANOVA, Tukey mct: ****P < 0.001, ***P < 0.01, **P < 0.02.

Each summary panel are presented as mean±SD, summary plot (A-B). ****P < 0.001, ***P < 0.01, **P < 0.02, nsP > 0.05.

(3) Due to additional experiments, we have reorganized the layout of Figure 4, thus the original Figure 1C has been updated to the new Supplementary Figure 5a. As previously mentioned, repeating this segment of the experiment consistently showed that in cell experiments, the introduction of Bay 73-6691 into T246M mutant model cells (T246M+Bay) led to an increase in p-CHIP levels, yet this increase did not surpass that seen in CHIP overexpression model cells (CHIP) (Supplementary Figure 5a). However, in *CHIP* homozygous mutant animals treated with Bay 73-6691(Hom+Bay(+)), CHIP levels increased beyond those observed in wild-type rats (WT). This suggests that differences exist between the cell model and the in vivo animal model, with live animals demonstrating significantly higher sensitivity to Bay 73-6691, thus leading to a substantial increase in CHIP protein reactivity in vivo (see Figure 4). In addition, to support this observation, we conducted supplementary experiments adding Bay 73-6691 to normal HEK293T cells (NC+Bay(+)) and CHIP wild-type cells(CHIP+Bay(+)) and observed an increase in p-CHIP protein levels. However, compared to the increase in normal wild-type rats treated with Bay 73-6691(WT+Bay(+)), the increase in p-CHIP protein levels in rats was greater than in the cell experiments, further indicating stronger reactivity and sensitivity to Bay 73-6691 in vivo (we have supplemented this content, see Supplementary Figure 5B-5C). These results indicate that differences in CHIP levels lead to variations in p-CHIP levels both in vitro and in vivo. While we believe that these differences in protein levels between in vivo and in vitro experiments do not significantly impact the conclusions of this part of the study, further research should continue to explore these comparative dynamics in subsequent experiments. (we have expanded on this content, see the Figure 4 and Supplementary Figure 5).

Figure 4 - PDE9A inhibits CHIP phosphorylation.

A: The phosphorylation state of CHIP at serine 20 in cerebellar and hippocampal tissues was assessed using a phospho-specific S20 antibody across the Bay 73-6691 treatment cohort. Upper: gels showed the CHIP and p-CHIP (phosphorylated CHIP) levels in Hom(6M⁺) (6.25-month-old p.T246M homozygous rat), Hom(6M⁺)+Bay(+) (Bay 73-6691 treated homozygous rat), and WT (wild-type littermates) with each lane containing a mixed tissue protein sample from

three rats. Lower: summary data, n = 3 biological replicates/group, 1WANOVA, Tukey mct. ****P < 0.001, ***P < 0.01.

B: Quantification of CHIP phosphorylation levels in cerebellum and hippocampus was conducted for the AAV-CHIP injection cohorts. Upper: gels showed the CHIP and p-CHIP levels among Hom(7M⁺) (7-month-old p.T246M homozygous rat), Hom(7M⁺)+AAV-CHIP (AAV-CHIP injected homozygous rat), Hom(7M⁺)+AAV-NC (AAV-NC injected homozygous rat), and WT (wild-type littermates), with each lane containing a mixed tissue protein sample from three rats. Lower: summary data, n = 3 biological replicates/group, 1WANOVA, Tukey mct. ****P < 0.001.

Each summary panel are presented as mean±SD, summary plot (A-B), ****P < 0.001, ***P < 0.01, **P < 0.02, *P < 0.05, ^{ns}P > 0.05.

Supplementary Figure 5a: Changes in CHIP levels to evaluate the levels of CHIP phosphorylation. Transfection of HEK293T cells with 4ug blank-vector plasmids (NC) or 4ug *CHIP* mutant plasmids (T246M, CHIP), or 4ug varying *PDE9A* expression plasmids (PDE9A, sh-PDE9A). T246M+Bay indicates cells transfecting *CHIP-T246M* plasmids and treated with 200 µg/mL Bay 73-6691. Upper: gels showed the CHIP and p-CHIP levels within cellular models. Lower: summary data, n = 3 biological replicates/group, 1WANOVA, Tukey mct. ****P < 0.001, **P < 0.02, *P < 0.05.

• The authors introduce the concept of PKG regulation by PDE9A: Given the absence of a cGMP-binding domain in PDE9A, its catalytic activity is not enhanced by cGMP, leading Bay 73-6691 to elevate PKG levels by increasing baseline cGMP concentrations (Van Staveren et al, 2002). Hence, PDE9A attenuates PKG content and thus indirectly curtails the phosphorylated CHIP, truncating the half-life of CHIP and accelerating its hydrolysis. However, for the reader who is not familiar with this topic, the concept, the way it has been written, may not be easily caught.

Responds: Thank you for your kind reminder. We have carefully reviewed the entire document and had it professionally edited to ensure that our arguments are articulated accurately and clearly, avoiding any phrases that might lead to misunderstandings. (Please see the latest version of the full manuscript).

• The fragment about mitophagy is written and explained poorly. For instance, the sentence: Thus, we endeavoured to discern whether the U-box inactivation mutation in CHIP and impeded CHIP-mediated ubiquitination of PDE9A's K63 and K27 chains, which collectively disrupted mitophagy and reduced the dysfunctional CHIP alongside the increased PDE9A aggregation.

Responds: Thank you for your suggestion. We have rewritten this section to make the content clearer and easier for readers to understand.

The reduction in CHIP levels and the concurrent increase in PDE9A levels elicit cellular toxicity, culminating in neuronal necrosis and a decrement in neuronal population. This phenomenon can be attributed to the inactivation of CHIP, which impedes the formation of K63 and K27 ubiquitin chains on PDE9A, thus diminishing autophagic-lysosomal degradation of PDE9A and fostering its abnormal accumulation within the cell. It has been reported that the concatenation of K63 and K27 ubiquitin chains primarily initiates a signaling cascade that activates mitochondrial autophagy, targeting the aberrantly aggregated substrate proteins for degradation. The equilibrium of mitochondrial autophagy plays a pivotal role in maintaining the homeostasis of intracellular proteins and mitochondria. When mitophagy becomes dysregulated, it mediates the accumulation of proteins that induce cytotoxic damage.

Interestingly, PDE9A can localize to mitochondria, where its aggregation inhibits mitochondrial respiration, leading to mitochondrial swelling and impairment of autophagic function. Additionally, CHIP plays a crucial role within mitochondria, regulating mitochondrial autophagy. CHIP maintains the normal quantity of mitochondria within the cell by activating LC3B-mediated autophagy. A decline in CHIP levels triggers mitochondrial oxidative stress, resulting in an increased number of aberrant mitochondria. Similarly, ubiquitin chains linked through K27, mediated by CHIP, are recognized by the autophagy receptor P62, triggering mitophagy and maintaining the balance of mitophagy within the cell. Thus, we endeavoured to discern whether a reduction in CHIP due to mutations and a consequent increase in neuronal PDE9A levels could lead to mitochondrial abnormalities and disrupt the equilibrium of mitophagy. Coincidentally, in *CHIP* homozygous mutation rats (Hom), heightened P62 and diminished LC3B levels were discerned in both the hippocampus and cerebellum. Nonetheless, Bay 73-6691-treated (Hom+Bay(+)) and the AAV-mediated exogenous CHIP expression (Hom+AAV-CHIP) exhibited a decrease in P62 and a surge in LC3B levels (Fig. 5C-5D). Concurrently, a decrease in mitochondrial inner membrane fusion protein OPA1 and mitochondrial autophagy protein BNIP3, located on the outer membrane, alongside an increase in mitochondrial fission protein FIS1, was noted in the *CHIP* homozygous mutant rats. These changes imply a disruption in mitochondrial fusion function and an imbalance in mitochondrial autophagy, attributed to CHIP-mediated cerebellar ataxia. However, normalization of these proteins was observed following treatment with Bay 73-6691 or AAV-CHIP, demonstrating that enhancing CHIP levels and reducing PDE9A levels can maintain mitochondrial autophagy equilibrium (Fig. 5E-5F). (We have updated and elaborated on this part; for more details, please see the results section of the manuscript).

- The data obtained by electron microscopy should be quantitative. A description of mitochondria as recorded on images is too general. Changes observed in mitochondrial size, morphology, structure, cristae should be described in detail.

Responds: Thank you once again for your suggestions. We have enhanced and revised the experimental content accordingly.

TEM analysis revealed a reduction in normal mitochondria, an increase in abnormally swollen mitochondria with compromised membrane integrity, extensive cristae disruption, matrix dissolution or vacuolization, and an increased number of autolysosomes and autophagosomes in the cerebellum of *CHIP* homozygous mutant rats (Hom). Conversely, treatment with Bay 73-6691(Hom+Bay(+)) and AAV-CHIP (Hom+AAV-CHIP) ameliorated the mitochondrial number, size variability, and membrane integrity in cerebellar tissues of mutation rat, presenting healthier mitochondria (Fig. 5G). Mitochondrial assessment using the Flameng method revealed significantly higher scores in the Hom group compared to the other three groups, with a decrease in mitochondrial perimeter and area. Treatment of Bay 73-6691 or AAV-CHIP resulted in a reduction in scores, indicating mitigated mitochondrial damage and morphological improvements of mutation rat (For detailed information, please refer to Fig. 5G, Fig. S6D, and Table S2).

We have also added specific details about the mitochondria in the legend for Figure 5G:

Figure 5G: TEM observations of mitochondrial alterations in rat cerebellar tissue. Ultrastructure of cerebellum under TEM, Hom (*CHIP* p.T246M homozygous mutant rat), Hom+Bay(+) (Bay73-6691 treated the homozygous mutant rat), Hom+AAV-CHIP (AAV-CHIP injected the homozygous mutant rat), and WT (wild-type littermates). In the Hom group, mitochondria (M, indicated by the red dashed circle) are less frequent and exhibit localised membrane blurriness and rupture, abundant cristae disintegration and dissolution and matrix vacuolisation. Autophagolysosomes (ASS, red arrows) are also visible. N: Nucleus; Go: Golgi apparatus; Lip: Lipofuscin; RER: Rough endoplasmic reticulum, AP: Autophagosomes (This information has been expanded upon in the Figure 5G legend).

Moreover, it would be helpful to measure markers specific for mitophagy and mitochondrial

networking, e.g. BNIP3, OPA1, FIS1 and perform tests measuring mitochondrial activity. This experiment could be usefully accompanied by in vitro studies on cells, e.g. on primary neuronal cells with visualization of both mitochondria and mitophagy components.

Responds: Thank you once again for your thorough and careful review. We have expanded upon the mitochondrial signal markers and found significant changes in rat models: the decrease in mitochondrial inner membrane fusion protein OPA1 and mitochondrial autophagy protein BNIP3, located on the outer membrane, alongside an increase in mitochondrial fission protein FIS1, was noted in the CHIP homozygous mutant rats (Hom). These changes imply a disruption in mitochondrial fusion function and an imbalance in mitochondrial autophagy, attributed to CHIP-mediated cerebellar ataxia. However, normalization of these proteins was observed following treatment with Bay 73-6691(Hom+Bay(+)) or AAV-CHIP (Hom+AAV-CHIP), demonstrating that enhancing CHIP levels and reducing PDE9A levels can maintain mitochondrial autophagy equilibrium. (This updated content is detailed in the Results section). Additionally, we thank you again for your suggestions. Due to time constraints and practical limitations, we are currently unable to obtain primary neurons from homozygous mutant rats. Therefore, we will present these results in subsequent related studies.

• Regarding scRNA-seq the group description is again limited. Did the authors use WT or homozygous mutant mice for CHIP overexpression? It is not stated explicitly at the beginning. These scRNA-seq results sections are lengthy, not very clear and require to be in a concise format.

Responds: Thank you sincerely for your careful review and valuable suggestions, which have undoubtedly made our manuscript more professional and rigorous. Regarding the scRNA-seq experiments, the groups assessed were as follows: Four experimental groups were assessed, Hom (n=3, *CHIP* p.T246M homozygous mutant rat), Hom+Bay(+) (n=3, Bay73-6691 treated the homozygous mutant rat), Hom+AAV-CHIP (n=3, AAV-CHIP injected the homozygous mutant rat), and WT (n=3, wild-type littermates), hippocampus and cerebellum tissues were 1:1 mixed in each rat, with three rats mixed for per group.

We have reorganized and rewritten the section on scRNA-seq results. Due to the extensive nature of this content, we will not elaborate further here, but detailed information can be found in the results section of the manuscript.

Referee #3:

In this manuscript Dr Hao et al investigate the role of the ubiquitin ligase CHIP in regulating PDE9A and cAMP and cGMP signaling in vitro, in cells, and in a rat model where the U-box of CHIP is mutated. Mutations in CHIP have become highly relevant in the past decade as mutations in it cause both recessive and dominantly inherited ataxia. Overall the manuscript is investigating a potentially interesting interaction between CHIP and PDE9A however the manuscript suffers from missing controls, over interpretation of data, and at some points conflicting data. In addition some key information appears to be missing/hard to interpret in the manuscript making it difficult to fully evaluate some figures.

Responds: Thank you for your meticulous review of our manuscript. We have thoroughly reorganized and reviewed the entire document, rephrasing any writing errors and enhancing clarity based on the expert suggestions from the reviewers. We have supplemented a significant amount of experimental data and articulated all results clearly. And we have made extensive revisions to both the text and graphics throughout the document, as well as reorganized the layout. Additionally, we have categorized figures that convey similar conclusions, with some now placed in supplementary materials. Due to the substantial changes made to the manuscript, some of the content changes are too detailed to be fully displayed here and should be reviewed directly in the corresponding sections of the manuscript. For complete information, please refer to the full text of the manuscript.

Major Concerns

*Overall the connection between how CHIP decreases PDE9A levels and how PDE9A levels in unclear. The general claim appears to be that CHIP ubiquitinates PDE9A leading to its degradation via autophagy through an interaction mediated by HSP70. No data outside of a docking experiment (that is unconvincing as chaperones can bind many different substrates) directly demonstrates a role for HSP70. In addition data in Figure 1d directly counter a role for ubiquitiation as the expression of U-box dead CHIP decreases PDE9A levels to a similar level than WT CHIP. The K30A also appears to significantly decrease PDE9A levels as well.

Responds: Thank you for your diligent and responsible work, which has contributed significantly to making our article more professional and comprehensive. We will respond point-by-point to your queries:

(1) We have refined the immunoprecipitation assays between PDE9A protein and HSP70 protein, and the results further confirm their interaction. (We have added this information, detailed in Figure 2.)

Co-immunoprecipitation assays to identify interacting between PDE9A and HSP70. Antibodies used: V5-rabbit and Myc-mouse. Experiment replicated x3.

(2) Due to additional experiments, we have reorganized the layout of Figure 1, thus the original Figure 1D has been updated to the new Figure 1E. Through multiple repeated experiments, we have verified that changes in CHIP mRNA levels do not affect PDE9A mRNA variations. However, CHIP protein levels do change in CHIP mutants—especially with the T246M mutation showing a greater reduction compared to the K30A mutation. This is accompanied by an opposite trend in PDE9A protein levels, without significant changes in PDE9A mRNA levels, suggesting that the regulation between PDE9A and CHIP occurs at the post-translational level. We have thoroughly updated and supplemented this experiment (For more details, please refer to the latest version of Figure 1, Supplementary Figure 1,

and the results section).

Figure 1D: Changes in *PDE9A* mRNA levels following transfection with various *CHIP* mutant plasmids, involved transfecting each well of a six-well plate with 2 μ g of blank-vector plasmid, along with 2 μ g of each mutant *CHIP* variant into HEK293T cells; NC consisted of 2 μ g blank-vector plasmid, T246M represents homozygous mutations p.T246M site, K30A represents homozygous mutations at p.K30A site, and CHIP+Bay indicates cells overexpressing *CHIP* treated with 200 μ g/mL Bay 73-6691 (*PDE9A* inhibitor, dissolved in cell culture medium using a sonicator); sh-*CHIP* is shRNAs targeting *CHIP* mRNA expression, summary data, $n = 6$ biological replicates/group, 1-WANOVA, Tukey mct, $^{ns}P > 0.05$.

Figure 1E: Changes in *PDE9A* protein levels following transfection with various *CHIP* mutant plasmids, involved transfecting each well of a six-well plate with 2 μ g of *PDE9A* plasmid, along with 2 μ g of each mutant *CHIP* variant into HEK293T cells. Upper: gels showed the *CHIP* mRNA variation affecting subsequent *PDE9A* protein levels. Lower: summary data, $n = 3$ biological replicates/group, 2-WANOVA, Tukey mct **** $P < 0.001$, *** $P < 0.01$, ** $P < 0.02$, * $P < 0.05$, $^{ns}P > 0.05$.

*In figure 1 a half life is stated but half the protein is never degraded so this claim is unsubstantiated. Also the treatments in 1H and I are unimpressive and do not always match the authors claims.

Responds: Thank you for your review and detailed suggestions. We have re-conducted the protein degradation half-life tests for *PDE9A* and *CHIP*, adjusting the plasmid transfection concentrations and the total protein loaded for the experiments. After multiple repetitions, we observed the degradation bands and were able to preliminarily estimate the changes in their half-lives. Using the autophagy inhibitor bafilomycin A1 and the proteasome inhibitor bortezomib, we confirmed a 6-hour half-life for *CHIP* through proteasomal degradation, and a 4-hour half-life for *PDE9A* through autophagy-mediated degradation. These revisions ensure clarity and enhance the scientific rigor of our manuscript, aligning our findings more closely with the current understanding of the complex interactions between *CHIP* and *PDE9A*. Due to extensive modifications and reformatting of Figure 1, we are now using updated numbering to describe our results. The original Figure 1G now corresponds to the new Figure 1H (we have updated this content, see Figure 1I and results).

Figure 1I: Co-transfections were performed in each well of a six-well plate by introducing 2 μ g of *CHIP*-HA and 2 μ g of empty vector plasmid, or 2 μ g of *PDE9A*-Flag and 2 μ g of empty vector plasmid into HEK293T cells, followed by treatment with 50 nM bafilomycin A1 (autophagy inhibitor) or 100 nM bortezomib (proteasome inhibitor, PS-341) in with cycloheximide (CHX, inhibit protein synthesis) added at six time points (0h, 2h, 4h, 6h, 8h, 10h). Upper: gel

assess half-lives and degradation pathways of CHIP and PDE9A. Lower: summary data, n = 3 biological replicates/group, comparing slope difference by Analysis of Covariance (ANCOVA), ****P < 0.001.

CHIP protein facilitates autophagy-mediated degradation of PDE9A, while PDE9A promotes proteasomal degradation of CHIP. The original Figure 1H now corresponds to the new Supplementary Figure 1C (we have updated this content, see Supplementary Figure 1C and results).

Supplementary Figure 1C: To assess bidirectional degradation regulation between CHIP and PDE9A. Co-transfection of 2ug *HA-CHIP* and 2ug *Flag-PDE9A* plasmids into HEK293T cells, followed by treatment with 50 nM bafilomycin A1 (autophagy inhibitor) or 100 nM bortezomib (proteasome inhibitor, PS-341) in with cycloheximide (CHX, inhibit protein synthesis) added at six time points (0h, 2h, 4h, 6h, 8h,10h). Upper: gel assess half-lives and degradation pathways of CHIP and PDE9A. Lower: summary data, n = 3 biological replicates/group, comparing slope difference by Analysis of Covariance (ANCOVA), ****P < 0.001.

To further confirm the negative regulation of PDE9A activity on CHIP's half-life, we treated overexpression of CHIP cells with the PDE9A inhibitor Bay73-6691, which resulted in decreased PDE9A and extension of the half-life of CHIP (Fig. 1J), thereby illustrating the reciprocal inhibitory effects between CHIP and PDE9A, with CHIP promoting autophagic degradation of PDE9A, and PDE9A inhibitors suppressing the proteasomal degradation of CHIP. The original Figure 1H now corresponds to the new Figure 1J (we have updated this content, see Figure 1J and results).

Figure 1J: In the similar methodology, followed by treatment with 200 µg/mL Bay 73-6691, and also co-transfection of 2 µg *CHIP* plasmid and 2ug *PDE9A* plasmid into HEK293T cells. Upper: gel assess impact of PDE9A and Bay 73-6691 on the degradation of CHIP, and the impact of CHIP and Bay 73-6691 on the degradation of PDE9A. Lower: summary data, ANCOVA for each condition, n = 3 biological replicates/group, ****P < 0.001, **P < 0.02.

*Much of figure 2 is published work. We already know that CHIP can make all types of linkages with Ubch5 and works with Ubch5 and Ub13. The addition of a PDE9A as a different substrate doesn't largely impact our understanding of CHIP function.

*More importantly in Figure 2 no chaperone is utilized making it unclear how CHIP would interact with PDE9A as claimed by the authors. My guess is there is little ubiquitination occurring. Blots for PDE9A and CHIP should be included, not just Ub blots which can distort the degree of activity of ubiquitin transfer.

Responds: Thank you very much for your professional review and suggestions, which are crucial for improving the

presentation of our research. We will address both issues in a unified response. We first sincerely appreciate your thoughtful reminder, due to our oversight, we mistakenly labeled "IB: Ubiquitin" as "IB: PDE9A". To ensure the rigor and scientific validity of our assay, we conducted multiple repetitions of the *in vitro* ubiquitylation assay. The assay was conducted with a 10 μ L reaction mixture containing 1 μ M PDE9A (R&D, USA), 1 μ M CHIP (R&D, USA), 0.3 μ M E1 activating enzyme (R&D, USA), 3.4 μ M E2 conjugating enzyme Ube2V-Ubc13 (R&D, USA), 250 μ M ubiquitin (R&D, USA), and a 10x ATP assay buffer (200 mM HEPES-KOH, 50 mM MgCl₂, 20 mM ATP, 5 mM DTT). The mixture was incubated at 37°C for 1 hour, followed by the addition of a 5x SDS sample buffer and subsequent boiling. Western blotting was employed for detection. We have also refined the detection signals for CHIP, PDE9A, and ubiquitin. Due to the ability of Ube2V-Ubc13 to form ubiquitin chains, the ubiquitin antibody detection results show that the mixture containing Ube2V-Ubc13-Ub can form ubiquitin chain signals. However, no PDE9A ubiquitin chain formation was detected with the PDE9A antibody. All proteins used were purified proteins, indicating that CHIP cannot directly ubiquitinate PDE9A and requires the interaction of HSP70. Given the ubiquitous presence of HSP70 in cells, we have further confirmed these findings through cellular ubiquitination assays.

In vitro ubiquitination assays; E1: ubiquitin-activating enzymes, E2: ubiquitin-conjugation enzyme (Ube2V-Ubc13), ATP: 1x ATP buffer. Antibodies: Ubiquitin-mouse. Experiment replicated x 3.

***In Figure 2J the authors identify lysine 186 (not arginine as written in the text) is important. If ubiquitination at this residue in particular is important then all of the studies and effects observed in this manuscript (in vitro and in cells) should be repeated with at K186R mutant. This would give confidence that ubiquitination is playing a role here and provide more of a direct link.**

Responds: Thank you again for your kind reminder. We acknowledge the error in our previous description and have corrected the information based on mass spectrometry results. The correct description is: PDE9A, comprising 593 amino acids and 30 lysine residues, was analyzed using LS/MS to identify the CHIP-mediated ubiquitination site at the 186th lysine residue of PDE9A. Additionally, we conducted *in vivo* cellular ubiquitination experiments using a PDE9A K186R mutation plasmid. The results confirmed that after the K186R site mutation in PDE9A, the level of its ubiquitination modifications decreased, indicating that CHIP ubiquitination modifies the K186 site of PDE9A. (We have added this information to the manuscript; for more details, please see Figure 2G and the results section).

Figure2G: *In vivo* ubiquitination assays. Co-transfection of 6 μ g V5-His-PDE9A plasmid or 6 μ g V5-His-PDE9A-K186R plasmid, 6 μ g Flag-CHIP plasmid, and 6 μ g ubiquitin mutant variants plasmid into HEK293T cells. Experiment replicated x 3. Antibodies: V5-rabbit, HA-mouse, and Flag-mouse. Pired t test, ****P <

***For all animal studies only Male animals are used. All experiments should be repeated with Female animals to identify any differences caused by sex.**

Responds: Thank you again for your diligent attention to detail. The rat model used in our experiments is an autosomal dominant ataxia model, which does not exhibit sex-linked traits. Therefore, it is not necessary to segregate experimental groups based on gender. In experiments with the CHIP-related autosomal ataxia model, both the expression of CHIP protein and the phenotypic symptoms of ataxia, accompanied by cognitive decline, have been shown to be independent of sex. Additionally, there are no significant differences in the onset of disease symptoms or survival times between genders. Since our previous experiments were conducted using male rat models, to maintain consistency and continuity in our research, we continue to use male rat models.

***It is hard to understand what groups of animals are used. What is Bay? Is this treated wild-type animals? Moreover why are not all groups used in all experiments? The lack of some control groups make it difficult to interpret the data.**

Responds: Thank you for your diligent efforts and detailed review, which have significantly contributed to improving the clarity and comprehensiveness of our manuscript. We appreciate the thorough revisions made to address the ambiguities in the text, including the comprehensive checks and updates to figure annotations and abbreviations to enhance understanding. Given the extensive modifications made to the document, the explanations provided here for the specific issues raised are particularly helpful, and we look forward to reviewing the full text for more detailed changes.

We have made comprehensive updates to the experimental group design and corresponding annotations in our rat experiments. Here, we describe the modifications made to the latest version of the content:

In the behavioral experiments: Eight experimental groups were assessed: The Bay 73-6691 treatment cohort includes Hom(6M⁺) (6.25-month-old homozygous rats harboring the p.T246M mutation), Hom(6M⁺)+Bay(+) (Bay 73-6691 solubilized in corn oil administered intraperitoneally at 2mg/kg to homozygous rats), Hom(6M⁻) (4-month-old homozygous rats with the p.T246M mutation), and WT (wild-type littermates). The AAV-CHIP injection cohort encompasses Hom(7M⁺) (homozygous rats with the p.T246M mutation), Hom(7M⁺)+AAV-CHIP (tail vein injection of 1.2×10^{11} vg/mL HBAAV2/BBB-CHIP virus to elevate CHIP protein levels in the brains of homozygous rats), Hom(7M⁺)+AAV-NC (homozygous rats injected with an equivalent dose of control empty HBAAV2/BBB-blank virus) and WT(wild-type littermates).

In single-cell sequencing and pathological tissue experiments: Four experimental groups were assessed: Hom (CHIP p.T246M homozygous mutant rat), Hom+Bay(+) (Bay73-6691 treated the homozygous mutant rat), Hom+AAV-CHIP (AAV-CHIP injected the homozygous mutant rat), and WT (wild-type littermates).

Due to the scarcity and lengthy breeding cycle of the homozygous mutant rats used in our study, coupled with the high demand for animals in our experiments, we initially conducted behavioral assessments and tissue protein immunoblotting across all animal groups. The results confirmed significant differences between the homozygous mutant group (Hom) and the wild-type controls (WT), with notable behavioral improvements following treatment with Bay 73-6691. The externally overexpressed AAV-CHIP in the homozygous mutant group served as a positive control, effectively demonstrating the therapeutic impact of Bay 73-6691 on the mutant groups.

Considering the large number of animals required for each experimental group and the valuable nature of our rat models, we first excluded any specimens that, due to unforeseen circumstances, could not fulfill the requirements of subsequent experiments. Additionally, mindful of the constraints related to budget and other practical considerations, we integrated the results from behavioral assessments and immunoblotting tests to guide further analyses. Without

affecting the overall conclusions of our study, we selected the four main model cell groups mentioned above for further investigations in more resource-intensive tests such as electron microscopy and single-nucleus sequencing. This selection process did not impact the overarching results or conclusions of our experiments. In future work conducted by our research group, we plan to present further findings from these parts of the study, continuing to build on the data collected and analyzed thus far.

***It looks like there are no defects in motor activities between WT and mutant CHIP mice. Is the effect of Bay having anything to do with defects caused by CHIP or just generally improving function?**

Responds: Thank you for your thorough review. We hypothesized that inhibiting PDE9A activity or decreasing PDE9A steady-state protein levels via exogenous CHIP expression could facilitate the loss of sensory-motor and cognitive function. We applied a pharmacological and genetic approach with behavior assays to test our hypothesis (Fig. 3A).

Our rat behavioral experiments were structured into two distinct parts:

Pharmacological approach: The part of the study involved homozygous mutant rats treated with Bay 73-6691, which led to significant behavioral improvements. We divided the rats into four groups: The Bay 73-6691 treatment cohort includes Hom(6M⁺) (6.25-month-old homozygous rats harboring the p.T246M mutation), Hom(6M⁺)+Bay(+) (Bay 73-6691 solubilized in corn oil administered intraperitoneally at 2mg/kg to homozygous rats), Hom(6M⁻) (presymptomatic 4-month-old homozygous rats), and WT (wild-type littermates). By six months of age, Hom(6M⁺) rats tolerate less speed on an accelerating rotarod (Fig. 3B) and take more time to traverse a balance beam (Fig. 3C), consistent with a decrease in motor coordination. Hom(6M⁺) rats treated with Bay 73-6691 performed similarly to age-matched wild-type and presymptomatic Hom(6M⁻) rats consistent with improved motor function (Fig. 3B-3C). The Hom(6M⁺) rats exhibited decreased stride lengths, rear base width, and right front-rear overlap by six months of age relative to age-matched wild-type rats. Decreases in stride length, base width, and front-rear overlap during gait analysis can indicate various underlying conditions. A shorter stride length suggests muscular or neurological issues affecting movement. At the same time, a reduced base width could point to decreased stability or balance, and less front-rear overlap might reflect alterations in coordination. The decreases in gait parameters were reversed with treatment of either Bay 73-6691 compared to age-matched wild-type or presymptomatic 4-month-old homozygous mutant rats (Fig. 3D-3E). These data suggest that Bay 73-6691 treatment approaches improve motor coordination. Additionally, memory enhancements were noted, evidenced by reduced latency and increased frequency in locating the platform in the Morris water maze test (Fig. 3F-3I), which implicate that Bay 736691 combats the clinical phenotype of ARCA by inhibiting PDE9A protein accumulation-mediated toxic effects.

Genetic approach: In the second part of our study, we employed AAV-CHIP injection in homozygous mutant rats to assess functional changes related to CHIP and to serve as a positive reference for evaluating the effects of Bay 73-6691 treatment. This cohort was divided into four groups: The AAV-CHIP injection cohort encompasses Hom(7M⁺) (homozygous rats with the p.T246M mutation), Hom(7M⁺)+AAV-CHIP (tail vein injection of 1.2×10^{11} vg/mL HBAAV2/BBB-CHIP virus to elevate CHIP protein levels in the brains of homozygous rats), Hom(7M⁺)+AAV-NC (homozygous rats injected with an equivalent dose of control empty HBAAV2/BBB-blank virus) and WT(wild-type littermates). And compared with homozygous mutant (Hom(7M⁺)), behavioural observations revealed improved balance, coordination and gait of mutation rat (Fig. 3A, Fig. 3J-3M) and a significant enhancement in memory and cognitive abilities after AAV-CHIP treatment (Fig. 3N-3Q). These findings underscore the significant ameliorative impact of CHIP on ataxic gait and cognitive memory impairment.

Figure 3 - Inhibition of PDE9A activity and exogenous supplementation of CHIP ameliorate ARCA clinical phenotype.

A: Timeline for behavioural assessments in Bay 73-6691 treatment and AAV-CHIP injection cohorts. The Bay 73-6691 treatment cohort includes Hom(6M⁺) (6.25-month-old homozygous rats harboring the p.T246M mutation), Hom(6M⁺)+Bay(+) (Bay 73-6691 solubilized in corn oil administered intraperitoneally at 2mg/kg to homozygous rats), Hom(6M) (4-month-old homozygous rats with the p.T246M mutation), and WT (wild-type littermates). The AAV-CHIP injection cohort encompasses Hom(7M⁺) (n = 8, homozygous rats with the p.T246M mutation), Hom(7M⁺)+AAV-CHIP (n=8, tail vein injection of 100 μ l of 1.2×10^{12} vg/mL HBAAV2/BBB-CHIP virus to elevate CHIP protein levels in the brains of homozygous rats), Hom(7M⁺)+AAV-NC (n = 8, homozygous rats injected with an equivalent dose of control empty HBAAV2/BBB-blank virus) and WT(n = 8, wild-type littermates).

B&J: Tolerance to maximal velocity in the rodent rotarod test. Summary data: Bay 73-6691 treatment cohort includes WT (n = 16), Hom(6M)(n = 17), Hom(6M⁺) (n = 12), and Hom(6M⁺)+Bay(+) (n = 9). Each rat replicated test X 3, 1WANOVA, Tukey mct. ****P < 0.001, ***P < 0.01, ^{ns}P > 0.05.

C&K: Time required for the rats to traverse the balance beam in its entirety during the balance beam. Summary data: Bay 73-6691 treatment cohort includes WT (n = 15), Hom(6M⁺) (n = 11), Hom(6M)(n = 14), Hom(6M⁺)+Bay(+) (n = 9). Each rat replicated test X 3, 1WANOVA, Tukey mct. ****P < 0.001, ***P < 0.01, **P < 0.02, ^{ns}P > 0.05.

D&L: Gait analysis illustrating rodent footprints, where red represents the forelimbs and blue represents the hindlimbs. Parameters measured include LFSL (Left front stride length), RFSL (Right front stride length), LRSL (Left rear stride length), RRSL (Right rear stride length), LFRO (Left front rear overlap), RFRO (Right front rear overlap), FBW (Front-base width), RBW (Rear-base width). Bay 73-6691 treatment cohort includes WT (n = 15), Hom(6M⁺) (n = 11), Hom(6M)(n = 14), Hom(6M⁺)+Bay(+) (n = 9).

E&M: Summary gait analysis data. Each rat replicated test X 3, 1WANOVA, Tukey mct****P < 0.001, ***P < 0.01, **P < 0.02, *P < 0.05, ^{ns}P > 0.05.

F&N: Morris water maze trajectory depicting rat exploration patterns post-hidden platform introduction in the third quadrant. Bay 73-6691 treatment cohort includes WT (n = 15), Hom(6M⁺) (n = 11), Hom(6M)(n = 11), Hom(6M⁺)+Bay(+) (n = 9).

G&O: Latency period for rats to locate the hidden platform in the third quadrant, monitored over a consecutive 5-day period. Summary data: 2WANOVA.

H&P: On the sixth day post-platform removal, the proportion of time the rats spent in the NE quadrant post-water entry. Summary data: 1WANOVA, Tukey mct.

I&Q: Number of platform crossings within 60 seconds and 5 minutes, respectively, post-water entry in the NE quadrant after platform removal, summary data: 1WANOVA, Tukey mct.

Each summary panel are presented as mean±SD, summary plot (B-C, E, H-K, M, P-Q) and regressions (F, O), ****P < 0.001, ***P < 0.01, **P < 0.02, *P < 0.05, ^{ns}P > 0.05.

Additionally, in the Bay 73-6691 treatment cohort, the selection of the Hom(6M⁺) group, presymptomatic 4-month-old homozygous rats with the p.T246M mutation, is based on our previous studys confirming that homozygous mutant rats accurately replicate the human phenotypic manifestations associated with CHIP mutations, developing overt symptoms of ataxia and memory impairments by six months of age, while behavioral changes are not significant at 4 months. Therefore, presymptomatic 4-month-old homozygous rats, employed as controls for behavioral alterations at age of six months. It allowed us to further validate the behavioral improvements in 6-month-old homozygous mutant rats after treatment with Bay 73-6691. By comparing these treated rats to both 6-month-old wild-type (WT) rats and the 4-month-old mutant controls, we could demonstrate that behavioral abnormalities develop in the mutant rats by six months of age. This comparison not only continues the line of inquiry established by previous research but also highlights the optimal timing for initiating Bay 73-6691 treatment, which should ideally occur before these abnormalities manifest. This group helps the understanding of the progression of behavioral anomalies in this model and underscores the effectiveness of early intervention with Bay 73-6691.

(Please refer to Figure3 and results for the detail).

Minor Comments

***Figures should occur in order. For example 1J should be Figure 1A based on the text.**

Responds: We appreciate your continued guidance and scrutiny, which are invaluable in enhancing the quality and clarity of our manuscript. We have incorporated additional data and results, thoroughly revised the content, and introduced a new layout for the figures. We have also conducted a detailed examination and organization of the entire manuscript, ensuring that the images are appropriately positioned in accordance with the corresponding sections of the text. Due to the extensive nature of these updates, please refer to the full manuscript for a comprehensive view of the revisions.

***n's for all experiments should be noted and proper statistical tests should be shown.**

Responds: Thank you for your kind reminder. We have detailed the sample sizes and the number of repetitions for

both in vivo and in vitro experiments in each figure legend. For further details, please refer to the Figure Legend section of the manuscript.

***the writing should be more concise to aid the reader in understanding what are the important parts of the manuscript.**

Responds: Thank you once again for your professional comments, which has been instrumental in enhancing the rigor and completeness of our manuscript. We have thoroughly reorganized and reviewed the text, improving the clarity and presentation of previously unclear descriptions. Furthermore, the manuscript has undergone language polishing by a professional agency to ensure precision and readability. For detailed information on these revisions, please refer to the full text of the manuscript.

Dr. Changhe Shi
The First Affiliated Hospital of Zhengzhou University
Neurology
First affiliated hospital of Zhengzhou University, Zhengzhou, Henan, China
Henan 450000
China

4th Sep 2024

Re: EMBOJ-2024-116833R-Q
E3 Ubiquitin Ligase CHIP Facilitates the cAMP and cGMP Signaling Cross-talk by Polyubiquitinating PDE9A

Dear Dr. Shi,

Thank you for submitting a new version of your manuscript on CHIP regulation of PDE9A, and my apologies for the delay in getting back to you with an editorial decision. Given the substantial extensions and modifications since the last submission, I sent it back to the original referees 2 and 3, who had raised the most significant concerns during the original evaluation. As you can see from the reports below, both reviewers consider the study much improved and also better and more clearly presented. Nevertheless, they still have some important concerns that would need to be addressed before publication. If you can adequately address these remaining issues during a final round of formal revision, we would be happy to consider the re-revised manuscript for eventual publication in The EMBO Journal.

For this final revision, please carefully answer all remaining concerns of referees 2 and 3, both in an accompanying response letter and through changes to text and figures. In particular:

- Please use the non-ubiquitinatable PDE9A-K186R mutant to assess its effect on PDE9A half-life, as requested by referee 3.
- Since both referees consider conclusions regarding HSP70 involvement not fully supported by the presented data, it would be important to strengthen this aspect with additional experiments, or to tone down these conclusions.
- Please consider reviewer 2's suggestion to further elucidate the details associated with mitophagy, which would increase the mechanistic understanding, or again make sure to tone down conclusions appropriately and to clearly explain which parts of the model are speculative.
- Both referees emphasize that the manuscript would still require extensive presentational revisions and editing. I would therefore ask you to involve detailed proof-reading, ideally by native English-speaking colleagues and/or colleagues not directly involved with the study.

Finally, please pay close attention to guidelines below for preparing, formatting, and uploading a revised manuscript (as well as to the relevant sections in our online Author Guidelines). Adhering to these instructions as closely as possible should greatly facilitate our editorial consideration at the time of resubmission.

Please do not hesitate to contact me should you have any further questions regarding the referee reports or this final revision. I look forward to receiving your revised manuscript.

Yours sincerely,

Hartmut Vodermaier

*** PLEASE NOTE: All revised manuscripts are subject to initial checks for completeness and adherence to our formatting guidelines. Revisions may be returned to the authors and delayed in their editorial re-evaluation if they fail to comply with the following requirements (see also our Guide to Authors for further information):

9) To facilitate reproducibility and cross-laboratory adoption of methodologies, please structure the Materials & Methods section as outlined in our guide to authors, including a completed Reagents and Tools Table that can be downloaded from our author guidelines as well (<https://www.embopress.org/page/journal/14602075/authorguide#structuredmethods>).

10) Digital image enhancement is acceptable practice, as long as it accurately represents the original data and conforms to community standards. If a figure has been subjected to significant electronic manipulation, this must be clearly noted in the figure legend and/or the 'Materials and Methods' section. The editors reserve the right to request original versions of figures and the original images that were used to assemble the figure. Finally, we generally encourage uploading of numerical as well as gel/blot image source data; for details see: embopress.org/page/journal/14602075/authorguide#sourcedata

At EMBO Press, we ask authors to provide source data for the main manuscript figures. Our source data coordinator will contact you to discuss which figure panels we would need source data for and will also provide you with helpful tips on how to upload and organize the files.

In the interest of ensuring the conceptual advance provided by the work, we recommend submitting a revision within 3 months (3rd Dec 2024). Please discuss the revision progress ahead of this time with the editor if you require more time to complete the revisions. Use the link below to submit your revision:

Link Not Available

Referee #2:

The majority of my previous suggestions have been addressed or discussed. Notably, the manuscript has been enhanced by providing detailed information on sample sizes, including the number of repetitions for in vitro and cellular experiments, as well as the number of animals used in behavioral tests. The description of the animal group names is now clearer, and the validation of CHIP overexpression in the rodent model has been appropriately included. However, I find the proposed model of HSP70 involvement in CHIP-mediated processing of PDE9A to be unclear and

unconvincing. The current data do not sufficiently confirm or characterize the role of HSP70 in this process. I suggest that additional experiments be conducted to expand upon this aspect of the study, or alternatively, that HSP70 be removed from the proposed model.

The section on mitophagy presents valuable insights, but the proposed mechanism requires further clarification. It remains unclear where PDE9A aggregates-whether on the surface of the mitochondria, in the cytoplasm, or elsewhere-and this uncertainty extends to the site of PDE9A ubiquitination by CHIP. Is this ubiquitination occurring within the mitochondria, followed by processing via autophagy? The manuscript should more clearly elucidate these details, as they are critical to understanding the proposed mechanism.

Lastly, I strongly recommend that the authors undertake a thorough review of the entire manuscript to enhance its clarity, writing style, and overall communication. Persistent grammatical and typographical errors detract from the quality of the work, and I have provided a list of comments on the revised manuscript that address these issues. However, this list is not exhaustive. Further revisions are necessary to ensure the manuscript meets the highest standards of scientific communication.

1. The revised abstract could be still improved for clarity and conciseness. The goal should be explicitly stated, in a simple practical way. „The underlying detail of PDE9A in CHIP-associated ataxia remains unclear.'- this sounds too vaguely. Do the Authors aim to understand the interplay between CHIP and PDE9A because they hypothesized that PDE9A upregulation contributes significantly to the aggravation of ataxia pathology induced by CHIP mutations? Was their goal to expand our understanding of the molecular mechanism of ataxia?

„Herein, our findings reveal that CHIP binds PDE9A to facilitate its polyubiquitination and direct PDE9A towards autophagy-lysosomal degradation. Conversely, a dysfunctional CHIP impairs this degradation pathway, resulting in elevated cGMP hydrolysis. This impairment hinders the phosphorylation of CHIP at serine 19, reduces CHIP's stability, and diminishes the ubiquitination of protein kinase A (PKAc). An increase in cAMP levels exacerbates the situation by further diminishing the ubiquitin-mediated degradation of PDE9A. Over-aggregation of PDE9A eventually disrupting cellular mitophagy equilibrium and inducing neuronal apoptosis."- this text is presenting rather a model of molecular events (based on the Authors' experimental evidence) than, in my view, the main findings, which could be composed more generally. Space, which is spared, could be used for better introduction of the medical problem and the disease connected to CHIP mutations as well as describing an experimental approach undertaken here to carry out this piece of research.

Moreover, in the sentence: „Over-aggregation of PDE9A eventually disrupting cellular mitophagy equilibrium and inducing neuronal apoptosis.'- the language should be corrected.

2. Introduction explains the subject matter, particularly the role of cGMP and PKG.

Sentences with required corrections (bold text)

Our previous research discovered the first causal mutation in the CHIP monogenetic mutations in a family with cerebellar ataxia, cognitive impairment, and hypogonadism led to the classification of a new autosomal recessive form of SCA, known as SCAR16.

Meanwhile, recent discoveries have highlighted that PKG directly phosphorylates CHIP at serine 19 to increase CHIP level and extend its half-life, this phosphorylation (CHIP-pS19) increases CHIP's stability and enhances its affinity for HSP70(Ranek et al., 2020). (replace with ; after half-life)

This process dampens cAMP-mediated signaling and attenuates the cAMP-PKA signaling pathway (Rinaldi et al, 2019).

3. Results

P5:

CHIP is fundamentally structured around three major domains: TPR (26-127aa), GC (128-226aa) and U-box (227-300aa), a mutation at position p.246M in CHIP results in the inactivation of its E3 ligase activity while preserving its cochaperone functionality (T246M).

Did the Authors mean CC (coiled-coil domain), then GC?

P5:

Using exogenous expression of CHIP and PDE9A in a cell culture system, we found that CHIP decreased PDE9A protein levels in a dose-dependent manner (Fig. 1B). Conversely, PDE9A protein levels decreased CHIP protein levels in a dose-dependent manner (Fig. 1C). The decrease in steady-state protein levels was not reflected at the mRNA level, indicating that while CHIP protein levels change in CHIP mutants-especially with T246M showing a greater reduction compared to K30A-this is accompanied by an opposite trend in PDE9A protein levels, without significant changes in PDE9A mRNA levels. Similarly, PDE9A protein levels do not affect CHIP mRNA levels, suggesting that mutual negative regulation between PDE9A and CHIP occurs at the post-translational level (Fig. 1D-1G, Fig. S1A-S1B)

Suggested changes in the above text:

Using exogenous expression of CHIP and PDE9A in a cell culture system, we found that CHIP decreased the level of PDE9A in a dose-dependent manner (Fig. 1B), whereas PDE9A upregulation decreased CHIP level also in a dose-dependent manner (Fig. 1C). The reduction in steady-state protein levels was not reflected at the mRNA level, suggesting that mutual negative regulation between PDE9A and CHIP occurs at the post-translational level (Fig. 1D-1G, Fig. S1A-S1B).

Moreover, K30A mutation affect ubiquitination activity of CHIP <https://www.ncbi.nlm.nih.gov/pmc/articles/PMC4638040/>, not only binding to HSPs.

The Authors write:

while CHIP protein levels change in CHIP mutants-especially with T246M showing a greater reduction compared to K30A-this is accompanied by an opposite trend in PDE9A protein levels, without significant changes in PDE9A mRNA levels.

Which figure are the Authors referring to? If Fig. 1E, this conclusion is not supported by the presented evidence.

Fig. 1B and Fig. 1C clearly present dose-dependent effects on PDE9A and CHIP levels. The only puzzle I have, concerns the Fig. 1B: why is FLAG signal apparently detected in untransfected cells and quantified?

The Authors repeatedly throughout the manuscript write about gels, instead of blots, when they describe the Western blot results.

Fig. S1C, which presents a bidirectional regulation of CHIP and PDE9A, does not match the actual figure title: The mRNA levels of cell model were detected by qRT-PCR.

Fig. 1H

legend: language corrections required

H: Co-transfection of 2ug PDE9A and 2ug CHIP plasmids in HEK293T cells followed by immunofluorescence staining was observed the co-localization of PDE9A and CHIP within the cytoplasm, Imaging was performed using an oil immersion lens under multiPhoto laser scanning microscopy. Experiment replicated \times 3. CHIP (red), PDE9A (green), DPAI (blue).

Fig.1I

There is indeed a 6-hour half-life indicated on the blots; however, the charts show a 10-hour half-life. Could you explain the reason for this discrepancy?

Fig. 2A

In the blot of the lysate, the CHIP protein lacking TPR appears to be approximately 25 kDa in size. However, the blot for the immunoprecipitation (IP) is truncated, leaving it unclear whether this variant indeed fails to interact with PDE9A.

Given that PDE9A interacts with TPR, what is the functional relevance of its association with HSP70? Is it possible that PDE9A and HSP70 bind simultaneously to different sites within TPR? Additionally, Das et al. (2022) and others demonstrated that the interaction of TPR with HSP70 inhibits CHIP activity. How do your observations align with these findings and other studies concerning the impact of HSP70 on CHIP activity within your proposed model?

Fig.S2

Not clear what is what colour-coded.

Fig. S3 :

Could Authors explain why the ubiquitination profile of PDE9A in panel A differs significantly from that in panel F, despite the assays being virtually identical, particularly regarding CHIP WT-driven ubiquitination?

Double G:

B&C&D&E&G&H&G&I

Fig. S3G Mismatch between figure legend and an actual figure

The title: CHIP-mediated poly-ubiquitination of PDE9A.

Fig. S4 :

Panels F and G exhibit a mismatch between the actual figures and the absence of corresponding legends. Additionally, these panels are not referenced or discussed in the main text.

For panel G, the quantifications do not convincingly correlate with the representative blot provided.

"Treatment with Bay 73-6691 Improves Mitophagy Dysfunction and Cell Apoptosis in CHIP Mutations

PDE9A reduces the levels of intracellular CHIP via decrements in CHIP phosphorylation through the cGMP-PKG signaling cascade. The depletion of CHIP curtails the levels of K63- and K27-linked polyubiquitination of PDE9A, inhibiting its lysosomal degradation and thus promoting intracellular PDE9A aggregation."

However, is it accurate to describe this accumulation as 'aggregation'?

P12:

In the text the Authors cite a paper by Mishra et al., 2021:

Interestingly, PDE9A can localize to mitochondria, where its aggregation inhibits mitochondrial respiration, leading to mitochondrial swelling and impairment of autophagic function(Mishra et al, 2021).

In this cited paper Authors concluded: „PDE9 localizes to mitochondria where it can suppress mitochondrial FA oxidation and respiration, and that its inhibition augments both.'. However, they don't mention any impact of PDE9A on autophagy.

P13:

A sentence with required correction:

Thus, we endeavoured to discern whether whether a reduction in CHIP due to mutations and a consequent increase in neuronal PDE9A levels could lead to mitochondrial abnormalities and disrupt the equilibrium of mitophagy.

Fig. 5E and F

On figures a name of the protein BNIP3 was incorrectly changed to BINP3.

Fig. 5G

For mitochondrial assessment the Authors use Flameng score: Mitochondrial assessment using the Flameng method revealed significantly higher scores in the Hom group compared to the other three groups, with a decrease in mitochondrial perimeter and area.

However, they neither explain the procedure in the Methods section nor cite a paper describing this method.

P14:

A sequence for correction:

We obtained four sets of scRNA-seq results, encompassing the CHIP homozygous mutation group (Hom), AAV-mediated exogenous CHIP expression group (Hom+AAV-CHIP), Bay 73-6691 treatment group (Hom+Bay(+)) and a cohort of wild rats as the control group (Wt), capturing 19,279, 25,238, 23,177 and 13,473 individual cells, respectively (Fig. 6A, Fig. S7A).

Fig. 8 and Fig. 9

Mismatch between these figures and legends

P15:

Missing data related to the measurement of Purkinje neurons:

Through cell proportion analysis, we found that the Purkinje neuron proportion in the Hom group was significantly lower than that in the Wt group. Interestingly, treatment with Bay 73-6691 and AAV-CHIP resulted in an increase in the number of Purkinje neurons of Hom group, suggesting that Bay 73-6691 and AAV-CHIP treatments could partially restore the number, thereby ameliorating the associated symptoms in CHIP mutation rats.

P16:

Moreover, PKG mediates the reduction of p-CHIP and the decrease in CHIP half-life resulting from the increased PDE9A levels. Whereas at the page 4 they write: Meanwhile, recent discoveries have highlighted that PKG directly phosphorylates CHIP at serine 19 to increase CHIP level and extend its half-life, this phosphorylation (CHIP-pS19) increases CHIP's stability and enhances its affinity for HSP70 (Ranek et al., 2020).

P16:

A wording should be more precise (e.g. on CHIP homozygous mutants)

We, therefore, also performed scRNA-seq on the CHIP homozygous mutations treated with Bay 73-6691

P16:

In addition, to provide a comprehensive assessment of the effects of AAV-NC and Bay 73-6691 treatment on CHIP mutations animal model, we used a heatmap to highlight the prominent changes in gene expression in pathways such as cAMP signaling, cGMP-PKG signaling, oxidative phosphorylation, ubiquitin-mediated proteolysis, mitophagy and autophagy.

Should be replaced with AAV-CHIP

P17:

Heatmap is shown on Fig. S9 (not Fig. S8).

Fig. S10

Regulation of cAMP-cGAMP signaling pathway by PDE9A inhibitors. (inhibitor)

B: Upper: Gel evaluate cGMP-PKA signalling protein expression in the cerebellum and hippocampus tissues of WT+Bay(+) group and WT group. Tissue protein sample of three rat mixed for each lane.

Each summary panel are presented as mean{plus minus}SD

Suggested correction: Blot evaluates PRKG1 and PRKG2 levels in the cerebellum and hippocampus of Bay 73-6691-treated rats (WT+Bay(+)) relative to wild-type rats. Tissue protein samples of three rats were mixed for each lane.

P17:

The scRNA-seq results not only support the notion that CHIP mutations lead to the abnormal (reduced?) formation of K63 and K27 polyubiquitin chains on PDE9A, impacting PDE9A's degradation in autophagolysosomes, but also indicate that PKG mediates the reduction of p-CHIP and the shortened half-life of CHIP resulting from increased PDE9A levels.

Referee #3:

This revised submission investigating the interplay between CHIP and PDE9A and the dysregulation of this in SCA is much improved from its initial submission. Overall the clarity of the manuscript has been greatly improved. Additional controls also have improved the manuscript. Overall while much improved there are still some areas that need tightening up and some conclusions should be tempered.

Major Concern

1) The PDE9A K186R mutant is an excellent control that now alleviates the concerns that the ubiquitin detected was not due to co-immunoprecipitated proteins. This now provides the authors with a reagent to directly test the hypothesis that CHIP mediated ubiquitination of PDE9A is targeting it for degradation. Is the steady state levels of the K186R mutant increased or does having this mutation increase the half-life of PDE9A in cells?

Minor Concerns

1) In figure 2 the authors clearly demonstrate that the CHIP TPR domain is needed for both the interaction with PDE9A and with HSP70. They then conclude that PDE9A is interacting with CHIP and that this is facilitated via HSP proteins. The data presented do not support this conclusion as there is no evidence the three proteins exist in a complex. While this is the most likely explanation, this conclusion is not supported by the data.

2) In Figure 2a no delta TPR CHIP is present in the IP blot (appears to be cropped out?).

3) There is inconsistent referencing to figures throughout. Some interpretations are made but no corresponding figure is referenced.

4) "lysine residue requisite of ubiquitin chain assembly on PDE9A" in reference to figure 2. That is not what is tested here, instead ubiquitin chain linkages are being tested.

5) When discussing figure 8 of the text the authors reference figure 9.

6) There are still a number of grammatical errors and typos that should be corrected.

Dear Dr. Hartmut Vodermaier,

Thank you very much again for your valuable feedback and for giving us the opportunity to revise our manuscript titled "***E3 Ubiquitin Ligase CHIP Facilitates the cAMP and cGMP Signaling Cross-talk by Polyubiquitinating PDE9A***" (manuscript number: EMBOJ-2024-116833R-Q). We sincerely appreciate the time and effort you, as well as the reviewers, have invested in evaluating our work. We are especially grateful for the your positive recognition of the improvements made in our manuscript and their insightful suggestions, which have significantly enhanced its quality and scientific rigor.

1. Use of the non-ubiquitinatable PDE9A-K186R mutant

We have conducted additional experiments to assess the effect of the PDE9A-K186R mutant on PDE9A half-life. The results show that Mutation of PDE9A at K186R disrupted CHIP-mediated ubiquitination, resulting in an increased half-life and overaccumulation of PDE9A in cells. These results are now included in Supplementary Figure 3F and have been integrated into the revised manuscript. We are very grateful for this constructive suggestion, as it has allowed us to strengthen our conclusions and provide a more comprehensive understanding of the underlying mechanisms.

2. Conclusions regarding HSP70 involvement

We have performed additional experiments, which are now included in Supplementary Figure 2C-2D. Our findings show that HSP70 colocalises with PDE9A and the TPR domain of CHIP in the cytoplasm, indicating the potential formation of a complex between PDE9A, HSP70, and the TPR domain of CHIP; this complex likely promotes the recognition and binding of PDE9A by CHIP. Additionally, after thoroughly considering the suggestions from the editor and reviewers, and reviewing relevant literature and our current results, we believe that the possible role of HSP70 does not significantly impact the core conclusions of our study. Therefore, we have appropriately toned down this conclusion in the text. We sincerely appreciate this valuable feedback, which has helped us present a more balanced and scientifically supported narrative, making our conclusions more rigorous.

3. Further elucidation of mitophagy mechanisms

We have also elucidated the mechanism of mitophagy in more detail: Dysregulation of mitophagy results in the accumulation of cytotoxic proteins. Notably, PDE9A can localize to mitochondria, where it regulates mitophagy via the second messenger cGMP. cGMP promotes autophagosome formation and the initiation of mitophagy through pathways involving PKG signalling, oxidative stress responses, and BNIP3. However, overaccumulation of PDE9A decreases cGMP levels, inhibiting mitochondrial respiration and leading to mitochondrial swelling and impaired mitophagy function. The related expressions have been expanded in the Results section. This suggestion has greatly enriched the mechanistic understanding of our study, and we are thankful for the reviewer's thoughtful recommendation.

4. Extensive revisions for language and presentation

We acknowledge the reviewers' and your emphasis on the need for extensive revisions to improve the manuscript's language and presentation. We are grateful for this recommendation, and we have had the manuscript thoroughly reviewed and edited by native English-speaking colleagues. Additionally, we have had the entire manuscript comprehensively edited by AJE, which has substantially improved the quality of the manuscript. We are confident that the text has been significantly enhanced, and we are submitting the highest quality version of the manuscript.

Once again, we would like to express our heartfelt thanks to the reviewers and the editorial team for their positive recognition of our work and their invaluable suggestions. Their feedback has been instrumental in refining and improving the manuscript, and we believe the revised version is much stronger as a result. We hope that the revisions we have made address all the remaining concerns and that the manuscript will eventually be published in EMBO Journal.

We look forward to your favorable consideration of our revised manuscript.

Yours sincerely,

Changhe Shi

Address: Department of Neurology, The First Affiliated Hospital of Zhengzhou University, Zhengzhou University, 1 Jian-she east road, Zhengzhou 450000, Henan, China

mail: shichanghe@gmail.com

Referee #2:

The majority of my previous suggestions have been addressed or discussed. Notably, the manuscript has been enhanced by providing detailed information on sample sizes, including the number of repetitions for in vitro and cellular experiments, as well as the number of animals used in behavioral tests. The description of the animal group names is now clearer, and the validation of CHIP overexpression in the rodent model has been appropriately included.

However, I find the proposed model of HSP70 involvement in CHIP-mediated processing of PDE9A to be unclear and unconvincing. The current data do not sufficiently confirm or characterize the role of HSP70 in this process. I suggest that additional experiments be conducted to expand upon this aspect of the study, or alternatively, that HSP70 be removed from the proposed model.

The section on mitophagy presents valuable insights, but the proposed mechanism requires further clarification. It remains unclear where PDE9A aggregates-whether on the surface of the mitochondria, in the cytoplasm, or elsewhere-and this uncertainty extends to the site of PDE9A ubiquitination by CHIP. Is this ubiquitination occurring within the mitochondria, followed by processing via autophagy? The manuscript should more clearly elucidate these details, as they are critical to understanding the proposed mechanism.

Lastly, I strongly recommend that the authors undertake a thorough review of the entire manuscript to enhance its clarity, writing style, and overall communication. Persistent grammatical and typographical errors detract from the quality of the work, and I have provided a list of comments on the revised manuscript that address these issues. However, this list is not exhaustive. Further revisions are necessary to ensure the manuscript meets the highest standards of scientific communication.

Responds: We are deeply grateful for your continued review of our manuscript and for your valuable feedback. We sincerely appreciate the time and effort you have dedicated to helping us enhance the quality and readability of our paper. Your professional suggestions and insights have been instrumental in guiding the revisions and improvements of our manuscript. Additionally, we would like to express our highest respect for your meticulous and rigorous review process. In accordance with your guidance, we have addressed the issues raised and provided detailed responses to each point.

1. We are especially thankful for the constructive guidance provided by the reviewers. Your comments have prompted us to further explore the relationship between HSP70, PDE9A, and CHIP, expanding our understanding of these interactions. Particularly, we acknowledge that the interactions between these molecules, as currently described in the literature, remain inconclusive. Your insights have led us to realize that the role of HSP70 in the interaction between PDE9A and TPR, while speculative in our study, requires more detailed investigation in future research. In this context, we hypothesize that HSP70 may potentially play a role in mediating CHIP's binding to PDE9A. We further detected endogenous HSP70 protein signals during the interaction between PDE9A and the TPR domain, indicating the possibility of a tripartite interaction between the TPR domain, PDE9A, and HSP70. Additionally, co-localization of HSP70 and PDE9A within the cytoplasm suggests the formation of a complex involving PDE9A, HSP70, and the TPR domain of CHIP, which may facilitate CHIP's recognition and binding to PDE9A.

However, after careful consideration of the reviewer's suggestions, and upon reviewing the relevant literature and our own findings, we believe that the potential role of HSP70 does not significantly alter the core conclusions of our study. Therefore, we have downplayed this aspect in the main text and relocated the relevant details to the Supplementary File (revised content can be found on P6-P7, Appendix Figure S2C-S2D, highlighted in yellow).

Appendix Figure S2: Prediction model of molecular docking of PDE9A and HSP70.

A: β -SBD of HSP70 binds to PDE9A, NBD: N-terminal nucleotide-binding domain; β -SBD: β -sandwich C-terminal substrate-binding domain; α -SBD: α -helical lid C-terminal substrate-binding domain.

B: Top one prediction of the hydrogen bond interface of β -SBD and PDE9A. The CYS 338 site of PDE9A and the TYR 183 site of HSP70 are connected by hydrogen bonding. GRAMM-X software (<http://vakser.bioinformatics.ku.edu/resources/gramm/grammx>) predicted possible models. Visualisation of the prediction in PDBePISA and dockeasy online (www.dockeasy.cn).

C: Co-immunoprecipitation assays to identify interacting domains between PDE9A and CHIP and endogenous HSP70. Antibodies used: V5-rabbit, Flag-mouse and HSP70-mouse. Experiment replicated \times 3.

D: Co-transfection of 2 μ g of *PDE9A-Flag* and 2 μ g of *HSP70-Myc* plasmids in HEK293T cells followed by immunofluorescence staining was observed the co-localization of PDE9A and HSP70 within the cytoplasm, Imaging was performed through multiPhoto laser scanning microscopy. Experiment replicated \times 3. HSP70 (red), PDE9A (green), DPAI (blue).

Summary plot (C).

2. Currently, there is limited research on the function of PDE9A in mitochondria. Based on the results of our study, we propose that the ubiquitination of PDE9A by CHIP is a ubiquitous process within cells, occurring both in the cytoplasm and on the outer mitochondrial membrane. In the cytoplasm, mutations in CHIP disrupt the normal ubiquitination of PDE9A, resulting in an accumulation of PDE9A that can be degraded through the autophagy-lysosomal pathway. In contrast, abnormal ubiquitination of PDE9A on mitochondria leads to an increase in PDE9A levels, which, in turn, causes mitochondrial damage. This triggers mitophagy, where the damaged mitochondria and excess PDE9A are transported to lysosomes for degradation. However, when CHIP is mutated, the accumulation of PDE9A, both in the cytoplasm and on mitochondria, reduces the degradation of the second messenger, cGMP.

cGMP plays multiple roles in promoting the formation of autophagosomes and the initiation of mitophagy: (1) cGMP activates oxidative stress signaling, accelerating mitochondrial dysfunction and inducing mitophagy; (2) cGMP activates the PKG signaling pathway, which regulates key autophagy proteins to initiate mitophagy; (3) cGMP influences BNIP3 signaling activity, thereby modulating the occurrence of mitophagy.

When cGMP levels decrease, there is impairment in the PKG and oxidative stress signaling pathways, hindering the normal process of mitophagy. As a result, damaged mitochondria cannot be efficiently cleared via autophagy, leading to further accumulation and exacerbation of mitochondrial dysfunction. Additionally, the reduction in cGMP weakens the activity of mitophagy-related pathways, such as BNIP3, thereby reducing the efficiency of mitophagy and

resulting in the accumulation of damaged mitochondria, further aggravating mitochondrial structural damage. Each of these disruptions can contribute to an imbalance in mitophagy. (We have added this detail, see P11 and P17, highlighted in yellow).

3. We have engaged a native English-speaking expert to thoroughly revise our manuscript, ensuring the fluency, accuracy, and consistency of the writing. Additionally, we have subjected the final manuscript to AJE for language editing. We are confident that these changes will significantly enhance the quality of our paper (see the full text).

The revised abstract could be still improved for clarity and conciseness. The goal should be explicitly stated, in a simple practical way. „The underlying detail of PDE9A in CHIP-associated ataxia remains unclear.'- this sounds too vaguely. Do the Authors aim to understand the interplay between CHIP and PDE9A because they hypothesized that PDE9A upregulation contributes significantly to the aggravation of ataxia pathology induced by CHIP mutations? Was their goal to expand our understanding of the molecular mechanism of ataxia? ‚Herein, our findings reveal that CHIP binds PDE9A to facilitate its polyubiquitination and direct PDE9A towards autophagy-lysosomal degradation. Conversely, a dysfunctional CHIP impairs this degradation pathway, resulting in elevated cGMP hydrolysis. This impairment hinders the phosphorylation of CHIP at serine 19, reduces CHIP's stability, and diminishes the ubiquitination of protein kinase A (PKAc). An increase in cAMP levels exacerbates the situation by further diminishing the ubiquitin-mediated degradation of PDE9A. Over-aggregation of PDE9A eventually disrupting cellular mitophagy equilibrium and inducing neuronal apoptosis.'- this text is presenting rather a model of molecular events (based on the Authors' experimental evidence) than, in my view, the main findings, which could be composed more generally. Space, which is spared, could be used for better introduction of the medical problem and the disease connected to CHIP mutations as well as describing an experimental approach undertaken here to carry out this piece of research.

Moreover, in the sentence: „Over-aggregation of PDE9A eventually disrupting cellular mitophagy equilibrium and inducing neuronal apoptosis.'- the language should be corrected.

Responds: Thank you very much for your professional suggestions. We have revised the abstract accordingly, as detailed in the abstract section.

Abstract

The carboxyl terminus of Hsc70-interacting protein (CHIP) is pivotal for managing misfolded and aggregated proteins via chaperone networks and degradation pathways. In a preclinical rodent model of CHIP-related ataxia, we

observed that CHIP mutations lead to increased levels of phosphodiesterase 9A (PDE9A), whose role in this context remains poorly understood. Here, we investigated the molecular mechanisms underlying the role of PDE9A in CHIP-related ataxia and demonstrated that CHIP binds to PDE9A, facilitating its polyubiquitination and autophagic degradation. Conversely, dysfunctional CHIP disrupts this process, resulting in PDE9A accumulation, increased cGMP hydrolysis, and impaired PKG phosphorylation of CHIP at serine 19. This cascade further amplifies PDE9A accumulation, ultimately disrupting mitophagy and triggering neuronal apoptosis. Elevated PKA levels inhibit PDE9A degradation, further exacerbating this neuronal dysfunction. Notably, pharmacological inhibition of PDE9A via Bay 73-6691 or virus-mediated CHIP expression restored the balance of cGMP/cAMP signalling. These interventions protect against cerebellar neuropathologies, particularly Purkinje neuron mitophagy dysfunction. Thus, PDE9A upregulation considerably exacerbates ataxia associated with CHIP mutations, and targeting the interaction between PDE9A and CHIP is an innovative therapeutic strategy for CHIP-related ataxia.

Introduction explains the subject matter, particularly the role of cGMP and PKG.

Sentences with required corrections (bold text)

Our previous research discovered the first causal mutation in the CHIP monogenetic mutations in a family with cerebellar ataxia, cognitive impairment, and hypogonadism led to the classification of a new autosomal recessive form of SCA, known as SCAR16.

Meanwhile, recent discoveries have highlighted that PKG directly phosphorylates CHIP at serine 19 to increase CHIP level and extend its half-life, this phosphorylation (CHIP-pS19) increases CHIP's stability and enhances its affinity for HSP70(Ranek et al., 2020). (replace with ; after half-life)

This process dampes cAMP-mediated signaling and attenuates the cAMP-PKA signaling pathway (Rinaldi et al, 2019).

Responds: Thank you for your meticulous review and constructive suggestions. We have corrected these sentences accordingly to make our expressions more rigorous and accurate.

1. Our previous research revealed the first CHIP monogenetic mutations in a family with cerebellar ataxia, cognitive impairment, and hypogonadism; this led to the classification of a new autosomal recessive form of spinocerebellar ataxia (SCA), known as SCAR16. (We have revised this section accordingly, as detailed on P4, highlighted in yellow.)

2. Moreover, recent discoveries have shown that PKG directly phosphorylates CHIP at serine 19 to increase the CHIP level and extend its half-life; this phosphorylation (CHIP-pS19) increases CHIP stability and enhances its affinity for HSP70(Ranek et al., 2020). (We have revised this section accordingly, as detailed on P4-5, highlighted in yellow.)

2. This process attenuates the cAMP-PKA signalling pathway(Rinaldi et al, 2019). (We have revised this section accordingly, as detailed on P5, highlighted in yellow.)

Results

P5:

CHIP is fundamentally structured around three major domains: TPR (26-127aa), GC (128-226aa) and U-box (227-300aa), a mutation at position p.246M in CHIP results in the inactivation of its E3 ligase activity while preserving its cochaperone functionality (T246M).

Did the Authors mean CC (coiled-coil domain), then GC?

Responds: Thank you very much for your kind reminder. The correct term here is the CC (coiled-coil domain), and we have made the necessary correction. Please see the highlighted section on P5.

P5:

Using exogenous expression of CHIP and PDE9A in a cell culture system, we found that CHIP

decreased PDE9A protein levels in a dose-dependent manner (Fig. 1B). Conversely, PDE9A protein levels decreased CHIP protein levels in a dose-dependent manner (Fig. 1C). The decrease in steady-state protein levels was not reflected at the mRNA level, indicating that while CHIP protein levels change in CHIP mutants-especially with T246M showing a greater reduction compared to K30A-this is accompanied by an opposite trend in PDE9A protein levels, without significant changes in PDE9A mRNA levels. Similarly, PDE9A protein levels do not affect CHIP mRNA levels, suggesting that mutual negative regulation between PDE9A and CHIP occurs at the post-translational level (Fig. 1D-1G, Fig. S1A-S1B)

Suggested changes in the above text:

Using exogenous expression of CHIP and PDE9A in a cell culture system, we found that CHIP decreased the level of PDE9A in a dose-dependent manner (Fig. 1B), whereas PDE9A upregulation decreased CHIP level also in a dose-dependent manner (Fig. 1C). The reduction in steady-state protein levels was not reflected at the mRNA level, suggesting that mutual negative regulation between PDE9A and CHIP occurs at the post-translational level (Fig. 1D-1G, Fig. S1A-S1B).

Moreover, K30A mutation affect ubiquitination activity of CHIP <https://www.ncbi.nlm.nih.gov/pmc/articles/PMC4638040/>, not only binding to HSPs.

The Authors write:

while CHIP protein levels change in CHIP mutants-especially with T246M showing a greater reduction compared to K30A-this is accompanied by an opposite trend in PDE9A protein levels, without significant changes in PDE9A mRNA levels.

Which figure are the Authors referring to? If Fig. 1E, this conclusion is not supported by the presented evidence.

Responds: We sincerely appreciate your professional advice and valuable feedback, which has helped to clarify and refine our statements. We have made the necessary revisions to address the points raised:

1. Thank you for the more precise scientific formulation. We have revised the section as follows:

"Using exogenous expression of CHIP and PDE9A in a cell culture system, we found that CHIP decreased the level of PDE9A in a dose-dependent manner (Fig. 1B), whereas PDE9A upregulation decreased CHIP levels also in a dose-dependent manner (Fig. 1C). The reduction in steady-state protein levels was not reflected at the mRNA level, suggesting that mutual negative regulation between PDE9A and CHIP occurs at the post-translational level (Fig. 1D-1G, Appendix Fig. S1A-S1B)."

(We have updated this section, see P5, highlighted in yellow.)

2. Thank you for your detailed comments. We have improved the following content: "Conversely, the K30A mutation disrupts the binding functionality of the TPR domain to proteins such as HSP70 and affects the E3-ligase activity of CHIP(K30A)." (See P5, highlighted in yellow.)

3. We apologize for the confusion caused by the incorrect expression in the original sentence. The intended message was that the CHIP mutants T246M and K30A exhibit significantly reduced protein levels compared to CHIP, while PDE9A protein levels increase, but without changes in PDE9A mRNA levels. After considering the reviewer's suggestions, we believe deleting this sentence does not affect our experimental conclusions, and the alternative phrasing provided by the reviewer offers a clearer explanation of this section. (We have revised this part, see P5, highlighted in yellow.)

Fig. 1B and Fig. 1C clearly present dose-dependent effects on PDEA9 and CHIP levels. The only puzzle I have, concerns the Fig. 1B: why is FLAG signal apparently detected in untransfected cells and quantified?

Responds: Thank you very much for your review and feedback. In the non-transfected cells, no Flag signal was

detected, and our results confirm the absence of any Flag signal. The confusion likely arose because, to ensure the authenticity of our experimental results, we presented the original images. Consequently, when using ImageJ to measure grayscale values, the background signal in these images may have contributed to some degree of grayscale value in the non-transfected group, resulting in a non-zero quantification. To eliminate this misunderstanding, we have removed the background signal and reanalyzed the quantification using ImageJ. (We have revised this section accordingly, as shown in Figure 1B.)

Figure 1B: Upper: In a set of experiments involving four 10 cm diameter Petri dishes for plasmid transfection, each dish was transfected with a total of 12 μ g of plasmid DNA, followed by cotransfection of 4 μ g of *Flag-CHIP* plasmid with increasing contents of *V5-His-PDE9A* (0, 2, 4, and 8 μ g) and decreasing contents of blank-vector plasmid (8, 6, 4, and 0 μ g) into HEK293T cells. This was followed by western blot analysis to examine the CHIP-mediated regulation of PDE9A levels. Lower: summary data, n = 3 biological replicates/group, 2-way ANOVA (2WANOVA) for each condition, 0 vs. 2, 2 vs. 4, 4 vs. 8, Tukey's multiple comparisons test (mct), V5: ****P < 0.0001 for all comparisons, **P = 0.0025; Flag: ****P < 0.0001 for all comparisons, ***P = 0.0005, P > 0.05 no marks.

The Authors repeatedly throughout the manuscript write about gels, instead of blots, when they describe the Western blot results.

Responds: Thank you very much for your reasonable and professional suggestion. We have replaced all instances of "gels" with "blots" throughout the manuscript to accurately refer to the Western blot results. (We have made this revision across the entire text, as indicated by the yellow highlights.)

Fig. S1C, which presents a bidirectional regulation of CHIP and PDEA9, does not match the actual figure title: The mRNA levels of cell model were detected by qRT-PCR.

Responds: Thank you for your constructive feedback. We have revised the title of Appendix Fig. S1 to "Bidirectional control of CHIP and PDE9A posttranslational modifications and degradation in a cellular model." (This section has been updated, as indicated in Appendix Fig. S1, highlighted in yellow.)

Fig. 1H

legend: language corrections required

H: Co-transfection of 2 μ g PDE9A and 2 μ g CHIP plasmids in HEK293T cells followed by immunofluorescence staining was observed the co-localization of PDE9A and CHIP within the cytoplasm, Imaging was performed using an oil immersion lens under multiPhoto laser scanning microscopy. Experiment replicated \times 3. CHIP (red), PDE9A (green), DPAI (blue).

Responds: Thank you once again for your thorough review. We have revised the sentence as follows:

“**Figure 1H:** After HEK293T cells were cotransfected with 2 μ g of *CHIP-HA* and 2 μ g of *PDE9A-Flag* plasmids, immunofluorescence staining was used to observe the colocalization of PDE9A and CHIP within the cytoplasm. Imaging was performed through multiphoton laser scanning microscopy. Experiment replicated \times 3. CHIP (red), PDE9A (green), and DPAI (blue).”

(Please refer to the legend of Figure 1H, highlighted in yellow.)

Fig.1I

There is indeed a 6-hour half-life indicated on the blots; however, the charts show a 10-hour half-life. Could you explain the reason for this discrepancy?

Responds: We greatly appreciate your thorough review and detailed comments, which have helped make our research results more scientifically rigorous. Similar to the issue raised earlier, since our images are original, the background color may have interfered with the grayscale measurements during ImageJ quantification. Your valuable feedback brought this issue to our attention, and we have now removed the background interference and reanalyzed the quantification. The blot images and corresponding quantification graphs have been unified. (The relevant content has been revised, as shown in Figure 1I, Figure 1J, and Appendix Figure S1C.)

Figure 1I: Cotransfections were performed in each well of a six-well plate by introducing 2 μ g of *CHIP-HA* and 2 μ g of empty vector plasmid or 2 μ g of *PDE9A-Flag* and 2 μ g of empty vector plasmid into HEK293T cells, followed by treatment with 50 nM bafilomycin A1 (autophagy inhibitor) or 100 nM bortezomib (proteasome inhibitor, PS-341) with cycloheximide (CHX, which inhibits protein synthesis) added at six time points (0, 2, 4, 6, 8, and 10 h). Upper: Blot assessment of the half-lives and degradation pathways of CHIP and PDE9A. Lower: summary data, $n = 3$ biological replicates/group, comparing slope differences by analysis of covariance (ANCOVA), **** $P < 0.0001$ for all comparisons.

Fig. 2A

In the blot of the lysate, the CHIP protein lacking TPR appears to be approximately 25 kDa in size. However, the blot for the immunoprecipitation (IP) is truncated, leaving it unclear whether this variant indeed fails to interact with PDE9A.

Given that PDE9A interacts with TPR, what is the functional relevance of its association with HSP70? Is it possible that PDE9A and HSP70 bind simultaneously to different sites within TPR? Additionally, Das et al. (2022) and others demonstrated that the interaction of TPR with HSP70 inhibits CHIP activity. How do your observations align with these findings and other studies concerning the impact of HSP70 on CHIP

activity within your proposed model?

Responds: We are once again deeply grateful for your insightful comments and thought-provoking questions, from which we have greatly benefited. As previously mentioned, we believe that the potential role of HSP70 does not significantly impact the core conclusions of this study. Therefore, we have downplayed this aspect and moved the relevant results to the supplementary materials. Nevertheless, we truly appreciate the opportunity to engage in this in-depth theoretical discussion with the reviewer, which has inspired our future research endeavors. As such, we will still provide a detailed response to each of your points.

1. We sincerely thank you for your meticulous and rigorous review. We apologize for the misunderstanding caused by the incomplete display of the image. We have now fully displayed the blot image, including the 25kDa molecular weight. (This content has been updated, as shown in Figure 2A.)

Figure 2A: Coimmunoprecipitation assays to identify interacting domains between PDE9A and CHIP. Antibodies: V5-rabbit and Flag-mouse. Experiment replicated $\times 3$.

2. As an E3 ubiquitin ligase, CHIP mediates ubiquitination through two possible mechanisms. The first and more common mechanism involves the TPR domain of CHIP, which typically binds to the C-terminal of molecular chaperones such as HSP70 or HSP90. These chaperones, in turn, bind to substrate proteins, indirectly bringing CHIP to these substrates, thus enabling the U-box domain to attach ubiquitin chains to the substrate. The second and less common mechanism is when the TPR domain of CHIP directly binds to specific sequences or regions of the substrate protein. However, this process is rare and generally relies on the mediation of chaperone proteins. Direct binding of the TPR domain to substrate proteins occurs under certain conditions, primarily when the substrate protein contains a C-terminal sequence similar to that of HSP70 or HSP90, such as an EEVD motif, allowing the TPR domain to recognize it directly. Alternatively, exposed hydrophobic regions or abnormal structures in the substrate protein may be directly recognized by the TPR domain. In cases where post-translational modifications alter the structural state of the substrate protein, it may also bind directly to the TPR domain. In the case of PDE9A, it does not possess a typical C-terminal sequence (e.g., EEVD motif), and our earlier *in vitro* experiments indicated that CHIP may not directly transfer ubiquitin chains to PDE9A. We hypothesize that CHIP may not directly bind to PDE9A. To support this, we further supplemented our findings by showing that HSP70 co-localizes with PDE9A and CHIP in the cytoplasm. Additionally, we observed interactions between HSP70, PDE9A, and the TPR domain of CHIP, which suggests that these proteins may form a complex, facilitating the recognition and binding of PDE9A by the TPR domain of CHIP. (See Appendix Figure S2C-S2D for details.)

Appendix Figure S2: The interplay between PDE9A and HSP70.

A: β -SBD of HSP70 binds to PDE9A; NBD: N-terminal nucleotide-binding domain; β -SBD: β -sandwich C-terminal substrate-binding domain; α -SBD: α -helical lid C-terminal substrate-binding domain.

B: Top prediction of the hydrogen bond interface of β -SBD and PDE9A. The CYS 338 site of PDE9A and the TYR 183 site of HSP70 are connected by hydrogen bonding. GRAMM-X software (<http://vakser.bioinformatics.ku.edu/resources/gramm/grammx>) was used to predict possible models. Visualisation of the prediction in PDBePISA and dockeasy online (www.dockeasy.cn).

C: Coimmunoprecipitation assays to identify interacting domains between PDE9A and CHIP and endogenous HSP70. Antibodies : V5-rabbit, Flag-mouse and HSP70-mouse. Experiment replicated $\times 3$.

D: HEK293T cells were cotransfected with $2 \mu\text{g}$ of *PDE9A-Flag* and $2 \mu\text{g}$ of the *HSP70-Myc* plasmid, after which immunofluorescence staining was used to observe the colocalisation of PDE9A and HSP70 within the cytoplasm. Imaging was performed via multiphoton laser scanning microscopy. Experiment replicated $\times 3$. HSP70 (red), PDE9A (green), and DAPI (blue).

. Summary plot (C)

3. The CHIP-Hsp70 complex plays a crucial role in cellular protein quality control and stress response. Hsp70 can act as either an activator or inhibitor of CHIP's E3-ligase activity, depending on the substrate. CHIP can bind to Hsp70 via its TPR domain to inhibit CHIP activity (for instance, by regulating Hsp70's ATPase activity, thereby extending the substrate protein's folding time), or it can bind to Hsp70 to promote the ubiquitination and degradation of substrate proteins. The key lies in how the dual function of CHIP and Hsp70 interaction coordinates the balance between protein folding and degradation. The interaction between CHIP's TPR domain and Hsp70 is central to this regulatory mechanism, ensuring the balance between folding and ubiquitination. When CHIP binds to Hsp70 through its TPR domain, it can suppress CHIP's ubiquitination activity, thus prolonging the folding time of the substrate protein and aiding in proper protein folding. However, when Hsp70 is unable to complete the folding task, CHIP activates its E3 ubiquitin ligase function via the same TPR domain, leading to the ubiquitination and subsequent degradation of the substrate protein. This regulatory mechanism ensures that CHIP can selectively determine the fate of the substrate protein based on its folding status, effectively balancing the processes of protein folding and degradation.

1) Research on the interaction between CHIP and Hsp70 and its function: Ballinger, C. A., Connell, P., Wu, Y., Hu, Z., Thompson, L. J., Yin, L. Y., & Patterson, C. (1999). Identification of CHIP, a novel tetratricopeptide repeat-containing protein that interacts with heat shock proteins and negatively regulates chaperone functions. *Molecular and Cellular*

Biology, 19(6), 4535–4545. [PMID: 10330192]

2) CHIP binds to Hsp70 via the TPR domain, extending the folding time of substrate proteins: Rosser, M. F., Washburn, E., Muchowski, P. J., & Patterson, C. (2007). The Hsp70 and Hsp110 molecular chaperones control the folding of huntingtin fragments: implications for Huntington's disease. *Journal of Biological Chemistry*, 282(1), 145–155. [PMID: 17046820]

3) CHIP interacts with Hsp70 through its TPR domain to mediate the ubiquitination and degradation of substrates when folding fails: Murata, S., Minami, Y., Minami, M., Chiba, T., & Tanaka, K. (2001). CHIP is a chaperone-dependent E3 ligase that ubiquitylates unfolded protein. *EMBO Reports*, 2(12), 1133–1138. [PMID: 11743030]

4) The CHIP-Hsp70 complex has a dual function in substrate protein folding and degradation, regulating the balance between these two processes: S. B., McDonough, H., Boellmann, F., Cyr, D. M., & Patterson, C. (2006). CHIP-mediated stress recovery by sequential ubiquitination of substrates and Hsp70. *Nature*, 440(7083), 551–555. [PMID: 16554823]

5) CHIP inhibits ubiquitination by regulating Hsp70's ATPase activity: Zhang, M., Windheim, M., Roe, S. M., Peggie, M., Cohen, P., Prodromou, C., & Pearl, L. H. (2015). Chaperoned ubiquitylation--crystal structures of the CHIP U box E3 ubiquitin ligase and a CHIP-Ubc13-Uev1a complex. *Molecular Cell*, 20(4), 525–538. [PMID: 16109373].

Therefore, previous studies, along with our research, further suggest that PDE9A can indirectly interact with CHIP via HSP70. CHIP binds to Hsp70 through its TPR domain, while HSP70 recognizes the substrate protein PDE9A, facilitating CHIP's transfer of ubiquitin to the HSP70-recognized PDE9A, leading to ubiquitination. As CHIP adds additional ubiquitin molecules to PDE9A, a polyubiquitin chain is formed. Through its E3 ubiquitin ligase function, CHIP marks PDE9A for degradation. This mechanism ensures timely processing of abnormal PDE9A within the cell, preventing toxic effects caused by excessive PDE9A accumulation.

Fig.S2

Not clear what is what colour-coded.

Responds: Thank you for your kind reminder to improve this image. The green color represents the PDE9A protein, and the red color represents the HSP70 protein. (We have fully incorporated this information, as shown in Appendix Figure S2.)

Appendix Figure S2: The interplay between PDE9A and HSP70.

A: β -SBD of HSP70 binds to PDE9A; NBD: N-terminal nucleotide-binding domain; β -SBD: β -sandwich C-terminal substrate-binding domain; α -SBD: α -helical lid C-terminal substrate-binding domain.

B: Top prediction of the hydrogen bond interface of β -SBD and PDE9A. The CYS 338 site of PDE9A and the TYR 183 site of HSP70 are connected by hydrogen bonding. GRAMM-X software (<http://vakser.bioinformatics.ku.edu/resources/gramm/grammx>) was used to predict possible models. Visualisation of the prediction in PDBePISA and dockeasy online (www.dockeasy.cn).

Fig. S3 :

Could Authors explain why the ubiquitination profile of PDE9A in panel A differs significantly from that in panel F, despite the assays being virtually identical, particularly regarding CHIP WT-driven ubiquitination?

Responds: We sincerely thank you once again for your thorough review. The issue with panel F might have been caused by low ubiquitin chain synthesis efficiency during that particular experiment, likely due to operational factors. As a result, we repeated the experiment, and the results now show a significant CHIP WT-driven ubiquitination reaction. (We have revised this section accordingly, as shown in Appendix Figure S3G.)

Additionally, we have restructured Figure S3, so the original Figure S3F is now positioned as Appendix Figure S3G.

Appendix Figure S3G: *In vivo* ubiquitination assays delineate the domains of CHIP crucial for ubiquitin chain synthesis. The absence of CHIP structural domains hinders ubiquitin chain formation. HEK293T cells were cotransfected with 6 μ g of the *V5-His-PDE9A* plasmid, 6 μ g of the *HA-Ub* plasmid, and 6 μ g of the U-box domain of the CHIP deletion or TPR domain of the CHIP deletion plasmid in 10 cm diameter Petri dishes. Experiment replicated \times 3. Antibodies: V5-rabbit, HA-mouse, and Flag-mouse.

Double G:

B&C&D&E&G&H&G&I

Responds: Thank you for pointing out the mismatch between the Figures and Figure Legends in our manuscript. We fully understand that such errors can impact the overall quality of the paper. As a result, we have carefully reviewed and revised the entire manuscript to ensure that all such mistakes have been corrected, and that every Figure and its corresponding Legend are now properly aligned. We apologize for any oversight and are committed to avoiding similar issues in future submissions. (This section has been revised, as noted in the legend of Appendix Figure S3.)

Fig. S3G Mismatch between figure legend and an actual figure

The title: CHIP-mediated poly-ubiquitination of PDEA9.

Responds: We would like to express our sincere gratitude once again for your valuable comments. We have restructured Appendix Figure 3 and have carefully reviewed and revised the legend of Appendix Figure 3 in detail.

Appendix Figure S3: CHIP mediates the polyubiquitination and degradation of PDE9A.

A: Intracellular ubiquitination of CHIP and PDE9A. HEK293T cells were cotransfected with 6 μ g of the *V5-His-PDE9A* plasmid, 6 μ g of the *Flag-CHIP* plasmid, and 6 μ g of the ubiquitin mutant variants in 10 cm diameter Petri dishes. Experiment replicated \times 3. Antibodies: V5-rabbit, HA-mouse, and Flag-mouse.

B–E&H–I: Ubiquitin content in the lysates of different groups from the *in vitro* ubiquitination assay.

F: Cotransfections were performed in each well of a six-well plate by introducing 2 µg of *V5-His-PDE9A-K186R* and 2 µg of empty vector plasmid or 2 µg of *V5-His-PDE9A-K186R* and 2 µg of *CHIP-Flag* plasmid into HEK293T cells, followed by treatment with cycloheximide (CHX, which inhibits protein synthesis) at six time points (0, 2, 4, 6, 8, and 10 h). Upper: Blotting was used to assess the half-lives and degradation of PDE9A. Lower: summary data, n = 3 biological replicates/group, comparing slope differences by analysis of covariance (ANCOVA), ****P < 0.001.

G: *In vivo* ubiquitination assays delineate the domains of CHIP crucial for ubiquitin chain synthesis. The absence of CHIP structural domains hinders ubiquitin chain formation. HEK293T cells were cotransfected with 6 µg of the *V5-His-PDE9A* plasmid, 6 µg of the *HA-Ub* plasmid, and 6 µg of the U-box domain of the CHIP deletion or TPR domain of the CHIP deletion plasmid in 10 cm diameter Petri dishes. Experiment replicated x 3. Antibodies: V5-rabbit, HA-mouse, and Flag-mouse.

Summary plot (A–J) and regressions (F), ****P < 0.001.

(This content has been revised, as detailed in the legend of Appendix Figure S3.)

Fig. S4 :

Panels F and G exhibit a mismatch between the actual figures and the absence of corresponding legends. Additionally, these panels are not referenced or discussed in the main text.

For panel G, the quantifications do not convincingly correlate with the representative blot provided.

Responds: We sincerely apologize for the error in our figure assembly, which resulted in the duplication of images. Fig. S4F and Fig. S4G were mistakenly repeated with Appendix Fig. S5B and Appendix Fig. S5C. We have corrected this mistake, thoroughly reviewed Appendix Figure 4, and removed the redundant Fig. S4F and Fig. S4G. These sections are now represented by Appendix Fig. S5B and Appendix Fig. S5C. Additionally, we performed a new quantification analysis after removing background interference. For aesthetic purposes, we set the Y-axis range for Fig. S4G (now Appendix Fig. S5C) from 0.6 to 1.0 rather than starting from 0, which may make the bar graph appear visually lower compared to the other figures.

Appendix Figure S5B: Phosphorylation levels of CHIP at serine 20 in cerebellar and hippocampal tissues across Bay 73-6691-treated wild-type rats. Upper: Blots showing the CHIP and p-CHIP (phosphorylated CHIP) levels in WT+Bay(+) (Bay 73-6691-treated wild-type rats) and WT (wild-type littermates), with each lane containing a mixed-tissue protein sample from three rats. Lower: summary data, n = 3 biological replicates/group, 1WANOVA, Tukey mct, ****P < 0.001, ***P < 0.01.

Appendix Figure S5C: The levels of CHIP phosphorylation in the cellular model. HEK293T cells were transfected with 4 µg of empty vector plasmid (NC) or 4 µg of *HA-CHIP* plasmid (CHIP). NC+Bay(+) indicates cells treated with 200 µg/mL Bay 73-6691, and CHIP+Bay(+) indicates cells transfected with *HA-CHIP* plasmids and treated with 200 µg/mL Bay 73-6691. Upper: Blots showing the CHIP and p-CHIP levels in the cellular models. Lower: summary data, n = 3 biological replicates/group, 1WANOVA, Tukey mct, ****P < 0.001, ***P < 0.01, **P < 0.02.

(We have revised this section, and the results can be found in Appendix Figure S4 and the legend of Appendix Figure S5.)

To ensure the quality of the manuscript, we have now thoroughly reviewed and verified all Figures and Figure Legends to prevent such errors from occurring again.

"Treatment with Bay 73-6691 Improves Mitophagy Dysfunction and Cell Apoptosis in CHIP Mutations
PDE9A reduces the levels of intracellular CHIP via decrements in CHIP phosphorylation through the cGMP-PKG signaling cascade. The depletion of CHIP curtails the levels of K63- and K27-linked polyubiquitination of PDE9A, inhibiting its lysosomal degradation and thus promoting intracellular PDE9A aggregation."

However, is it accurate to describe this accumulation as 'aggregation'?

Responds: Thank you for your careful review and very constructive suggestion. In this study, we primarily aimed to demonstrate the increase in intracellular PDE9A. Due to translation issues, this increase was inaccurately referred to as "aggregation." We agree that the term "accumulation," as you suggested, more accurately reflects the conclusions of our research. Therefore, we have replaced "PDE9A aggregation" with "PDE9A accumulation" throughout the manuscript.

PDE9A reduces the levels of intracellular CHIP by decreases in CHIP phosphorylation through the cGMP-PKG signalling cascade. Depletion of CHIP decreases the degree of K63- and K27-linked polyubiquitination of PDE9A, inhibiting its lysosomal degradation and thus promoting intracellular PDE9A accumulation.

(We have revised this section accordingly, as indicated in the manuscript, P10, P12, highlighted in yellow.)

P12:

In the text the Authors cite a paper by Mishra et al., 2021:

Interestingly, PDE9A can localize to mitochondria, where its aggregation inhibits mitochondrial respiration, leading to mitochondrial swelling and impairment of autophagic function(Mishra et al, 2021).

In this cited paper Authors concluded: „PDE9 localizes to mitochondria where it can suppress mitochondrial FA oxidation and respiration, and that its inhibition augments both.'. However, they don't mention any impact of PDE9A on autophagy.

Responds: We sincerely appreciate your meticulous review, and we apologize for the confusion caused by a translation error. This sentence was meant to convey that after PDE9A accumulation, mitochondrial respiration is inhibited, leading to mitochondrial swelling. The damaged mitochondria subsequently result in impaired mitophagy function. To ensure clearer understanding, we have revised and supplemented this section with more detailed explanations:

"Dysregulation of mitophagy results in the accumulation of cytotoxic proteins. Notably, PDE9A can localize to mitochondria, where it regulates mitophagy via the second messenger cGMP. cGMP promotes autophagosome formation and the initiation of mitophagy through pathways involving PKG signalling, oxidative stress responses, and BNIP3. However, overaccumulation of PDE9A decreases cGMP levels, inhibiting mitochondrial respiration and leading to mitochondrial swelling and impaired mitophagy function(Mishra *et al*, 2021)".

(We have updated this section, as indicated on P9, highlighted in yellow.)

P13:

A sentence with required correction:

Thus, we endeavoured to discern whether whether a reduction in CHIP due to mutations and a consequent increase in neuronal PDE9A levels could lead to mitochondrial abnormalities and disrupt the equilibrium of mitophagy.

Responds: Thank you once again for your patience and thorough review. We have made the following revision:

"Thus, we investigated whether the mutation of CHIP and the increase in PDE9A levels could lead to mitochondrial abnormalities and disrupt the equilibrium of mitophagy. "

(This section has been revised, as indicated on P12, highlighted in yellow.)

Fig. 5E and F

On figures a name of the protein BNIP3 was incorrectly changed to BINP3.

Responds: Thank you for your kind reminder. We have corrected this content, as indicated in Fig. 5E and Fig. 5F.

Fig. 5G

For mitochondrial assessment the Authors use Flameng score: Mitochondrial assessment using the Flameng method revealed significantly higher scores in the Hom group compared to the other three groups, with a decrease in mitochondrial perimeter and area.

However, they neither explain the procedure in the Methods section nor cite a paper describing this method.

Responds: We greatly appreciate your suggestion, which enhances the completeness of our content. The Flameng score method used in this study to assess mitochondrial structure is based on the following references:

1. Flameng W, Borgers M, Daenen W, et al. Ultrastructural and cytochemical correlates of myocardial protection by cardiac hypothermia in man [J]. The Journal of Thoracic and Cardiovascular Surgery, 1980, 79(3): 413-424.
2. Shaw G A, Hyer M M, Targett I, et al. Traumatic stress history interacts with sex and chronic peripheral inflammation to alter mitochondrial function of synaptosomes [J]. Brain, behavior, and immunity, 2020, 88: 203-219.

(We have added this information, as indicated on P12, highlighted in yellow.)

P14:

A sequence for correction:

We obtained four sets of scRNA-seq results, encompassing the CHIP homozygous mutation group (Hom), AAV-mediated exogenous CHIP expression group (Hom+AAV-CHIP), Bay 73-6691 treatment group (Hom+Bay(+)) and a cohort of wild rats as the control group (Wt), capturing 19,279, 25,238, 23,177 and 13,473 individual cells, respectively (Fig. 6A, Fig. S7A).

Responds: Thank you very much for your patient and thorough review. Your valuable suggestions have helped make our manuscript clearer and more logical. We have revised the sequence of the sentence as follows:

"We obtained four sets of scRNA-seq results, encompassing a cohort of wild-type rats as the control group (WT), CHIP homozygous mutation group (Hom), AAV-mediated exogenous CHIP expression group (Hom+AAV-CHIP), and Bay 73-6691 treatment group (Hom+Bay(+)), capturing 13,473, 19,279, 25,238, and 23,177 individual cells, respectively (Fig. 6A, Appendix Fig. S7A)."

(We have adjusted this section, as indicated on P13, highlighted in yellow.)

Fig. 8 and Fig. 9

Mismatch between these figures and legends

Responds: Thank you once again for pointing out the error in the order of the Figures in our manuscript. Due to an upload system error, the images for Figure 8 and Figure 9 appeared in the incorrect order in the generated manuscript; however, the descriptions of these two Figures in the text are accurate. Therefore, we have corrected the order of these images in the system. We sincerely apologize for this oversight. We have carefully reviewed all Figures and legends throughout the manuscript and have diligently corrected all instances of mismatch between Figures and legends. Additionally, multiple colleagues have conducted detailed reviews. We are committed to ensuring that similar errors do not occur in future submissions and will guarantee the submission of the highest quality text.

(We have corrected this error, as indicated in Figure 8 and Figure 9.)

P15:

Missing data related to the measurement of Purkinje neurons:

Through cell proportion analysis, we found that the Purkinje neuron proportion in the Hom group was significantly lower than that in the Wt group. Interestingly, treatment with Bay 73-6691 and AAV-CHIP resulted in an increase in the number of Purkinje neurons of Hom group, suggesting that Bay 73-6691 and AAV-CHIP treatments could partially restore the number, thereby ameliorating the associated symptoms in CHIP mutation rats.

Responds: Thank you once again for your reminder, which has helped make our manuscript more comprehensive. The proportions of Purkinje neurons in the four groups are as follows: Wt: 0.87%, Hom: 0.24%, Hom+AAV-CHIP: 4.54%, Hom+Bay(+): 0.78%. The complete description in the text is:

"Through cell proportion analysis, we found that the proportion of Purkinje neurons in the Hom group was significantly lower than that in the WT group. Interestingly, treatment with Bay 73-6691 and AAV-CHIP resulted in an increase in the number of Purkinje neurons in the Hom group, suggesting that Bay 73-6691 and AAV-CHIP treatments could partially restore the number of Purkinje neurons, thereby ameliorating the associated symptoms in *CHIP*-mutant rats (Wt: 0.87%, Hom: 0.24%, Hom+AAV-CHIP: 4.54%, Hom+Bay(+): 0.78%)."

(We have added this information, as indicated on P14-15, highlighted in yellow.)

P16:

Moreover, PKG mediates the reduction of p-CHIP and the decrease in CHIP half-life resulting from the increased PDE9A levels.

Whereas at the page 4 they write: Meanwhile, recent discoveries have highlighted that PKG directly phosphorylates CHIP at serine 19 to increase CHIP level and extend its half-life, this phosphorylation (CHIP-pS19) increases CHIP's stability and enhances its affinity for HSP70 (Ranek et al., 2020).

Responds: Thank you very much for taking the time to conduct a detailed and patient review of this manuscript. The confusion here was due to our inaccurate translation. We primarily intended to convey that an increase in PDE9A protein levels results in a decrease in PKG levels. This reduction in PKG leads to a concomitant decrease in p-CHIP and shortens CHIP's half-life. We have revised this as follows:

"Moreover, elevated levels of PDE9A result in a reduction in PKG levels. This decrease in PKG leads to a concomitant reduction in p-CHIP and a decrease in the half-life of CHIP."

(We have made this revision, as indicated on P15, highlighted in yellow.)

P16:

A wording should be more precise (e.g. on CHIP homozygous mutants)

We, therefore, also performed scRNA-seq on the CHIP homozygous mutations treated with Bay 73-6691

Responds: We would like to express our heartfelt gratitude for your valuable review. We have provided a more precise

description of this point:

"Therefore, we also performed scRNA-seq on the *CHIP* mutants treated with Bay 73-6691 to assess the aforementioned therapeutic effect on Purkinje neurons."

(We have revised this section, as indicated on P15, highlighted in yellow.)

P16:

In addition, to provide a comprehensive assessment of the effects of AAV-NC and Bay 73-6691 treatment on *CHIP* mutations animal model, we used a heatmap to highlight the prominent changes in gene expression in pathways such as cAMP signaling, cGMP-PKG signaling, oxidative phosphorylation, ubiquitin-mediated proteolysis, mitophagy and autophagy.

Should be replaced with AAV-CHIP

Responds: Thank you very much for your friendly reminder. This was a writing error, and the correct term is to change "AAV-NC" to "AAV-CHIP":

"In addition, to provide a comprehensive assessment of the effects of AAV-CHIP and Bay 73-6691 treatment on the animal model of *CHIP* mutation, we used a heatmap to highlight the prominent changes in gene expression in pathways such as cAMP signalling, cGMP-PKG signalling, oxidative phosphorylation, ubiquitin-mediated proteolysis, mitophagy and autophagy."

(We have corrected this section, as indicated on P15, highlighted in yellow.)

We sincerely apologize for our oversight. We have now carefully reviewed the entire manuscript and corrected all identified writing errors. We assure you that we will provide a high-quality manuscript in future submissions.

P17:

Heatmap is shown on Fig. S9 (not Fig. S8).

Responds: Thank you very much for your detailed and patient guidance. We have corrected the text by changing Fig. S8 to the correct Appendix Fig. S9, as indicated on P16, highlighted in yellow. We assure you that we will avoid similar mistakes in future submissions.

Fig. S10

Regulation of cAMP-cGAMP signaling pathway by PDE9A inhibitors. (inhibitor)

Responds: Thank you for your thorough review. We have corrected this error to: "Regulation of cAMP-cGAMP signaling pathway by PDE9A inhibitor." (See the legend of Appendix Figure S10.)

Additionally, we have had the entire manuscript professionally edited and proofread.

B: Upper: Gel evaluate cGMP-PKA signalling protein expression in the cerebellum and hippocampus tissues of WT+Bay(+) group and WT group. Tissue protein sample of three rat mixed for each lane.

Each summary panel are presented as mean{plus minus}SD

Suggested correction: Blot evaluates PRKG1 and PRKG2 levels in the cerebellum and hippocampus of Bay 73-6691-treated rats (WT+Bay(+)) relative to wild-type rats. Tissue protein samples of three rats were mixed for each lane.

Responds: Thank you for your professional suggestions and valuable feedback, which have made our figure legends clearer and contributed to the improvement of our manuscript. We have replaced the content with the following:

"Blot analysis of PRKG1 and PRKG2 levels in the cerebellum and hippocampus of Bay 73-6691-treated rats (WT+Bay(+)) relative to those in wild-type rats. The protein samples from three rats were mixed for each lane."

(See Appendix Figure S10 for details.)

P17:

The scRNA-seq results not only support the notion that CHIP mutations lead to the abnormal (reduced?) formation of K63 and K27 polyubiquitin chains on PDE9A, impacting PDE9A's degradation in autophagolysosomes, but also indicate that PKG mediates the reduction of p-CHIP and the shortened half-life of CHIP resulting from increased PDE9A levels.

Responds: Thank you once again for your thorough review, which has helped make our manuscript clearer. We have rephrased the statement as follows:

"The scRNA-seq results not only support the notion that *CHIP* mutations reduce the formation of K63 and K27 polyubiquitin chains on PDE9A, impacting PDE9A degradation in autophagolysosomes but also indicate that PKG mediates the reduction in p-CHIP and the shortened half-life of CHIP resulting from increased PDE9A levels."

(We have revised this section, as indicated on P16, highlighted in yellow.)

Fig.10

autophagy-lysosoma pathway

Responds: Thank you for your correction. We have replaced the content in the image with "autophagy-lysosomal pathway," as shown in Figure 10. Additionally, we have conducted a thorough and comprehensive review of the entire manuscript to ensure that similar errors do not occur again.

Referee #3:

This revised submission investigating the interplay between CHIP and PDE9A and the dysregulation of this in SCA is much improved from its initial submission. Overall the clarity of the manuscript has been greatly improved. Additional controls also have improved the manuscript. Overall while much improved there are still some areas that need tightening up and some conclusions should be tempered.

Responds: Thank you very much for your positive feedback and recognition of our revisions. We appreciate the valuable time you dedicated to our manuscript. Your professional review and constructive suggestions have made our article more rigorous and clear. We have enhanced the relevant content and strengthened our conclusions, and we have thoroughly polished the entire manuscript. We are committed to providing a high-quality submission in the future.

Major Concern

1) The PDE9A K186R mutant is an excellent control that now alleviates the concerns that the ubiquitin detected was not due to co-immunoprecipitated proteins. This now provides the authors with a reagent to directly test the hypothesis that CHIP mediated ubiquitination of PDE9A is targeting it for degradation. Is the steady state levels of the K186R mutant increased or does having this mutation increase the half-life of PDE9A in cells?

Responds: Thank you for your constructive question regarding this research; it helps to solidify our conclusions. We conducted experiments to assess the half-life of PDE9A protein using a PDE9A-K186R mutant cell model. The results confirmed that the K186R mutation in PDE9A leads to an increased half-life of the protein. Furthermore, CHIP is unable to ubiquitinate PDE9A at this site, resulting in reduced degradation and consequently elevated intracellular levels of PDE9A (we have refined this section, see Appendix Figure S3F).

Appendix Figure S3F: Cotransfections were performed in each well of a six-well plate by introducing 2 μ g of *V5-His-PDE9A-K186R* and 2 μ g of empty vector plasmid or 2 μ g of *V5-His-PDE9A-K186R* and 2 μ g of *CHIP-Flag* plasmid into HEK293T cells, followed by treatment with cycloheximide (CHX, which inhibits protein synthesis) at six time points (0, 2, 4, 6, 8, and 10 h). Upper: Blotting was used to assess the half-lives and degradation of PDE9A. Lower: summary data, n = 3 biological replicates/group, comparing slope differences by ANCOVA, ****P < 0.001.

Minor Concerns

1) In figure 2 the authors clearly demonstrate that the CHIP TPR domain is needed for both the

interaction with PDE9A and with HSP70. They then conclude that PDE9A is interacting with CHIP and that this is facilitated via HSP proteins. The data presented do not support this conclusion as there is no evidence the three proteins exist in a complex. While this is the most likely explanation, this conclusion is not supported by the data.

Responds:

Thank you for your insightful questions and deep engagement with our research; we've greatly benefited from your feedback. Based on your guidance, we've expanded our knowledge regarding the roles of HSP70, PDE9A, and CHIP, particularly the interactions among them, which remain unclear in the current literature. We've come to realize that our findings regarding HSP70's involvement in the interaction between PDE9A and CHIP represent a plausible hypothesis, but further detailed studies are necessary to establish concrete connections. In this study, we propose that HSP70 may play a role in facilitating the binding of CHIP to PDE9A. Furthermore, we detected endogenous HSP70 signals in the interaction assays between PDE9A and the TPR domain. These results suggest interactions among TPR, PDE9A, and HSP70, indicating a potential co-localization of HSP70 and PDE9A in the cytoplasm. This may imply that a complex forms between PDE9A, HSP70, and CHIP's TPR domain, potentially enhancing CHIP's recognition and binding to PDE9A (see Appendix Figure S2C-S2D).

Appendix Figure S2: Interplay between PDE9A and HSP70.

A: β -SBD of HSP70 binds to PDE9A; NBD: N-terminal nucleotide-binding domain; β -SBD: β -sandwich C-terminal substrate-binding domain; α -SBD: α -helical lid C-terminal substrate-binding domain.

B: Top prediction of the hydrogen bond interface of β -SBD and PDE9A. The CYS 338 site of PDE9A and the TYR 183 site of HSP70 are connected by hydrogen bonding. GRAMM-X software (<http://vakser.bioinformatics.ku.edu/resources/gramm/grammx>) was used to predict possible models. Visualisation of the prediction in PDBePISA and dockeasy online (www.dockeasy.cn).

C: Coimmunoprecipitation assays to identify interacting domains between PDE9A and CHIP and endogenous HSP70. Antibodies : V5-rabbit, Flag-mouse and HSP70-mouse. Experiment replicated $\times 3$.

D: HEK293T cells were cotransfected with 2 μ g of *PDE9A-Flag* and 2 μ g of the *HSP70-Myc* plasmid, after which immunofluorescence staining was used to observe the colocalisation of PDE9A and HSP70 within the cytoplasm. Imaging was performed via multiphoton laser scanning microscopy. Experiment replicated $\times 3$. HSP70 (red), PDE9A (green), and DAPI (blue).

. Summary plot (C)

Meanwhile, we carefully considered your suggestions, along with relevant research and our current findings. We

believe that the potential role of HSP70 does not significantly impact the core conclusions of this study. Therefore, we have downplayed this aspect in the text and moved the relevant content to the supplementary materials (we have revised this section, see P6).

We are once again grateful for the opportunity to engage in this in-depth discussion with you, which will inspire our future research endeavors.

2) In Figure 2a no delta TPR CHIP is present in the IP blot (appears to be cropped out?).

Responds: Thank you very much for your meticulous and rigorous work. We have now fully displayed the blot image, including the 25 kDa molecular weight marker. (We have updated this section, as shown in Figure 2A.)

Figure 2A: Coimmunoprecipitation assays to identify interacting domains between PDE9A and CHIP. Antibodies: V5-rabbit and Flag-mouse. Experiment replicated x 3.

3) There is inconsistent referencing to figures throughout. Some interpretations are made but no corresponding figure is referenced.

Responds: Thank you for pointing out the mismatches in the figure citations within our manuscript. We fully recognize that such errors can impact the overall quality of our work. Therefore, we have carefully reviewed and revised the entire manuscript to ensure that all such mistakes have been corrected, and that each figure is accompanied by detailed and clear interpretations in the text. We apologize for any oversights and are committed to avoiding similar errors in future submissions. (We have made the relevant corrections throughout the manuscript.)

4) "lysine residue requisite of ubiquitin chain assembly on PDE9A" in reference to figure 2. That is not what is tested here, instead ubiquitin chain linkages are being tested.

Responds: Thank you very much for your kind suggestion. We have revised this statement to: "To identify the ubiquitin chain linkages on PDE9A," (see P7, highlighted in yellow.)

5) When discussing figure 8 of the text the authors reference figure 9.

Responds: Thank you once again for pointing out the error in the order of the Figures in our manuscript. Due to an upload system issue, Figures 8 and 9 appeared in the incorrect order in the generated manuscript, although their descriptions in the text were accurate. We have corrected the order of these images in the system. We sincerely apologize for this oversight. We have thoroughly reviewed all Figures and legends throughout the manuscript and have diligently corrected all instances of mismatch between Figures and legends. Additionally, multiple colleagues have conducted detailed reviews to ensure that similar errors do not occur in the future. (We have corrected this error, as indicated in Figures 8 and 9.)

6) There are still a number of grammatical errors and typos that should be corrected.

Responds: We would like to express our gratitude once again for your attention to the language of this manuscript. We fully understand and support the importance of clear and concise writing in academic papers. In light of this, we have engaged a native English-speaking expert to comprehensively revise our language, ensuring fluency, accuracy, and consistency. Additionally, we have polished the manuscript, carefully revising the language while paying attention to sentence structure, word choice, and tone. We assure you that we will submit a text of the highest standard in our future submissions.

Prof. Changhe Shi
The First Affiliated Hospital of Zhengzhou University
Neurology
First affiliated hospital of Zhengzhou University, Zhengzhou, Henan, China
Henan 450000
China

9th Dec 2024

Re: EMBOJ-2024-116833R1
E3 Ubiquitin Ligase CHIP Facilitates cAMP and cGMP Signalling Cross-talk by Polyubiquitinating PDE9A

Dear Dr. Shi,

Thank you again for submitting your re-revised manuscript to The EMBO Journal, and once more apologies for the delay associated with its re-evaluation. I have now carefully reviewed your answers and modifications in response to referees 2 and 3, and found them overall satisfactory. We shall therefore be happy to accept this work for publication, as soon as a number of remaining editorial issues have been addressed:

- Please provide suggestions for a short 'blurb' text prefacing and summing up the study in two sentences (max. 250 characters), followed by 3-5 one-sentence 'bullet points' with brief factual statements of key results of the paper; they will form the basis of an editor-written 'Synopsis' accompanying the already uploaded synopsis image in the online version of the article.
- Please upload the Appendix PDF file without text markup or tracking of text changes. In the legend for Appendix Figure S2A-B, please clearly state as first sentence what is shown - a structural model? - and how it was generated.
- For the OMICS-based experiments in Figures 6, 7, 9, please note that our policies generally mandate deposition of the underlying datasets in public repositories (such as GEO), with an hyperlink/URL to the database and accession codes being provided in the "Data availability" section of the manuscript. For more information, please see: <https://www.embopress.org/page/journal/14602075/authorguide#datadeposition>
- Please note that Source data files need to be saved according to the scheme: one figure-one folder and then uploaded as .zip files. E.g., all the Source data files for Figure 1 need to be saved in a single folder, which then needs to be zipped and uploaded as "SD figure 1.zip" file. Furthermore, Source Data has to be labelled sufficiently to allow understanding how they relate to the final images, e.g. which blot panels and lanes correspond to which.
- Please rename the Conflict of Interest section into "Disclosure and Competing Interests Statement", in accordance with our updated Guide to Authors (<https://www.embopress.org/competing-interests>)
- As we are switching from a free-text author contribution statement towards a more formal statement based on Contributor Role Taxonomy (CRediT) terms, please remove the present Author Contribution section and instead specify each author's contribution(s) directly in the Author Information page of our submission system during upload of the final manuscript. See <https://casrai.org/credit/> for more information.
- Finally, I noticed that five authors are currently listed as co-corresponding authors, which is unfortunately not allowed under our editorial policies. Although we realize that this work includes several different sets of expertise, I should make clear that it is hard to understand how five investigators can have an equal senior role, which of course also means taking full responsibility for all content of the paper, as well as for the distribution of reagents and protocols. Clearly, we would consider a detailed description of each author's specific contributions (using CRediT taxonomy as well as the free-text boxes in the submission system) more valuable, and I would therefore ask you to reconsider the authorship status assignments, retaining at maximum three co-corresponding authors.
- Related to the previous point: please notice that we mandate that every corresponding author has linked their author profile to an ORCID identifier (which Dr. Yuming Xu and Dr Yanxia Gao currently had not), AND use a current institutional email address, rather than some free-mail address (as currently the case for Dr. Yanxia Gao, Dr. Yong Jiang, and yourself).

I am therefore returning the manuscript to you for a final round of revision, to allow you to make these adjustments and upload all modified files. Please do not hesitate to contact me, in case you should have any questions in this regard.

Yours sincerely,

Hartmut Vodermaier

*** PLEASE NOTE: All revised manuscripts are subject to initial checks for completeness and adherence to our formatting guidelines. Revisions may be returned to the authors and delayed in their editorial re-evaluation if they fail to comply to the following requirements (see also our Guide to Authors for further information):

- 1) Every manuscript requires a Data Availability section (even if only stating that no deposited datasets are included). Primary datasets or computer code produced in the current study have to be deposited in appropriate public repositories prior to resubmission, and reviewer access details provided in case that public access is not yet allowed. Further information: embopress.org/page/journal/14602075/authorguide#dataavailability
- 2) Each figure legend must specify
 - size of the scale bars that are mandatory for all micrograph panels
 - the statistical test used to generate error bars and P-values
 - the type error bars (e.g., S.E.M., S.D.)
 - the number (n) and nature (biological or technical replicate) of independent experiments underlying each data point
 - Figures may not include error bars for experiments with $n < 3$; scatter plots showing individual data points should be used instead.
- 3) Revised manuscript text (including main tables, and figure legends for main and EV figures) has to be submitted as editable text file (e.g., .docx format). We encourage highlighting of changes (e.g., via text color) for the referees' reference.
- 4) Each main and each Expanded View (EV) figure should be uploaded as individual production-quality files (preferably in .eps, .tif, .jpg formats). For suggestions on figure preparation/layout, please refer to our Figure Preparation Guidelines: <http://bit.ly/EMBOPressFigurePreparationGuideline>
- 5) Point-by-point response letters should include the original referee comments in full together with your detailed responses to them (and to specific editor requests if applicable), and also be uploaded as editable (e.g., .docx) text files.
- 6) Please complete our Author Checklist, and make sure that information entered into the checklist is also reflected in the manuscript; the checklist will be available to readers as part of the Review Process File. A download link is found at the top of our Guide to Authors: embopress.org/page/journal/14602075/authorguide
- 7) All authors listed as (co-)corresponding need to deposit, in their respective author profiles in our submission system, a unique ORCID identifier linked to their name. Please see our Guide to Authors for detailed instructions.
- 8) Please note that supplementary information at EMBO Press has been superseded by the 'Expanded View' for inclusion of additional figures, tables, movies or datasets; with up to five EV Figures being typeset and directly accessible in the HTML version of the article. For details and guidance, please refer to: embopress.org/page/journal/14602075/authorguide#expandedview
- 9) To facilitate reproducibility and cross-laboratory adoption of methodologies, please structure the Materials & Methods section as outlined in our guide to authors, including a completed Reagents and Tools Table that can be downloaded from our author guidelines as well (<https://www.embopress.org/page/journal/14602075/authorguide#structuredmethods>).
- 10) Digital image enhancement is acceptable practice, as long as it accurately represents the original data and conforms to community standards. If a figure has been subjected to significant electronic manipulation, this must be clearly noted in the figure legend and/or the 'Materials and Methods' section. The editors reserve the right to request original versions of figures and the original images that were used to assemble the figure. Finally, we generally encourage uploading of numerical as well as gel/blot image source data; for details see: embopress.org/page/journal/14602075/authorguide#sourcedata

At EMBO Press, we ask authors to provide source data for the main manuscript figures. Our source data coordinator will contact you to discuss which figure panels we would need source data for and will also provide you with helpful tips on how to upload and organize the files.

Further information is available in our Guide For Authors:

In the interest of ensuring the conceptual advance provided by the work, we recommend submitting a revision within 3 months (9th Mar 2025). Please discuss the revision progress ahead of this time with the editor if you require more time to complete the revisions. Use the link below to submit your revision:

Link Not Available

Dear Professor Hartmut Vodermaier,

We would like to express our sincere gratitude for your positive evaluation and recognition of our revised manuscript. We deeply appreciate the significant time and effort that you, the editorial team, and the reviewers have invested in improving and refining our manuscript. Your thoughtful feedback has been invaluable in enhancing the quality of our work.

We are also grateful for the final expert suggestions you provided, which ensured that the manuscript is presented in the best possible form for publication in the prestigious *The EMBO Journal*. We have carefully addressed each of the remaining editorial issues and thoroughly reviewed both the final version of our manuscript and the submission system to ensure that all necessary revisions have been implemented (see the following details).

Once again, we sincerely appreciate the collective effort of everyone involved in this process. Your professionalism, patience, and meticulous attention to detail have played a crucial role in helping us present the manuscript in its best possible version. We are truly honored to have the opportunity to publish our work in your esteemed journal.

Thank you again for the dedication and time you and the reviewers have invested in this manuscript. We have made sure that the final version has been fully refined and corrected to reflect all of your professional and constructive suggestions. Should you have any further questions, please do not hesitate to contact me.

Wishing you all the best.

Your sincerely,
Changhe Shi

Address: Department of Neurology, The First Affiliated Hospital of Zhengzhou University,
Zhengzhou University, 1 Jian-she east road, Zhengzhou 450000, Henan, China

Email: shichanghe@zzu.edu.cn

- Please provide suggestions for a short 'blurb' text prefacing and summing up the study in two sentences (max. 250 characters), followed by 3-5 one-sentence 'bullet points' with brief factual statements of key results of the paper; they will form the basis of an editor-written 'Synopsis' accompanying the already uploaded synopsis image in the online version of the article.

Respond:

a. Blurb Text (max. 250 characters):

CHIP mutations disrupt PDE9A degradation, leading to cGMP/cAMP signaling imbalance, impaired mitophagy, and neuronal apoptosis in CHIP-related ataxia. Pharmacological PDE9A inhibition or CHIP restoration rescues cerebellar neuropathology and mitigates neuronal dysfunction.

b. Bullet Points:

(1) CHIP mediates PDE9A polyubiquitination and degradation, maintaining cGMP/cAMP signaling Cross-talk for neuronal survival.

(2) CHIP dysfunction increases PDE9A levels, disrupts cGMP/PKG signaling, and impairs CHIP phosphorylation at serine 19, amplifying mitophagy defects and neuronal apoptosis.

(3) Elevated PKA competitively inhibits PDE9A degradation, with PDE9A accumulation further disrupting the balance between cGMP-cAMP signaling pathways.

(4) PDE9A inhibition via Bay 73-6691 or CHIP overexpression restores cGMP/cAMP signaling homeostasis and rescues mitophagy in Purkinje neurons.

(5) Targeting the CHIP-PDE9A interaction represents a potential therapeutic strategy for CHIP-related ataxia, restoring cerebellar neuronal function and preventing degeneration.

- Please upload the Appendix PDF file without text markup or tracking of text changes. In the legend for Appendix Figure S2A-B, please clearly state as first sentence what is shown - a structural model? - and how it was generated.

Respond: Thank you for the reminder. We have uploaded the Appendix PDF file (clean version), And we have revised the legend of Appendix Figure S2A and the legend of Appendix Figure S2B.

Appendix Figure S2A: β -SBD of HSP70 binds to PDE9A. Molecular docking analysis between PDE9A (PDB: 3QI4) and HSP70 (PDB: 4B9Q) was conducted using GRAMM-X software (<http://vakser.bioinformatics.ku.edu/resources/gramm/grammx>) to predict possible PDE9A and HSP70 interaction models. The top docking model was analyzed for interacting domains, highlighting the β -SBD (β -sandwich C-terminal substrate-binding domain), NBD (N-terminal nucleotide-binding domain), and α -SBD (α -helical lid C-terminal substrate-binding domain) of HSP70 in complex with PDE9A. The visualization was performed using Dockeasy online (www.dockeasy.cn) and illustrates the relative spatial arrangement of the PDE9A and HSP70.

Appendix Figure S2B: Top prediction of the hydrogen bond interface between PDE9A and the β -SBD of HSP70. A zoomed-in view of the top docking prediction from Appendix Figure S2A, focusing on the molecular interface residues. The hydrogen-bond interaction between CYS 338 site of PDE9A and TYR 183 site of HSP70 was identified as the highest-scoring site based on position and distance predictions. The interface residues and their positions were analyzed using PDBePISA (Proteins, Interfaces, Structures and Assemblies) and visualized with Dockeasy online.

- For the OMICS-based experiments in Figures 6, 7, 9, please note that our policies generally mandate deposition of the underlying datasets in public repositories (such as GEO), with an

hyperlink/URL to the database and accession codes being provided in the "Data availability" section of the manuscript.

Respond: We greatly appreciate your professional comments. We have deposited the underlying datasets of the OMICS-based experiments for Figures 6, 7, and 9 in a public repository. The accession codes are PRJCA033500, and the link is: <https://ngdc.cncb.ac.cn/omix/release/OMIX008238>. We have revised the Data Availability section to include this additional information.

- Please rename the Conflict of Interest section into "Disclosure and Competing Interests Statement", in accordance with our updated Guide to Authors (<https://www.embopress.org/competing-interests>)

Respond: We appreciate your thorough review. We have renamed the Conflict of Interest section to "Disclosure and Competing Interests Statement".

- As we are switching from a free-text author contribution statement towards a more formal statement based on Contributor Role Taxonomy (CRediT) terms, please remove the present Author Contribution section and instead specify each author's contribution(s) directly in the Author Information page of our submission system during upload of the final manuscript. See <https://casrai.org/credit/> for more information.

Respond: We once again appreciate your kind reminder. We have removed the Author Contribution section from the main text and have specified each author's contribution(s) directly on the Author Information page of the submission system.

- Finally, I noticed that five authors are currently listed as co-corresponding authors, which is unfortunately not allowed under our editorial policies. Although we realize that this work includes several different sets of expertise, I should make clear that it is hard to understand how five investigators can have an equal senior role, which of course also means taking full responsibility for all content of the paper, as well as for the distribution of reagents and protocols. Clearly, we would consider a detailed description of each author's specific contributions (using CRediT taxonomy as well as the free-text boxes in the submission system) more valuable, and I would therefore ask you to reconsider the authorship status assignments, retaining at maximum three co-corresponding authors.

Respond: We sincerely appreciate your professional suggestion. Given the substantial workload involved, all authors have contributed significantly and made tremendous efforts in refining and revising this manuscript. However, due to practical considerations, and in accordance with a fair ranking of contributions by the corresponding authors, all participants in the article have unanimously agreed that the last three authors in the previous version of the manuscript will remain as co-corresponding authors (Dr. Changhe Shi, Dr. Jonathan C. Schisler, and Dr. Yuming Xu). We have made the necessary updates accordingly.

- Related to the previous point: please notice that we mandate that every corresponding author has linked their author profile to an ORCID identifier (which Dr. Yuming Xu and Dr Yanxia Gao currently had not), AND use a current institutional email address, rather than some free-mail address (as currently the case for Dr. Yanxia Gao, Dr. Yong Jiang, and yourself).

Respond: Thank you once again for your meticulous and thorough feedback. We have now included the ORCID IDs of all corresponding authors and updated their email addresses to reflect their current institutional affiliations.

Prof. Changhe Shi
The First Affiliated Hospital of Zhengzhou University
Neurology
First affiliated hospital of Zhengzhou University, Zhengzhou, Henan, China
Henan 450000
China

12th Dec 2024

Re: EMBOJ-2024-116833R2
E3 Ubiquitin Ligase CHIP Facilitates cAMP and cGMP Signalling Cross-talk by Polyubiquitinating PDE9A

Dear Prof. Shi,

Thank you for submitting your final revised manuscript for our consideration. I am pleased to inform you that we have now accepted it for publication in The EMBO Journal.

Yours sincerely,

Hartmut Vodermaier
